# Brain structure and function link to variation in biobehavioral dimensions across the psychopathological continuum

Jasper van Oort[1,2]*, Alberto Llera[2,3], Nils Kohn[2,3], Ting Mei[2,3], Rose M Collard[1], Fleur A Duyser[1], Janna N Vrijsen[2,4], Christian F Beckmann[2,3,5], Aart H Schene[1,2], Guillén Fernández[2,3], Indira Tendolkar[1,2], Philip FP van Eijndhoven[1,2]

[1]Department of Psychiatry, Radboud University Nijmegen Medical Centre, Nijmegen, Netherlands; [2]Department of Cognitive Neuroscience, Radboud University Nijmegen Medical Centre, Nijmegen, Netherlands; [3]Donders Institute for Brain, Cognition and Behavior, Radboud University Nijmegen, Nijmegen, Netherlands; [4]Pro Persona Mental Health Care, Depression Expertise Center, Nijmegen, Netherlands; [5]Oxford Centre for Functional Magnetic Resonance Imaging of the Brain (FMRIB), University of Oxford, Oxford, United Kingdom

**Abstract** In line with the Research Domain Criteria (RDoC) , we set out to investigate the brain basis of psychopathology within a transdiagnostic, dimensional framework. We performed an integrative structural-functional linked independent component analysis to study the relationship between brain measures and a broad set of biobehavioral measures in a sample (n = 295) with both mentally healthy participants and patients with diverse non-psychotic psychiatric disorders (i.e. mood, anxiety, addiction, and neurodevelopmental disorders). To get a more complete understanding of the underlying brain mechanisms, we used gray and white matter measures for brain structure and both resting-state and stress scans for brain function. The results emphasize the importance of the executive control network (ECN) during the functional scans for the understanding of transdiagnostic symptom dimensions. The connectivity between the ECN and the frontoparietal network in the aftermath of stress was correlated with symptom dimensions across both the cognitive and negative valence domains, and also with various other health-related biological and behavioral measures. Finally, we identified a multimodal component that was specifically associated with the diagnosis of autism spectrum disorder (ASD). The involvement of the default mode network, precentral gyrus, and thalamus across the different modalities of this component may reflect the broad functional domains that may be affected in ASD, like theory of mind, motor problems, and sensitivity to sensory stimuli, respectively. Taken together, the findings from our extensive, exploratory analyses emphasize the importance of a dimensional and more integrative approach for getting a better understanding of the brain basis of psychopathology.

*For correspondence:
jasper.vanoort@radboudumc.nl

Competing interest: The authors declare that no competing interests exist.

## Editor's evaluation

This study presents a valuable method for performing an integrative structural-functional linked ICA analysis, investigating the relationship between the brain and a large set of symptoms and other biobehavioral measures transdiagnostically. The results show relations between multi-modal and unimodal independent components with the presence of autism spectrum disorder and variations in cognitive functioning and negative affect across individuals suffering from mood, anxiety, substance dependence, autism spectrum, or ADHD. Overall, the results are compelling and interesting to a wide readership.

## Introduction

Over the last decades, functional and structural imaging methods have helped to elucidate the brain circuits involved in psychiatric disorders. However, a growing number of concerns have been raised (*Cuthbert and Kozak, 2013*). First, the majority of neuroimaging studies in psychiatry have investigated group-level differences between traditional psychiatric diagnoses and healthy controls, often limited to one neuroimaging modality. As a consequence, we are confronted with an overwhelming amount of isolated findings, often unreplicated and without a clear understanding of how these findings relate to each other (*Marquand et al., 2016*; *Specht, 2019*). Second, the identification of valid neurobiological mechanisms has been hampered by the somewhat arbitrary boundaries and heterogeneity of traditional diagnostic categories, which fail to map unambiguously on core mechanisms of psychopathology, and also fail to adequately take comorbidity into account (*Kotov et al., 2017*). The absence of an integrative understanding of transdiagnostic core mechanisms underlying psychopathology may reduce the chance to develop biologically based, personalized forms of treatment.

The Research Domain Criteria (RDoC) have been developed to tackle the abovementioned problems by facilitating a paradigm shift from a categorical approach to a multilevel, transdiagnostic dimensional approach, with brain circuits at the central level (*Cuthbert and Insel, 2013*; *Morris et al., 2022*). Linked independent component analysis (ICA) is an innovative analysis technique, which is eminently apt to apply within this framework. Linked ICA performs multiple, simultaneous ICA factorizations that share the same unique mixing matrix and has been shown to be a powerful tool for identifying independent components (ICs) that reflect patterns of shared variance across multiple neuroimaging modalities (*Groves et al., 2012*, *Groves et al., 2011*). This allows for a principled integration of information from these imaging modalities at an early stage in the analysis pipeline, rather than a post hoc combination of unimodal results at the stage of final interpretation, and may provide a more integrative understanding at the brain level (*Groves et al., 2012*, *Groves et al., 2011*). Subsequently, relationships can be studied between interindividual differences in the resulting (multimodal) imaging components and variation in behavioral dimensions and psychopathology (*Llera et al., 2019*; *Wolfers et al., 2017*) (i.e. between the brain circuits level and the other units of analysis of the RDoC matrix).

Linked ICA has already established relationships of interindividual differences in multimodal brain components with behavioral measures and clinical profiles in specific patient groups, such as autism spectrum disorder (ASD), attention-deficit hyperactivity disorder (ADHD), and Huntington's disease (*Garcia-Gorro et al., 2019*; *Itahashi et al., 2015*; *Wolfers et al., 2017*). While linked ICA has provided a more coherent insight into the neural mechanisms within these specific populations, it has not yet been used to investigate core mechanisms of psychopathology transdiagnostically. It is important to adopt a transdiagnostic approach since the same disturbances in structural and functional brain networks may be associated with core symptom domains that transcend traditional disorder categories (*Menon, 2011*).

Therefore, in this study, we used linked ICA within a transdiagnostic, dimensional framework, using the MIND-Set database (Measuring Integrated Novel Dimensions in Neurodevelopmental and Stress-related Mental Disorders) (*van Eijndhoven et al., 2021*). This database provides us with a sample of mentally healthy participants and patients with diverse, highly prevalent non-psychotic psychiatric disorders (i.e. mood disorders, anxiety disorders, addiction, ASD, ADHD, and their comorbidity). A multimodal imaging battery was performed, and all participants were deeply phenotyped. An extensive set of biobehavioral measures was collected, including symptom dimensions, biological/physiological measures (like cortisol and heart rate variability), and also more general measures of physical and mental health. Since psychopathology can best be understood on a continuum from health to mental illness, we also included mentally healthy participants, which allows us to study the brain and biobehavioral dimensions of interest along a wider range from health to psychopathology (*Kotov et al., 2017*; *Morris et al., 2022*; *van Oort et al., 2022*).

We investigated the diverse non-psychotic psychiatric disorders (i.e. mood disorders, anxiety disorders, addiction, ASD, and ADHD) together as high levels of comorbidity suggest shared underlying mechanisms (*van Eijndhoven et al., 2021*). In addition, numerous studies that investigated these disorders separately provide converging evidence that symptoms across various major domains cut across the diagnostic boundaries of these disorders, with, among others, transdiagnostic symptom dimensions related to the negative valence domain (e.g. repetitive negative thinking) and cognitive

systems domain (e.g. regarding cognitive control problems) (*Kerns et al., 2015*; *Koob and Schulkin, 2019*; *McTeague et al., 2016*; *Richey et al., 2015*; *Shaw et al., 2014*; *van Eijndhoven et al., 2021*; *Woody and Gibb, 2015*). Interestingly, disturbances in the same brain networks may underlie these core symptom dimensions across these diverse disorders (*McTeague et al., 2016*; *Menon, 2011*). Therefore, it is crucial to adopt a transdiagnostic approach in order to identify the brain basis for symptom dimensions that transcend these diagnostic categories.

In our multimodal neuroimaging battery, we included functional scans under conditions of rest as well as under conditions of mild experimentally induced stress. It is clinically well established that vulnerability to stress is a common feature across a broad range of psychiatric disorders (*Ingram and Luxton, 2005*). Mood and anxiety disorders are characterized by a maladaptive stress response as their central feature (*de Kloet et al., 2005*; *Sharma et al., 2016*). Vulnerability to stress also plays a key role in addiction disorders, with impaired coping with stress being implicated in the onset, maintenance, and relapse in these disorders (*Koob, 2003*; *Koob and Schulkin, 2019*). While neurodevelopmental disorders have a relatively stable, trait-like course, there are clear indications for increased stress sensitivity, as exemplified by arousal and emotion regulation problems, which in turn may lead to the development of negative valence symptoms and stress-related comorbidity (*Kerns et al., 2015*; *Richey et al., 2015*; *Rommelse et al., 2011*; *Shaw et al., 2014*; *van Eijndhoven et al., 2021*). We included stress scans within the linked ICA setup as a novel feature of our approach as we hypothesized that shared mechanisms of stress vulnerability would become visible under conditions of stress.

Taken together, we set out to perform a transdiagnostic structural–functional linked ICA analysis across the spectrum from health to diverse non-psychotic psychiatric disorders in order to discover components with shared variance in brain structure and function. Our extensively phenotyped sample made it possible to subsequently perform correlational analyses to investigate how interindividual differences in the neuroimaging components relate to a broad set of biobehavioral measures. Together, these analyses allow us to identify neuroimaging components that are important for the understanding of transdiagnostic biobehavioral dimensions. Because of our primary interest in psychopathology, we decided in advance to focus on the neuroimaging components that are associated with measures of psychopathology and to focus especially on the transdiagnostic symptom dimensions.

## Results

### Study population and general results

Of the 295 participants that were included in this study, the median age was 32 years (range: 18–74 years) and 56.6% of participants were male (see *Table 1* for demographic and clinical characteristics). Of these participants, 70 were mentally healthy and 225 were patients with one or more psychiatric disorder(s). The patients had diagnoses in the following categories: current mood disorder (n = 116), anxiety disorder (n = 63), addiction disorder (n = 59), ASD (n = 63), and ADHD (n = 93) (see *Figure 1* for a Venn diagram displaying the high rate and diverse patterns of comorbidity).

The analyses confirmed that our experimentally well-controlled stressor induced mild psychological stress, with an increase in both subjective stress (median subjective stress score after neutral movie: 3, after aversive movie: 5; T = –12.50, p<0.001) and heart rate (median heart rate during neutral movie [beats per minute]: 65.59, during aversive movie: 67.10; T = –8.60, p<0.001).

### Linked ICA decomposition and correlational results

Linked ICA was used to decompose the MRI data into 50 independent components (ICs) (*Figure 2*, operations A, B, and C). Of these 50 components, 15 were multimodal components, reflecting shared variance across modalities (*Figure 3*). The correlational analysis (*Figure 2*, operation D) resulted in 87 significant correlations (false discovery rate [FDR]-corrected q < 0.001) between the components and measures of interest (all p-values mentioned below are FDR-corrected values [unless mentioned otherwise]; see *Supplementary file 2a* for all significant correlations). Of these 87 correlations, 19 were with multimodal components. Most of these correlations were related to age, sex, body mass index (BMI), blood pressure, and heart rate variability. Furthermore, we identified a multimodal component that was associated with a classification of ASD (IC32) and a multimodal component associated with cognitive symptoms (inhibition and self-monitoring) (IC30). In addition, there were eight more significant correlations between the ICs and symptom dimensions, which were all with unimodal components

**Table 1.** Demographics and clinical characteristics.

| | Total subject group (n = 295) | Patients (n = 225) | Healthy controls (n = 70) |
|---|---|---|---|
| **Demographics** | | | |
| Age (years) (median, range) | 32 (18–74) | 32 (18–74) | 32 (20–70) |
| Sex, % male (M/F) | 56.6% (167/128) | 59.6% (134/91) | 47.1% (33/37) |
| Level of education<br>No (n = .., (%))<br>Low (n = .., (%))<br>Middle (n = .., (%))<br>High (n = .., (%)) | 1 (0.0%)<br>41 (13.9%)<br>125 (42.4%)<br>128 (43.4%) | 1 (0.0%)<br>37 (16.4%)<br>103 (45.8%)<br>84 (37.3%) | 0 (0.0%)<br>4 (5.7%)<br>22 (31.4%)<br>44 (62.9%) |
| **Symptom questionnaires** | | | |
| **IDS-SR (median, range)** | | | |
| Mood/cognition | 13 (0–42) | 18 (0–42) | 1 (0–15) |
| Anxiety/somatic | 4 (0–17) | 5 (0–17) | 1 (0–5) |
| Sleep | 2 (−3–9) | 2 (−3–9) | 1 (−1–5) |
| **ASI (median, range)** | | | |
| Physical concerns | 3 (0–23) | 4 (0–23) | 1 (0–13) |
| Mental incapacitation concerns | 2 (0–16) | 3 (0–16) | 0 (0–5) |
| Social concerns | 4 (0–12) | 5 (0–12) | 3 (0–8) |
| **PTQ (median, range)** | | | |
| Core characteristics | 21 (0–36) | 22 (0–36) | 12 (0–23) |
| Unproductiveness | 6 (0–12) | 7 (0–12) | 3 (0–8) |
| Capturing mental capacity | 6 (0–12) | 6 (0–12) | 2 (0–7) |
| **CAARS (median, range)** | | | |
| Inattention/memory problems | 7 (0–15) | 8 (0–15) | 2 (0–8) |
| Hyperactivity/restlessness | 5 (0–15) | 6 (0–15) | 2 (0–8) |
| Impulsivity/emotional lability | 5 (0–15) | 6 (0–15) | 1 (0–7) |
| Problems with self-concept | 7 (0–15) | 8 (0–15) | 2 (0–7) |
| **AQ-50 (median, range)** | | | |
| Social skill | 24 (10–40) | 25 (10–40) | 17 (11–26) |
| Difficulty with change/attention switching | 25 (12–40) | 27 (14–40) | 19 (12–28) |
| Communication | 22 (11–39) | 23 (11–39) | 17 (11–24) |
| Imagination | 22 (12–37) | 22 (12–37) | 19.5 (13-28) |
| Attention to detail | 23 (10–40) | 24 (11–40) | 19.5 (10–31) |
| **TAS-20 (median, range)** | | | |
| Difficulty describing feelings | 15 (5–25) | 17 (5–25) | 11 (5–23) |
| Difficulty identifying feelings | 16 (7–32) | 18 (7–32) | 9 (7–18) |

*Table 1 continued on next page*

*Table 1 continued*

| | Total subject group (n = 295) | Patients (n = 225) | Healthy controls (n = 70) |
|---|---|---|---|
| Externally oriented thinking | 19 (9–35) | 19 (9–35) | 19 (11–30) |
| **PID-5 (median, range)** | | | |
| Negative affect | 7 (0–15) | 8 (0–15) | 2 (0–8) |
| Detachment | 5 (0–15) | 6 (0–15) | 1 (0–8) |
| Antagonism | 2 (0–12) | 2 (0–12) | 1 (0–7) |
| Disinhibition | 3 (0–15) | 4 (0–15) | 0.5 (0–6) |
| Psychoticism | 4 (0–15) | 5 (0–15) | 0 (0–6) |
| BRIEF-A (median, range) | | | |
| Inhibition | 14 (8–23) | 15 (8–23) | 10 (8–17) |
| Shift | 12 (6–18) | 13 (6–18) | 7.5 (6-12) |
| Emotional control | 17 (10–30) | 18 (10–30) | 11 (10–19) |
| Self-monitor | 9 (6-17) | 10 (6–17) | 7 (6-14) |
| Initiate | 16 (8–24) | 17 (8–24) | 10 (8–19) |
| Working memory | 16 (8–24) | 17 (8–24) | 10 (8–16) |
| Plan/organize | 19 (10–30) | 20 (10–30) | 12 (10–22) |
| Organization of materials | 15 (8–24) | 16 (8–24) | 12 (8–21) |
| Task monitor | 12 (6–18) | 12 (6–18) | 9 (6-15) |

ASI: Anxiety Sensitivity Index, AQ-50: Autism spectrum Quotient-50, BRIEF-A: Behavior Rating Inventory Executive Function – Adult, CAARS: Conners' Adult ADHD Rating Scale, F: female, IDS-SR: Inventory of Depressive Symptomatology Self Report, M: male, PID-5: Personality Inventory for DSM-5-Short Form, PTQ: Perseverative Thinking Questionnaire, TAS-20: Toronto Alexithymia Scale-20.

(see *Appendix 1—figure 1* for scatterplots displaying all 10 significant correlations between ICs and symptom dimensions). Interestingly, all symptom correlations were discovered for components that were mainly driven by functional scans.

Because of our primary interest in psychopathology, below we further discuss the ICs that have both interesting neuroimaging aspects and are also associated with psychopathology. First, we focus on the multimodal component associated with ASD, before turning to several components associated with symptom dimensions.

## Multimodal component associated with ASD

The analysis revealed a multimodal component associated with ASD (IC32). The relative contributions from the different modalities to this component were 10.1% for voxel-based morphometry (VBM), 8.8% fractional anisotropy (FA), 6.1% mean diffusivity (MD), 17.5% default mode network (DMN) (rest), 11.1% DMN (stress), 17.9% DMN (stress-aftermath), 6.3% frontoparietal network (FPN) (rest), 5.4% FPN (stress), 7.8% FPN (stress-aftermath), 3.8% executive control network (ECN) (rest), 1.6% ECN (stress), and 3.6% ECN (stress-aftermath) (*Figure 4A*). Besides the relatively large contributions to this component from the DMN modalities themselves, various other modalities also showed the involvement of regions of the DMN, highlighting its centrality within this component. The DMN modalities during the different functional scans (i.e. resting-state, stress, and stress-aftermath scan) showed similar spatial configurations, meaning that these different functional scans identified the same connectivity pattern. The DMN modalities revealed loadings in multiple brain regions that are part of (or commonly associated with) the DMN (i.e. the angular gyrus, precuneus, supramarginal gyrus). Thus this reflects connectivity of the DMN network template that was applied into dual regression with DMN (associated) regions (i.e. within DMN connectivity). Furthermore, the VBM, FA, FPN, and ECN spatial maps

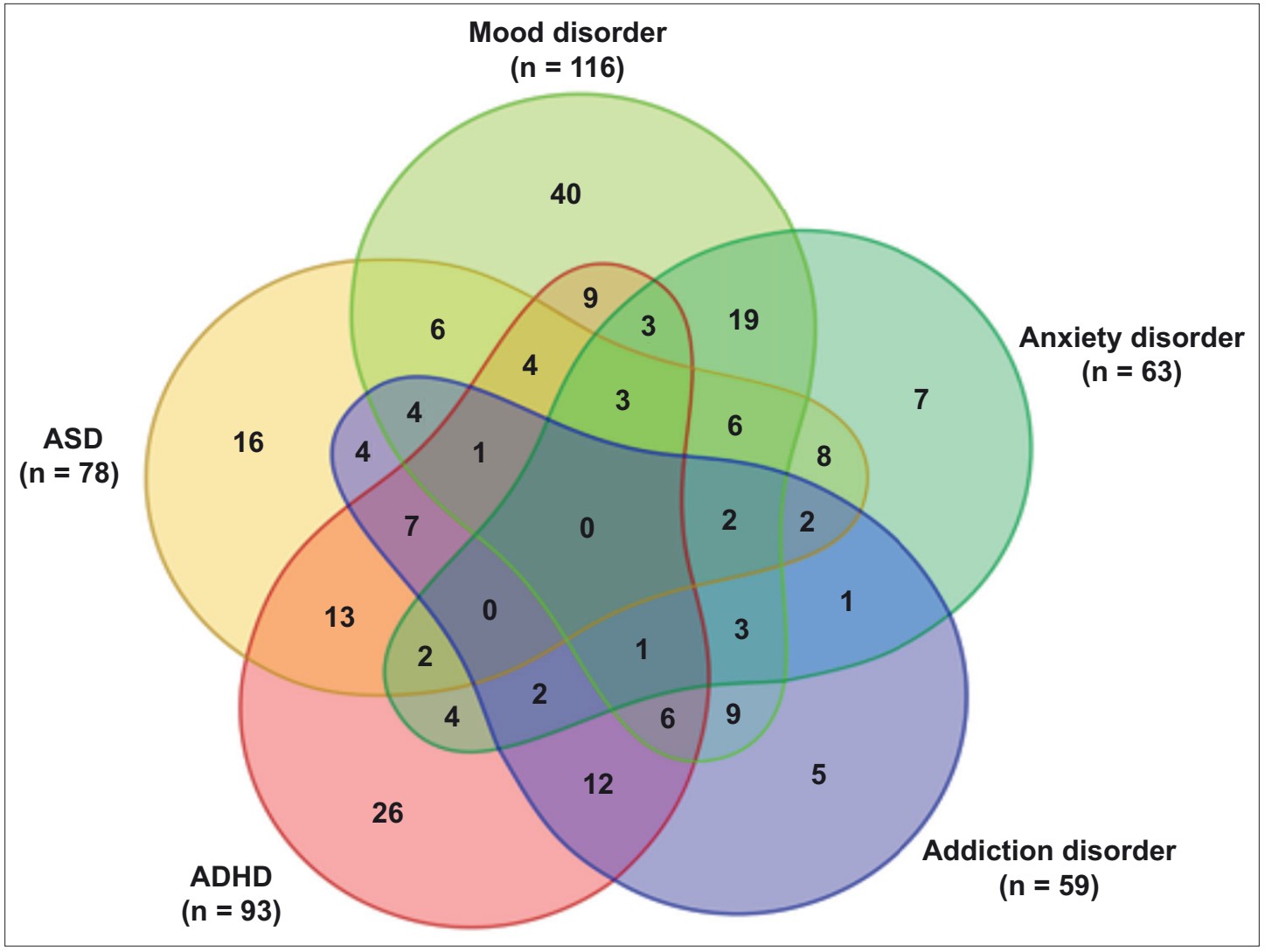

**Figure 1.** Venn diagram displaying the distribution of the psychiatric patients over the different diagnostic categories: mood disorder, anxiety disorder, addiction disorder, attention-deficit hyperactivity disorder (ADHD), and autism spectrum disorder (ASD). All diagnoses in this Venn diagram represent current diagnoses.

all showed loadings on brain regions that are part of the DMN (VBM: angular gyrus, precuneus, posterior cingulate cortex, medial prefrontal cortex; FA: angular gyrus; both FPN and ECN: angular and supramarginal gyrus, precuneus, medial prefrontal cortex). Additionally, the VBM and MD feature showed the involvement of the precentral gyrus and thalamus respectively (*Figure 4B*). The correlational analysis showed that this component was positively correlated with a classification of ASD ($r_s$ = 0.19, p=0.044), indicating that ASD is associated with higher subject loadings on this component. Hereafter, this component will be called the *multimodal ASD component*.

While this *multimodal ASD component* was associated with a classification of ASD, it was not correlated with any of the subscales of the Autism spectrum Quotient-50 (AQ-50). To further explore these results, we performed post hoc correlations between IC32 and the AQ-50 subscales for the patients with a classification of ASD and the participants without ASD separately (uncorrected for multiple comparisons). The only significant correlation was found within the ASD group for the 'social skill'-subscale ($r_s$ = –0.237, $p_{uncorrected}$ = 0.037) (see Appendix 1 for all correlational results).

## Correlations between components and symptom dimensions

All 10 correlations between symptom dimensions and ICs were with ICs that were driven by functional scans. Moreover, 9 out of 10 correlations were with components that have an important contribution

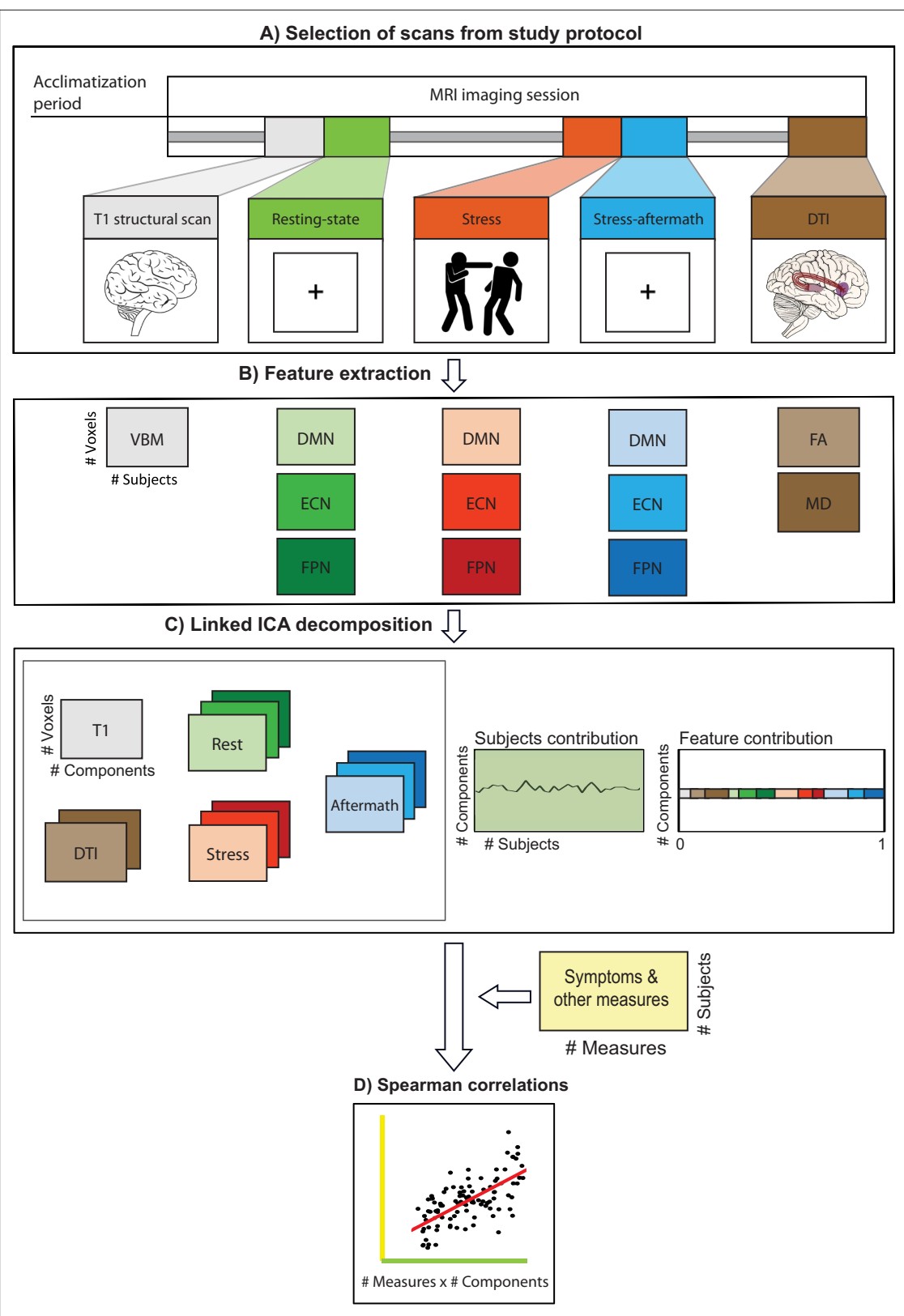

**Figure 2.** Data processing pipeline. (**A**). Experimental design: subjects entered the scanner after a 45 min acclimatization period outside the scanner. The whole MIND-Set MRI protocol consists of a series of scans, of which we selected the following scans for the present study: two structural scans: T1 structural scan and diffusion tensor imaging (DTI) scan. Furthermore, we selected three functional scans, representing a baseline resting-state scan (rest), the scan during stress induction with an aversive movie clip (stress scan), and the resting-state scan directly after the stress induction, which will

*Figure 2 continued*

be referred to as the stress-aftermath scan. (**B**) The relevant features were extracted from the selected scans. From the structural scans: voxel-based morphometry (VBM), fractional anisotropy (FA), and mean diffusivity (MD). From each functional scan, we extracted the whole-brain spatial maps of our networks of interest: default mode network (DMN), executive control network (ECN), and frontoparietal network (FPN). (**C**) These features were used as input in the linked ICA algorithm. (**D**) Spearman correlations were performed between the subject loadings of each independent component and all the (bio)behavioral measures of interest (i.e. symptom questionnaires, demographics, other biobehavioral measures). (This figure is inspired by the figure of *Llera et al., 2019*.)

from the ECN. We start by discussing four components (IC7, IC8, IC13, and IC30) that are driven by the ECN during different functional scans and show interesting similarities in their connectivity profiles, before turning to a component driven by the DMN during the stress scan.

## Components reflecting connectivity between the ECN and FPN

Linked ICA resulted in four components that reflect connectivity of the ECN with itself and with the FPN (IC7, IC8, IC13, and IC30). Three of these components are unimodal components that are each driven by the ECN modality during a distinct functional scan (IC7 by the ECN during the stress-aftermath scan [99.5% of contribution] [*Figure 5*], IC8 by the ECN during the stress scan [99.8%], and IC13 by the ECN during the resting-state scan [99.9%] [*Appendix 1—figure 2*]). All three of these components reflect the connectivity of the ECN with itself and the right FPN. While there are important similarities between these components, there are also differences. Compared to IC7, IC8

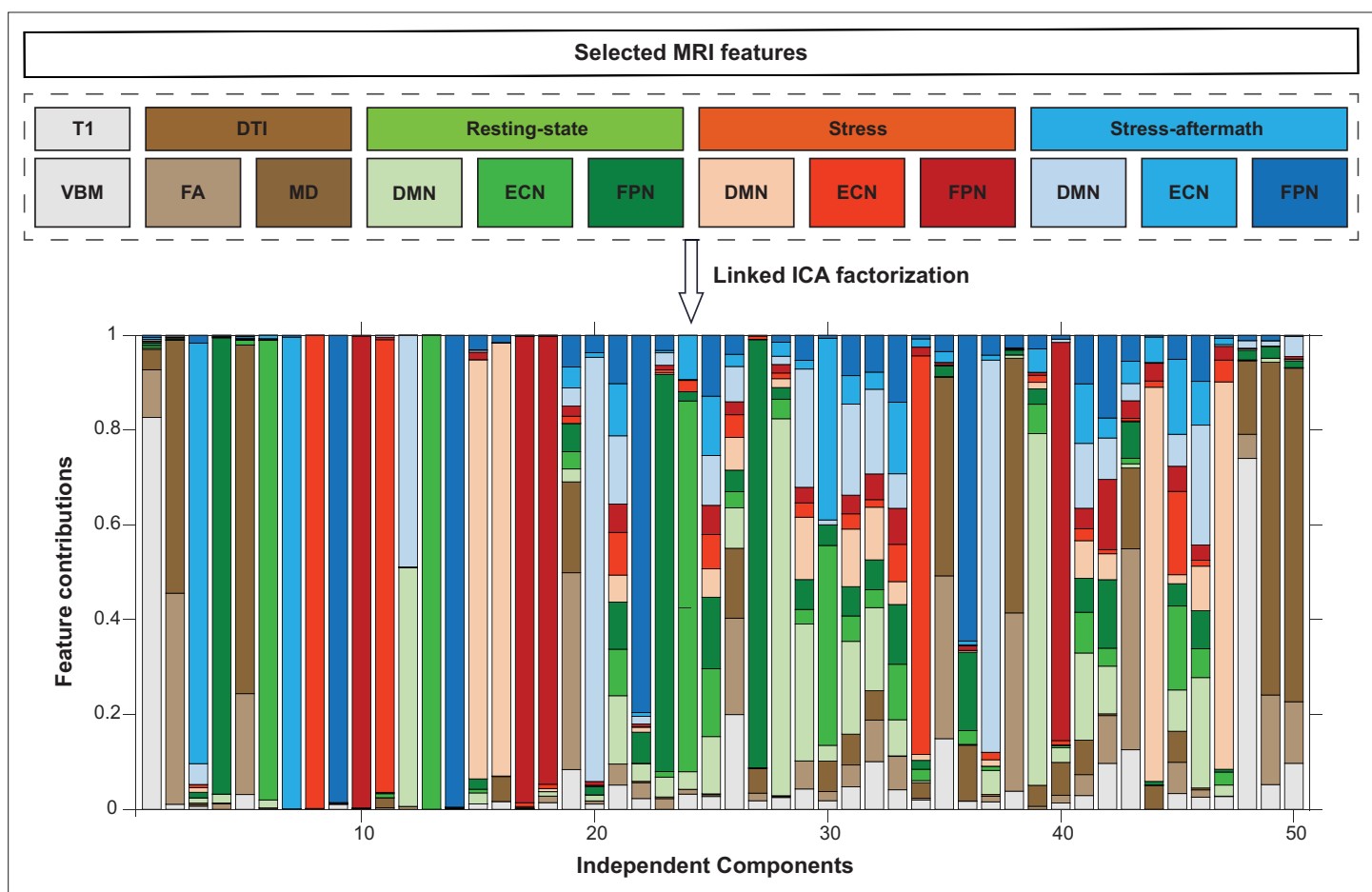

**Figure 3.** Linked independent component analysis (ICA) decomposition. Linked ICA was used to simultaneously factorize the selected MRI features into 50 independent components. The stacked bargraph displays to what extent these independent components are driven by the different imaging features. DMN: default mode network; DTI: diffusion tensor imaging scan; ECN: executive control network; FA: fractional anisotropy; FPN: frontoparietal network; MD: mean diffusivity; VBM: voxel-based morphometry.

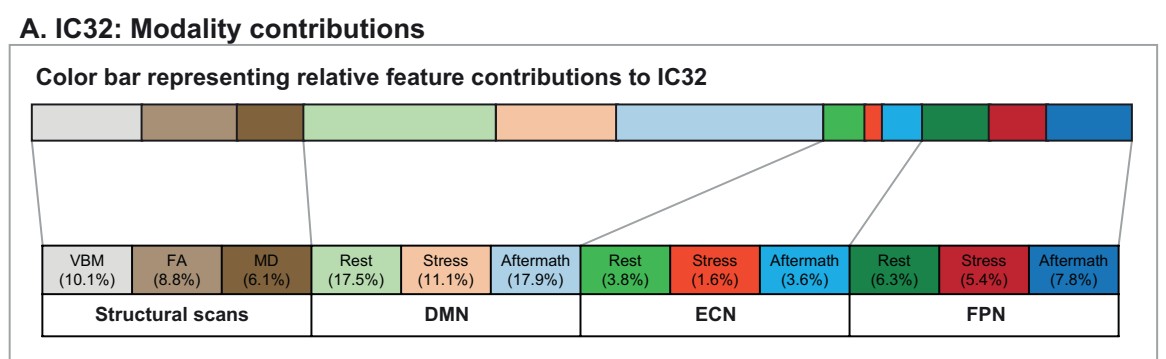

**Figure 4.** Multimodal component 32 (IC32). (**A**) Modality contributions to IC32. The color bar is a graphical representation of the relative contribution of the 12 feature modalities to IC32. The numbers below state the exact percentages of the different modality contributions. (**B**) The subject loadings on IC32 have a significant correlation with a psychiatric classification of autism (yes/no). This component is a multimodal component, with contributions from all 12 features. From top to bottom, we visualize voxel-based morphometry (VBM), fractional anisotropy (FA), mean diffusivity (MD), and the spatial

*Figure 4 continued on next page*

*Figure 4 continued*

maps for the following networks: the default mode network (DMN), executive control network (ECN), and frontoparietal network (FPN). Since the three functional networks of interest showed similar spatial configurations during the different functional scans (i.e. resting-state, stress, and stress-aftermath scan), we only display the spatial maps from one functional scan here (i.e. the stress-aftermath scan). Note: in this figure, the right side of the brain is displayed on the right side of the image. R: right.

The online version of this article includes the following source data for figure 4:

**Source data 1.** Correlation for IC32.

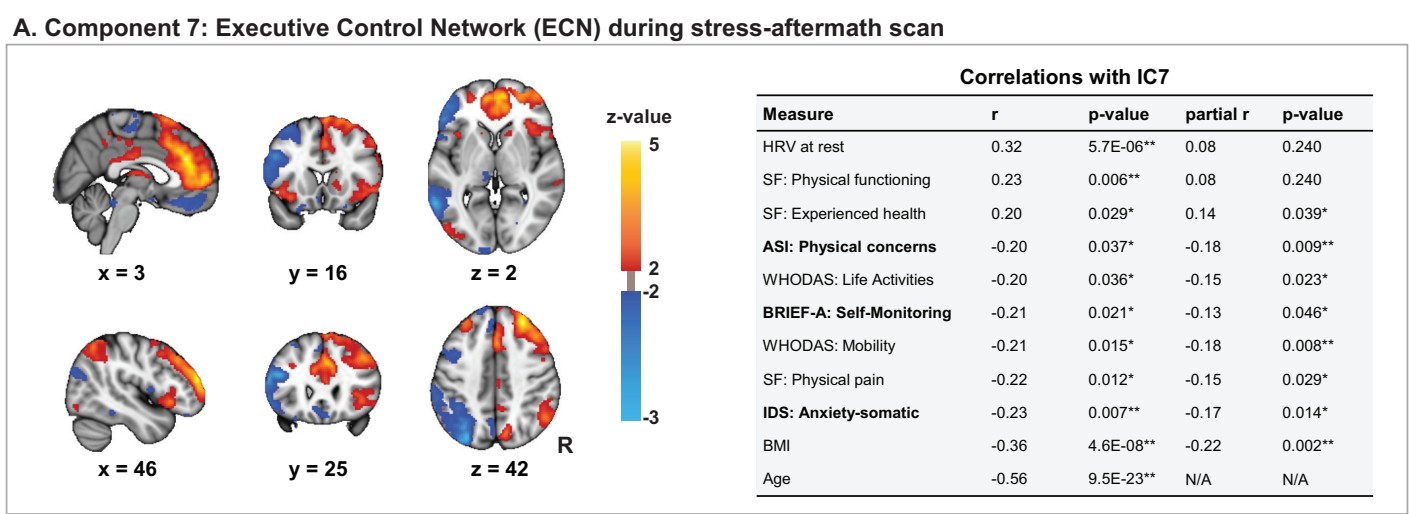

## A. Component 7: Executive Control Network (ECN) during stress-aftermath scan

### Correlations with IC7

| Measure | r | p-value | partial r | p-value |
|---|---|---|---|---|
| HRV at rest | 0.32 | 5.7E-06** | 0.08 | 0.240 |
| SF: Physical functioning | 0.23 | 0.006** | 0.08 | 0.240 |
| SF: Experienced health | 0.20 | 0.029* | 0.14 | 0.039* |
| **ASI: Physical concerns** | -0.20 | 0.037* | -0.18 | 0.009** |
| WHODAS: Life Activities | -0.20 | 0.036* | -0.15 | 0.023* |
| **BRIEF-A: Self-Monitoring** | -0.21 | 0.021* | -0.13 | 0.046* |
| WHODAS: Mobility | -0.21 | 0.015* | -0.18 | 0.008** |
| SF: Physical pain | -0.22 | 0.012* | -0.15 | 0.029* |
| **IDS: Anxiety-somatic** | -0.23 | 0.007** | -0.17 | 0.014* |
| BMI | -0.36 | 4.6E-08** | -0.22 | 0.002** |
| Age | -0.56 | 9.5E-23** | N/A | N/A |

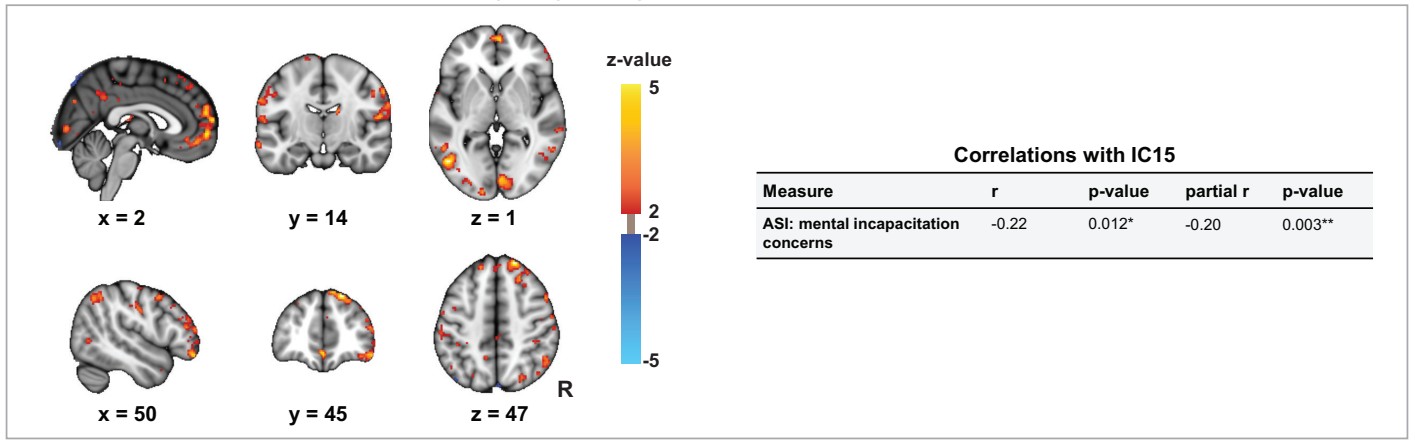

## B. Component 15: Default Mode Network (DMN) during stress scan

### Correlations with IC15

| Measure | r | p-value | partial r | p-value |
|---|---|---|---|---|
| **ASI: mental incapacitation concerns** | -0.22 | 0.012* | -0.20 | 0.003** |

**Figure 5.** ECN-stress aftermath component (IC7) and DMN-stress component (IC15). (**A**) Independent component 7 (IC7) is driven by the executive control network (ECN) during the stress-aftermath scan (99.5%). This component reflects the connectivity of the ECN with itself and with the right frontoparietal network (FPN) and has significant (partial) Spearman correlations with several symptoms and other measures of interest. For this component, we used the 10th and 90th percentiles for thresholding, for display purposes, since the underlying distribution was not z-distributed. (**B**) Independent component 15 (IC15) is mainly driven by the default mode network (DMN) during the stress scan (88.4%). IC15 negatively correlates with the fear of losing control/losing one's mind under stress (mental incapacitation concerns subscale). In this figure, the right side of the brain is displayed on the right side of the image.ASI: Anxiety Sensitivity Index;BRIEF-A: Behavior Rating Inventory Executive Function – Adult; HRV: heart rate variability; IDS: Inventory of Depressive Symptomatology Self Report; R: right; SF: Short Form-20; WHODAS: WHO-Disability Assessment Schedule 2.0. Cave: in general, a higher score on a questionnaire reflects more severe symptoms/problems, except for the SF subscales 'experienced health' and 'physical functioning,' for which this is reversed.

The online version of this article includes the following source data for figure 5:

**Source data 1.** Correlations for IC7 and IC15.

loaded less on the right posterior parietal cortex (FPN) and IC13 loaded less on the frontal poles and thalamus. Finally, IC30 is a multimodal component, mainly driven by the ECN during the resting-state scan (42.2%) and stress-aftermath scan (38.3%) (*Appendix 1—figure 2*). This component loaded on the ECN and left FPN (contralateral FPN compared to components described above). Interestingly, the spatial maps during both these functional scans are largely the same, suggesting that the same variance is picked up in both these functional scans.

All four of these ECN components are negatively correlated with cognitive symptoms, indicating that a lower subject loading on these components is associated with more severe symptoms (see Appendix 1 for additional information on the symptom dimensions mentioned below). All four of these components are negatively correlated with self-monitoring (IC7: $r_s$ = –0.21, p=0.021; IC8: $r_s$ = –0.22, p=0.008; IC13: $r_s$ = –0.20, p=0.038; IC30: $r_s$ = –0.22, p=0.012). Additionally, IC13 also negatively correlated with working memory ($r_s$ = –0.22, p=0.012) and IC30 with inhibition ($r_s$ = –0.19, p=0.044).

While all four components are negatively correlated with cognitive symptoms, IC7 is the only component with a negative correlation with symptoms from the negative valence domain. This component negatively correlated with the following symptom dimensions: anxiety/somatic ($r_s$ = –0.23, p=0.007) and physical concerns ($r_s$ = –0.20, p=0.037). In parallel with the correlations with the symptom dimensions, a lower subject loading on the *ECN-stress aftermath component* (IC7) was also associated with various measures that are generally associated with worse health, such as higher age ($r_s$ = –0.56, p=9.5E-23), higher BMI ($r_s$ = –0.36, p=4.6E-8), lower heart rate variability ($r_s$ = 0.32, p=5.7E-6), more physical pain ($r_s$ = –0.22, p=0.012), and worse experienced health ($r_s$ = 0.20, p=0.029).

Next, we performed post hoc tests in order to explore whether the correlations between the *ECN-stress aftermath component* (IC7) and the negative valence symptoms differed from the correlations between the other ECN-FPN components (i.e. IC8, IC13, and IC30) and these same negative valence symptoms (using Fisher's r to z transform; alpha = 0.05). We refer to Appendix 1 for a complete overview of these results. Here, we would like to note that the results showed that the correlations did not differ between the *ECN-stress aftermath component* (IC7) and IC13 (ECN during the resting-state scan) (IDS anxiety/somatic: IC7 $r_s$ = –0.23; IC13: $r_s$ = –0.14, z = –1.07, $p_{uncorrected}$ = 0.284; ASI physical concerns: IC7: $r_s$ = –0.20; IC13: $r_s$ = –0.10; z = –1.17, $p_{uncorrected}$ = 0.242). Based on the results of these post hoc analyses, we cannot exclude that it is a threshold effect that we only found these relationships with negative valence symptoms for the *ECN-stress aftermath component*. At the same time it is important to note that in our analyses relatively few results for the symptom questionnaires survived multiple comparison correction, and that the *ECN-stress aftermath component* was the most sensitive component (of these ECN-FPN components) for finding relationships with the negative valence symptoms. Thus, the stress induction may have played an important role in revealing these results at a statistically significant level.

## DMN connectivity during stress scan

IC15 was driven by the DMN during the stress scan (88.4%). This IC showed widespread connectivity patterns of the DMN, including within DMN connectivity (medial prefrontal cortex, posterior cingulate cortex), and connectivity with the FPN (posterior parietal cortex, dorsolateral prefrontal cortex) and with the visual regions (occipital cortex). This component negatively correlated with the fear of losing control/one's mind under stress (mental incapacitation concerns) ($r_s$ = –0.22, p=0.012) (*Figure 5B*). In the 'Discussion' section, this IC is called the *DMN-stress component*.

## Discussion

In this study, we performed a structural–functional linked ICA analysis to get a more complete understanding of psychopathology from a transdiagnostic perspective, investigating a sample with both mentally healthy participants and psychiatric patients with diverse non-psychotic disorders. Our deeply phenotyped sample allowed us to investigate which brain components may in particular be important for the understanding of transdiagnostic mechanisms in psychopathology by performing extensive, exploratory correlational analyses. Linked ICA resulted in various multimodal components, uncovering shared variance across modalities. While these multimodal components were in particular related to age, sex, BMI, blood pressure, and heart rate variability, we also identified multimodal components that were associated with cognitive symptoms and a diagnosis of ASD. Interestingly, the

transdiagnostic symptom dimensions were most strongly related to components that were driven by the large-scale functional networks during the various functional scans.

In line with *Llera et al., 2019* and several other linked ICA studies (*Garcia-Gorro et al., 2019*; *Itahashi et al., 2015*; *Wolfers et al., 2017*), we chose to not only focus on the multimodal components but to perform the correlational analyses for all the components that resulted from the linked ICA algorithm. This way, our unbiased analysis was able to identify multimodal components that were related to important aspects of psychopathology (like our multimodal ASD component), but also unimodal components that correlated with a wide range of biobehavioral measures, like our ECN stress-aftermath component. While this ECN-stress aftermath component may contribute less to a more integrative understanding at the brain level, it does contribute to a more integrative understanding of psychopathology by showing how this component is related to various biobehavioral measures at different levels of the RDoC matrix (like physiology and self-reports) and also across different domains (i.e. the negative valence and cognitive systems domain) (*Cuthbert and Insel, 2013*). Below, we continue with a more in-depth discussion of the components that are most important for the understanding of psychopathology. First, we discuss the multimodal ASD component, before turning to the components related to the transdiagnostic symptom dimensions across the negative valence and cognitive systems domains.

Although set within transdiagnostic research, the linked ICA analysis was sensitive enough to pick up a multimodal component that is associated with a traditional, diagnostic classification of ASD. This relationship was specifically found for ASD as this component was not associated with the other diagnostic groups, nor with the variable that divided our sample in mentally healthy participants versus patients. The multimodal ASD component loaded on the DMN, precentral gyrus, and thalamus. Interestingly, these regions have been implicated in ASD by earlier studies and are associated with core domains of this disorder, like theory of mind (*Murdaugh et al., 2012*), motor problems (*Duffield et al., 2013*; *Mahajan et al., 2016*), and sensitivity to sensory stimuli, respectively (*Ayub et al., 2021*). While both *Itahashi et al., 2015* and *Mei et al., 2022* also implicated these regions in ASD in their linked ICA analysis, there were also differences with our findings. In the study of *Itahashi et al., 2015*, these regions did not show up together in one component, and the multimodal component of *Mei et al., 2022* loaded more extensively on the white matter tracts. While the differences with our findings may be related to the differences in study setup and specific sample characteristics (e.g. related to the included MRI modalities, patterns of psychiatric comorbidity, IQ, and age range of the participants), this may also be related to the heterogeneous nature of psychiatric classifications like ASD (*Itahashi et al., 2015*). Still, these multimodal results may help to get a more coherent understanding of the neurobiology of ASD by not only showing which brain regions are involved but also how these findings covary across different modalities.

Importantly, this *multimodal ASD component* was associated with a diagnostic label of ASD, but not with the subscales of the AQ-50 across the whole sample. As a diagnosis of ASD reflects a complex clinical phenotype that spans several functional domains, a multimodal brain component spanning brain regions that are involved in these various functions may be more strongly associated with such a complex phenotype than with specific symptom dimensions (*Mei et al., 2022*). While the AQ-50 subscales did not show any correlations with this *multimodal ASD component* across the whole sample, the post hoc tests showed that this component was specifically associated with social skill symptoms within the ASD group. This may be explained by the loading of this component in the DMN, which is involved in social functions related to theory of mind (*Murdaugh et al., 2012*). Based on these results, an alternative explanation for not finding correlations for the AQ-50 across the whole sample could be related to the limitations regarding the application of this questionnaire in our diverse transdiagnostic sample. The AQ-50 may not measure a uniform dimension across our participants. Higher scores on this self-report questionnaire may stem from ASD, but could also stem from other causes, like social interaction problems due to (social) anxiety, or patients may score higher if they have an overly negative judgment of their social skills (e.g. in depression). Thus, when investigating brain behavior relationships using the AQ-50, it may be important to also take into account the judgment of a trained clinician regarding whether a participant has ASD or not. These considerations are important since one of the central goals of RDoC is the identification of reliable and valid measures that can be applied in transdiagnostic research (*Morris et al., 2022*).

Interestingly, all three functional networks of interest (i.e. DMN, ECN, and FPN) were involved in components that showed transdiagnostic associations with cognitive and/or negative valence symptoms. While the literature thus far indeed suggests that aberrations in these large-scale networks cause major dysfunctions in core cognitive and affective domains across psychiatric disorders, this idea was largely based on indirect evidence in separate psychiatric disorders (*Menon, 2011*; *Sanislow et al., 2010*). We now identified these relationships in a diverse sample spanning the psychopathological continuum.

The inclusion of a stress challenge is a novel feature in our linked ICA analysis. While several components that reflect the connectivity between the ECN and FPN (including the resting-state component) were associated with cognitive symptoms, two components that were related to the stress induction were most sensitive in revealing relationships with negative valence symptoms (i.e. the *ECN-stress aftermath component* and *DMN-stress component*). This could indicate that it may be especially relevant to include a stress induction paradigm when studying the negative valence domain.

Especially the ECN seems to be important in explaining transdiagnostic symptom dimensions. We found four different components (IC7, IC8, IC13, and IC30) driven by the ECN during different functional scans that reflected connectivity of the ECN with itself and with the FPN. In line with the known role of these networks in a wide range of cognitive tasks (*McTeague et al., 2016*; *Smith et al., 2009*), all four components showed significant correlations with executive/higher-order cognitive symptoms. Below, we further elaborate on our *ECN-stress aftermath component* (i.e. IC7) as this is the only one of these four components that revealed a relationship with both cognitive and negative valence symptoms.

The *ECN stress-aftermath component* reflects connectivity of the ECN with itself and with the right FPN in the aftermath of stress. Interestingly, the anterior insula and dorsal anterior cingulate cortex are part of as well this component as the ECN template (*Smith et al., 2009*), and are both considered core regions of the salience network (SN) (*Hermans et al., 2011*; *Seeley et al., 2007*; *Shirer et al., 2012*). In line with the known role of these regions in the dynamic reallocation of resources under stress (*Hermans et al., 2014*), our results suggest that the ECN engages the right FPN in the aftermath of stress for emotion regulation/top-down control (*Buhle et al., 2014*; *Hermans et al., 2014*; *Hermans et al., 2011*; *Kohn et al., 2014*; *McTeague et al., 2016*). The negative relationship between this component and negative valence symptoms suggests that an inadequate engagement of the FPN by the ECN can be seen as a crucial transdiagnostic vulnerability factor for the development of negative valence symptoms (under stress).

Our extensive, exploratory analyses allowed us to identify not only a relationship of this ECN component with symptom dimensions, but also with a broad range of negative health outcomes related to physiological/cardiovascular measures, BMI, and also general health/functioning. It is known from the literature that there are complex relationships between mental health, physical health, and coping with stress (*de Kloet et al., 2005*; *Juster et al., 2010*; *McEwen, 2003*). Physical and mental health problems may affect each other and may also result in a maladaptive stress response. Vice versa a repeatedly maladaptive stress response, including inadequate recovery in the aftermath of stress, may lead to mental and physical wear-and-tear (*Juster et al., 2010*; *McEwen, 2000*; *McEwen, 2003*). As the brain is the central organ that coordinates the stress response (*Ulrich-Lai and Herman, 2009*), our results suggest the key importance of the ECN and FPN in this complex web of relationships. Together, these results emphasize the importance of an integrative approach to stress, and physical and mental health.

The results for our DMN-stress component add to a growing body of evidence on the importance of the DMN in the stress response in health and psychopathology (*van Oort et al., 2020*; *van Oort et al., 2017*; *Zhang et al., 2019*). This component reflects the connectivity of the DMN with the FPN and visual regions during stress induction with a stressful movie clip with an eyewitness instruction. The connectivity between the DMN (which is known to be involved in self-referential processing) and visual regions may reflect the processing of the stressful movie from a self-referential/eyewitness perspective (*Buckner et al., 2008*; *van Oort et al., 2017*). The connectivity with the FPN may reflect the role of the FPN in top-down control during stress (*Kohn et al., 2014*). Interestingly, this component is negatively associated with the fear of losing one's mind under stress. Together, this may suggest that stronger connectivity of the DMN with itself, the FPN and visual regions, may be an

adaptive response to facilitate the (self-referential) processing of the stressor, while there is top-down control of this fear-related symptom.

The main strength of our study is that this is the first study performing an integrative structural–functional linked ICA analysis, investigating the relationship between the brain and broad set of symptoms and other (bio)behavioral measures transdiagnostically. However, our study has to be interpreted in light of some limitations. First, linked ICA is characterized by sign indeterminacy, which made it necessary to infer the direction of the associations between our components and biobehavioral measures of interest, by studying the relationship between global gray matter volume and age. While we think that this is a valid way of determining the direction of the relationships for our results, we would like to note here that it is a limitation that the direction of the effects is not a mathematical certainty but an inference. Future studies should focus on replicating the findings described in this study. Second, our results were found in a specific sample of patients with mainly mood, anxiety, addiction, and neurodevelopmental disorders. Future studies should investigate whether our results can be replicated in other (transdiagnostic) samples. Third, the cross-sectional nature of our study prevents us from making definitive causal inferences about the relationships that were found. Future studies should prospectively investigate the relationship between brain structure, function, and psychiatric symptoms. Finally, although we did include measures related to psychotropic medication and substance use in our correlational analyses, we did not correct for these factors in the analyses. We did not include these factors as covariates as these factors are not independent from our other measures of interest, like the symptom dimensions and diagnostic labels. Future studies in samples that are well matched on these measures of interest and only differ in medication status or substance use could help to further disentangle the effects of these factors from other aspects of psychopathology.

Taken together, the structural–functional linked ICA analysis followed by extensive correlational analysis revealed several components that are important for the understanding of psychopathology. We identified a multimodal component that was specifically associated with ASD. The involvement and covariation of the DMN, precentral gyrus, and thalamus across the different modalities of this component may reflect the broad functional domains that may be affected in ASD, like theory of mind, motor problems, and sensitivity to sensory stimuli, respectively. Furthermore, the results emphasized the importance of especially the ECN for the understanding of transdiagnostic symptom dimensions. The aftermath of stress revealed that the connectivity between the ECN and FPN was associated with symptom dimensions across both the cognitive systems and negative valence domain of the RDoC framework, and also with various other health-related (bio)behavioral measures. This suggests a key role for these networks in adaptive coping with stress and thereby for mental and physical health.

Our results provide initial insight into the neural mechanisms underlying transdiagnostic (bio)behavioral dimensions and provide avenues for future research. First, further research is necessary into which (biobehavioral) dimensions serve as reliable and valid measures for transdiagnostic research and how they can be assessed best (*Morris et al., 2022*). While various dimensions can be measured well with a self-report questionnaire, in other cases it may be important to complement these measures with more objective measures (e.g. neuropsychological tests or assessments by a trained clinician). In addition, the field of multimodal imaging is still relatively new. Multimodal imaging techniques have the potential to take maximal advantage of the strengths of the different types of imaging data, with each data type having a limited but complementary view of the brain (*Sui et al., 2012*). Since our results show the potential of the functional scans for revealing relationships with transdiagnostic symptom dimensions, future studies should investigate how multimodal analysis techniques may further capitalize on this potential of the functional scans. It should be investigated which structural and functional modalities can best be combined and how multimodal analyses techniques can be optimized in order to further the integrative understanding of brain function and structure (*Sui et al., 2012*).

## Materials and methods
### Participants

This study used the database of the MIND-Set project (*van Eijndhoven et al., 2021*), which includes adult patients with ASD, ADHD, addiction, mood, and/or anxiety disorders. Patients were included in this study if they had at least one current diagnosis in one of these categories. A mentally healthy control group was also included. Patients were diagnosed and classified by a trained clinician according

to the Diagnostic and Statistical Manual of Mental Disorders (DSM) (*American Psychiatric Association, 2013*) using semi-structured interviews (see Appendix 1 and *van Eijndhoven et al., 2021*).

## Procedure

The MIND-Set protocol contains an extensive neuroimaging battery with multiple imaging modalities, of which we used the T1 structural scan, DTI, and three functional scans for this study (*Figure 2A*). We selected the following functional scans: (1) a baseline resting-state scan, (2) the functional scan during stress induction, which hereafter will be called the 'stress scan,' and (3) the resting-state scan, directly after the stress induction. Since this last scan reflects a combination of continued stress and recovery in the aftermath of acute stress, this scan will be referred to as the 'stress-aftermath scan'.

Stress was induced with a mild psychological stressor using an experimentally well-controlled paradigm in the form of an aversive movie clip (*Qin et al., 2009*; *van Oort et al., 2020*). A neutral movie clip served as the control condition. We used the following two measures to assess stress levels during scanning: heart rate (beats per minute) and subjective stress (11-point rating scale: 0 = no stress, 10 = maximal stress). The subjective stress level was assessed directly after the aversive and neutral movie, and the heart rate was measured during these two movie clips. For these two measures, we assessed the effects of stress using a Wilcoxon signed-rank test (non-normal distribution).

## Biobehavioral measures of interest

To cover a broad range of clinically relevant dimensions, we included 80 measures for our exploratory correlational analyses, including 35 symptom dimensions from validated questionnaires and 45 other demographic and biobehavioral measures of interest. We describe these variables shortly below (see *Appendix 1—tables 1 and 2* and *van Eijndhoven et al., 2021* for an elaborate description).

We used questionnaires measuring symptom dimensions that commonly occur across neurodevelopmental, mood, anxiety, and addiction disorders. These questionnaires measure symptoms related to the following topics (questionnaire between brackets): depressive symptoms (IDS-SR), anxiety sensitivity (ASI), ADHD symptoms (CAARS), autistic traits (AQ-50), alexithymia (TAS-20), personality traits across different personality domains (PID-5-B-Adult), repetitive negative thinking (PTQ), and behavioral regulation (BRIEF-A). For the correlational analyses, we used the scores of the 35 subscales of these eight questionnaires (*Appendix 1—table 1*). These subscales represent clinically relevant symptom dimensions across the spectrum of non-psychotic psychiatric disorders. To provide further insight into the distribution of the symptom dimensions across broad diagnostic groups in our sample, we display these results in dot plots (*Appendix 1—figure 3*). For this purpose, we divided our sample into the following four subgroups: mentally healthy control group, stress-related group, neurodevelopmental group, and comorbidity group (see Appendix 1 for a description of these subgroups).

Besides the symptom questionnaires, we included a more extensive set of demographics and (bio) behavioral measures (*Appendix 1—table 2*). These measures span different units of analysis and broadly include measures from the following topics: demographics (age, sex, and level of education), anthropometric (BMI), biological/physiological measurements (e.g. saliva and hair cortisol, heart rate variability), traumatic childhood events, psychiatric classifications according to the DSM (e.g. ADHD [yes/no]), number of chronic somatic disorders, general health and functioning (i.e. SF-20 and WHODAS questionnaires), substance use (smoking, alcohol, and cannabis), and medication use (e.g. use of an antidepressant [yes/no]). We refer to *Appendix 1—table 3* for information regarding psychotropic medication use at the time of the MRI scan.

## MRI data acquisition

All images were collected using a 3T Siemens Magnetom Prisma MRI scanner (Erlangen, Germany). High-resolution structural images (1.0 mm isotropic) were acquired using a T1-weighted MP-RAGE sequence (TE/TR = 3.03/2300 ms). In addition, diffusion tensor imaging (DTI) scans were obtained using a multi-band 3 protocol (TE/TR = 70.2/2370 ms, voxel size = 2.0 mm isotropic, number of gradients = 85). For all three functional scans, T2*weighted EPI BOLD-fMRI images were acquired using a multi-band 6 protocol (TR = 1000 ms, voxel size = 2.0 mm isotropic). The resting-state and stress-aftermath scans were both 500 volumes, while the stress scan consisted of 150 volumes (see Appendix 1 for more details).

## MRI preprocessing and feature extraction

To acquire the input for the linked ICA algorithm, feature extraction operations were performed on the selected structural and functional scans (*Figure 2B*; see Appendix 1 for details regarding preprocessing and feature extraction). The T1 scans were used to estimate gray matter volumes, using VBM (*Ashburner and Friston, 2000*). FA and MD images were acquired from the DTI scan and served as measures for white matter integrity (*Jenkinson et al., 2012*; *Smith et al., 2006*). Together, this resulted in three features from the structural scans.

For all three functional scans (i.e. resting-state, stress, and stress-aftermath), we extracted the spatial maps of our three networks of interest (i.e. DMN, ECN, and FPN; with the left and right FPN merged into one FPN template). We used the network templates from *Smith et al., 2009* to select our networks of interest without biasing the results toward one of our functional scans. Dual regression was used to acquire spatial maps, which reflect the whole-brain connectivity of the networks during the different functional scans. The nine spatial maps that resulted from dual regression (3 scans × 3 networks) were used as input in the linked ICA algorithm (see Appendix 1 for a more elaborate description of the used methods).

## fMRI analyses

### Linked ICA

Linked ICA was used to simultaneously factorize the 12 MRI features of our N = 295 participants into independent sources (or components) of spatial variation (*Groves et al., 2011*). In general, the linked ICA model order is recommended to be less than 25% of the sample size (*Groves et al., 2012*; *Groves et al., 2011*). In addition, the 'optimal' dimensionality depends on the detail desired from the decomposition (*Groves et al., 2012*) as it has been shown that components that are identified with linked ICA at a lower dimensionality may split into finer subdivisions at a higher dimensionality (*Groves et al., 2012*). Because of our interest in large-scale networks, we decided a priori to choose a relatively low-dimensional decomposition. In line with the lower-dimensionality decomposition performed by *Groves et al., 2012*, we chose a priori to decompose our data into 50 independent components.

In brief, linked ICA is a Bayesian extension of ICA (*Choudrey, 2002*) to multiple input sets, where all individual ICA factorizations are linked through a shared common mixing matrix that reflects the subject-wise contribution to each component. Such factorization provided us for each component with (1) a set of spatial maps (one per feature), (2) a vector of feature loadings that describes the degree to which the component is 'driven' by the different modalities, and (3) a subject loading that describes how each individual subject contributes to a given component (*Figure 2C*). Importantly, these subject loadings can subsequently be used for the correlational analysis with the symptoms and other measures of interest. In addition to the analysis described above, in which we used all 12 MRI features together, we also performed linked ICA factorizations for the separate structural and functional imaging modalities as a supplemental analysis (see Appendix 1).

Since the vectors of feature loadings (see point '2' above) describe the degree to which each component is 'driven' by the different MRI features, these feature loadings can be used to determine whether a component is a multimodal component or not. We defined multimodal components as components that have a meaningful contribution (>10%) from two or more MRI features and no single feature contributing >50% to the total variance of the component. Linked ICA can, however, also result in unimodal components, here defined as one feature contributing >80% to the total variance and no other feature contributing >10% to the component.

### Correlational analysisis

In line with *Llera et al., 2019*, we performed full correlations between the subject loadings on the independent components, obtained by linked ICA, and our measures of interest (symptom questionnaire subscales and other measures of interest). This resulted in 50 × 80 Spearman correlations (non-normal distribution). We addressed the multiple comparisons by applying FDR correction (p<0.05) (*Benjamini and Hochberg, 1995*; *Figure 2D*). As a supplementary analysis, we performed partial Spearman correlations (correcting for age and sex) for the significant results from this main analysis (see Appendix 1). Finally, we performed Spearman correlations (non-normal distribution) between our biobehavioral measures of interest to provide further insight into these relationships (see *Supplementary file 1*).

## Direction of correlational results

It should be noted that linked ICA is characterized by sign indeterminacy, meaning that the signs (positive or negative) of the component loadings and corresponding components are ambiguous (*Comon and Jutten, 2010*). To understand the direction of the correlations, we inferred the direction of the signs by investigating the relationship between global gray matter volume and age since this is a well-known and relatively strong relationship in an adult sample with a large age span (*Good et al., 2001*), like our sample. For this purpose, we used IC1, which is driven by the VBM modality (contribution: 82.7%). This component covers the whole brain and reflects global gray matter volume (*Appendix 1—figure 4*). Our correlational results (*Supplementary file 2a*) show that this component is negatively correlated with age ($r_s$ = –0.50, p<0.001). In line with extensive evidence for a decrease in global gray matter volume related to aging (*Good et al., 2001*), we can infer from this that younger age should be related to a higher positive z-stat score on this component, and that the positive signs on the components indeed reflect a positive signal.

## Acknowledgements

This research did not receive any specific grant from funding agencies in the public, commercial, or not-for-profit sectors.

## Additional information

### Funding
No external funding was received for this work.

### Author contributions
Jasper van Oort, Conceptualization, Data curation, Formal analysis, Validation, Investigation, Visualization, Methodology, Writing – original draft, Project administration, Writing – review and editing; Alberto Llera, Conceptualization, Formal analysis, Supervision, Validation, Investigation, Visualization, Methodology, Writing – review and editing; Nils Kohn, Conceptualization, Formal analysis, Supervision, Validation, Visualization, Methodology, Writing – review and editing; Ting Mei, Conceptualization, Data curation, Software, Supervision, Investigation, Visualization, Methodology, Writing – review and editing; Rose M Collard, Conceptualization, Supervision, Investigation, Project administration, Writing – review and editing; Fleur A Duyser, Conceptualization, Data curation, Investigation, Project administration, Writing – review and editing; Janna N Vrijsen, Conceptualization, Investigation, Project administration, Writing – review and editing; Christian F Beckmann, Aart H Schene, Guillén Fernández, Conceptualization, Supervision, Investigation, Methodology, Writing – review and editing; Indira Tendolkar, Conceptualization, Supervision, Investigation, Methodology, Writing – original draft, Writing – review and editing; Philip FP van Eijndhoven, Conceptualization, Formal analysis, Supervision, Investigation, Methodology, Writing – original draft, Writing – review and editing

### Author ORCIDs
Jasper van Oort ⓘ http://orcid.org/0000-0003-2281-0349

### Ethics
The MIND-Set study has been approved by the Ethical Review Board of the Radboud UMC and all participants signed informed consent before participation.

### Decision letter and Author response
Decision letter https://doi.org/10.7554/eLife.85006.sa1
Author response https://doi.org/10.7554/eLife.85006.sa2

## Additional files

### Supplementary files
• Supplementary file 1. Correlations between biobehavioral measures of interest.

• Supplementary file 2. Correlations between independent components from linked ICA decompositions and biobehavioral measures of interest.

• MDAR checklist

## Data availability

All data analyzed in this study is stored in a Data Sharing Collection (DSC) (https://data.donders.ru.nl/collections/di/dccn/DSC_3013061.01_147?5) on the institutional repository of the Donders Institute for Brain, Cognition and Behavior, and is available upon reasonable request (N.B. the data in this DSC is only available for scientific, non-profit research). When a request for data access is submitted through the website mentioned above, the data collection manager receives this request. For getting access to the data, a project proposal has to be submitted and the data use agreement (DUA) has to be signed. For European Union (EU)-countries (and countries that offer an adequate level of data protection according to the EU; see the list of non-EU countries with an adequate protection level) a standard DUA is in place (RU-RA-DUA-1.0). Based on the project proposal and signed DUA the collection manager will assess whether the data can be shared. For non-EU countries (that do not offer an adequate level of data protection according to the EU), the data manager will reach out to the data steward within the Donders Institute, who in turn will reach out to the legal department of the Radboud university, providing details on who requests access to the data (including affiliation, country, and intended purpose of data usage). Based on this information a DUA will be drafted on a case by case basis.Furthermore, relevant data generated by the analyses we performed are included in the manuscript and supporting files. The linked ICA decomposition was performed using the Linked ICA toolbox, which was made available earlier by *Llera et al., 2019* (https://github.com/allera/Llera_elife_2019_1/tree/master/matlab_flica_toolbox; *Llera, 2019*).

The following dataset was generated:

| Author(s) | Year | Dataset title | Dataset URL | Database and Identifier |
|---|---|---|---|---|
| van Oort J, van Eijndhoven P | 2023 | MIND-Set – van Oort and colleagues 2023 – Linked ICA project | https://doi.org/10.34973/t6m1-x414 | Data Sharing Collection, 10.34973/t6m1-x414 |

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

## Appendix 1

### Methods

#### Subjects

MIND-Set (Measuring Integrated Novel Dimensions in Neurodevelopmental and Stress-related Mental Disorders) is an observational cross-sectional study. Inclusion of patients took place at the moment of intake at the outpatient clinic of the psychiatry department of the Radboud University Medical Center (Radboudumc), Nijmegen, the Netherlands. The MIND-Set study has been approved by the Ethical Review Board of the Radboudumc (Nijmegen, the Netherlands), and all participants signed an informed consent form before participation (*van Eijndhoven et al., 2021*). Data for this study were collected from June 2016 to July 2020.

Patients were diagnosed and classified by a trained clinician according to the DSM using semi-structured interviews (see *van Eijndhoven et al., 2021* for an extensive description of the diagnostic process). Mood and anxiety disorders were diagnosed using the Structured Clinical interview for DSM-IV Axis I Disorders (SCID-I) (*First et al., 1996*), and addiction disorders with the MATE-Crimi (*Schippers et al., 2010*; *Schippers and Broekman, 2012*). ADHD and ASD were assessed with a two-step diagnostic procedure. First, screeners were used. We used the World Health Organization Adult ADHD Self-Report Scale (ASRS)-short version for ADHD screening (*Silverstein et al., 2018*; *Kessler et al., 2005*). The ASRS is an ADHD screenings questionnaire consisting of six items (cut-off >3) with good psychometric properties (*Kim et al., 2013*). We screened for ASD by assessing autistic traits using the AQ-50 (50 items, cut-off >25) (*Baron-Cohen et al., 2006*; *Baron-Cohen et al., 2001*). Next, semi-structured interviews were performed for these disorders in case of a positive score on these screening instruments or if there was a clinical suspicion on one of these disorders during the extensive 3-hr clinical evaluation at the psychiatry department. We assessed the presence of ADHD with the semi-structured Diagnostic Interview for ADHD in Adults version 2.0 (Dutch: Diagnostisch Interview voor ADHD bij volwassenen 2.0 [DIVA 2.0]) (*Kooij and Francken, 2010*; *Ramos-Quiroga et al., 2019*). For ASD we administered the Dutch Interview for ASD in Adults (Dutch: Nederlands Interview ten behoeve van Diagnostiek Autismespectrumstoornissen bij volwassenen [NIDA]) (*Vuijk, 2014*). Both the DIVA and NIDA were completed in the presence of a partner and/or family member of the patient so that we were able to retrospectively and collaterally ascertain information on a broad range of symptoms in childhood and adulthood.

In order to provide more insight into the distribution of the symptom dimensions across broad diagnostic groups, we divided our samples into the following four broad subgroups: mentally healthy control group, stress-related group, neurodevelopmental group, and comorbidity group. The neurodevelopmental disorders (i.e. ASD and ADHD) were grouped together since both are lifelong disorders that start in early childhood and have a relatively stable, trait-like course and shared heritability (*Franke et al., 2018*; *Rommelse et al., 2011*). The group of stress-related disorders consists of the mood, anxiety, and addiction disorders. The mood and anxiety disorders share a common underlying dimension (*Kotov et al., 2017*), with a maladaptive stress response as a central feature in these disorders (*de Kloet et al., 2005*; *Sharma et al., 2016*). The addiction disorders are for this purpose also included in the stress-related group, given the important role of stress in the onset, maintenance, and relapse in these disorders (*Koob, 2003*; *Koob and Schulkin, 2019*). The comorbidity group consists of patients with both a stress-related and a neurodevelopmental disorder.

### Measures of interest

For our study, we included measures of interest consisting of symptom questionnaires, demographics, and additional biobehavioral measures of interest. In case of missing values, these missings were imputed by the median (non-normal distribution) of the available scores.

### Symptom questionnaires

Here, we provide additional information on the questionnaire subscales that showed significant correlations with the independent components. The anxiety/arousal (IDS-SR) and physical concerns (ASI) subscales both measure symptoms within the negative valence domain. More specifically, the anxiety/arousal subscale measures anxiety/panic, somatic/physiological arousal, and somatic

complaints. The physical concerns subscale measures fear of anxiety-related physical sensations, resulting from the belief that these sensations may have harmful consequences.

The subscales of the BRIEF-A pertain to the cognitive systems domain and measure higher-order cognitive/executive functions. The self-monitoring subscale (BRIEF-A) assesses the ability to regulate and monitor the effects of one's behavior. The inhibition subscale (BRIEF-A) measures inhibitory control and impulsivity. Finally, the working memory subscale (BRIEF-A) assesses the capacity to hold information in mind in order to complete a task, encode information, or perform sequential steps to achieve goals. Working memory is essential for carrying out multistep activities, the completion of mental manipulations (like mental arithmetics), and for following complex instructions.

## fMRI data acquisition

All images were collected using a 3T Siemens Magnetom Prisma MRI scanner (Erlangen) with a 32-channel head coil. For the T1 scan, high-resolution structural images (1.0 × 1.0 × 1.0 mm) were acquired using a T1-weighted MP-RAGE sequence (TE/TR = 3.03/2300 ms, flip angle = 8°, FOV = 256 × 256 × 192 mm, GRAPPA acceleration factor 2). For the DTI scan, a multi-band 3 protocol was used (TE/TR = 70.2/2370 ms, flip angle = 90°, slice thickness = 2.0 mm, number of slices = 69, in-plane resolution = 2.0 mm$^2$, b-values = 0/1,000 s/mm$^2$, number of gradients = 85).

For all three functional scans, T2*-weighted EPI BOLD-fMRI images were acquired using a multi-band 6 protocol with an interleaved slice acquisition sequence (number of slices = 66, TR = 1000 ms, TE = 34 ms, flip angle = 60°, voxel size = 2.0 × 2.0 × 2.0 mm, slice gap = 0 mm, FOV = 210 mm).

## MRI preprocessing and feature extraction

### Structural scans

VBM was used to extract gray matter densities from the T1 scans. The CAT-12 toolbox was used to preprocess the T1 data ('Computational Analysis Toolbox-12'; http://dbm.neuro.uni-jena.de/cat/) *Nenadic et al., 2015* in statistical parametric mapping 12 (SPM12) (Wellcome Department of Imaging Neuroscience, London, UK) (*Ashburner and Friston, 2000*). All T1 images were affinely aligned, before gray matter volume estimation. Next, images were segmented, normalized, and bias-field-corrected (*Ashburner and Friston, 2000*; *Elam and van Essen, 2013*). This resulted in images containing gray matter segments, white matter segments, and cerebrospinal fluid (CSF). Subsequently, DARTEL (*Ashburner, 2007*) was used to normalize all images to a standard gray matter template provided by the CAT-12 toolbox. All gray matter volumes were smoothed with a 9.4 mm FWHM Gaussian smoothing kernel (sigma = 4 mm). Quality control was performed for all VBM images using the quality measures calculated by the CAT-12 toolbox and expert visual inspection. Finally, for computational reasons, we spatially downsampled the VBM images to 2 mm isotropic (*Groves et al., 2011*; *Groves et al., 2012*).

The DTI scans were used to extract measures of white matter integrity. The DTI data was preprocessed using the DTIFIT routine from FSL (*Ashburner, 2007*; *Jenkinson et al., 2012*) (https://fsl.fmrib.ox.ac.uk/fsl) to create the FA and MD images. Next, these FA and MD images were fed into the TBSS pipeline (*Smith et al., 2006*). The resulting images had a voxel size of 1 mm isotropic. Quality control was performed by expert visual inspection of the registration, brain extraction, and results of the preprocessing pipeline described above (*Ashburner, 2007*; *Jenkinson et al., 2012*).

### Functional scans

The functional scans were preprocessed using FSL 5.0.11 (FMRIB, Oxford, UK). These scans were preprocessed using the FMRI Expert Analysis Tool (FEAT), which is part of the FMRIB Software Library (FSL) (*Jenkinson et al., 2012*). To allow for T2* equilibration effects, the first five images of each resting-state scan were discarded. Furthermore, the preprocessing steps included brain extraction, motion correction, bias field correction, high-pass temporal filtering with a cut-off of 100 s, spatial smoothing with a 4 mm full width at half maximum (FWHM) Gaussian kernel, and registration of functional images to high-resolution T1 using boundary-based registration and nonlinear registration to standard space (MNI152) (see also *van Oort et al., 2020*).

We used dual regression on the preprocessed data to generate subject-wise statistical maps (*Beckmann et al., 2009*; *Nickerson et al., 2017*), representing the whole brain connectivity of each network of interest (i.e. the DMN, ECN, and FPN). We used the network templates from *Smith et al., 2009* to select our networks of interest without biasing the results toward one of our functional scans (we merged the left and right FPN into one FPN template). We selected these large-scale

networks because they play an important role across broad functional domains, across psychiatric disorders. The DMN plays an important role in self-referential processing (*Buckner et al., 2008*), the ECN in emotion and action-inhibition (*Smith et al., 2009*), and the FPN in higher-order cognitive functions and top-down control (*Buhle et al., 2014*; *Corbetta and Shulman, 2002*). Aberrations in these networks are associated with core symptom domains, like negative valence and the cognitive systems, that transcend traditional disorder categories (*Menon, 2011*).

We applied dual regression on all three functional scans (i.e. resting-state, stress, and stress-aftermath scan), resulting in nine spatial maps per subject (3 scans × 3 networks of interest). We performed the following steps for all three our functional scans separately: we applied all the unthresholded network maps from *Smith et al., 2009* (i.e. 19 network maps since we merged the left and right FPN) as spatial maps into dual regression. Dual regression uses these spatial maps as input to generate subject-wise time courses for the networks of interest by correlating the mean time course of the network with all the voxels of the brain. Regression of these time courses against the data resulted in spatial maps of the 19 networks for each individual subject (*Filippini et al., 2009*). Afterward, we selected the outcomes for our three networks of interest (i.e. DMN, ECN, and FPN). This resulted for each subject in one spatial map per network. So in total this resulted in nine spatial maps per subject (3 scans × 3 networks of interest).

## Supplemental analyses
### Age and sex across diagnostic subgroups
For our broad diagnostic subgroups (i.e. mentally healthy controls, stress-related, neurodevelopmental, and comorbidity group), we tested whether there were any differences between these groups related to age (ANOVA) and sex (chi-square test).

## Linked ICA for separate MRI scans/modalities
We also performed the linked ICA factorization and full Spearman correlations for the separate MRI scans/modalities. For each of the functional scans, we selected the spatial maps of our networks of interest (i.e. DMN, ECN, and FPN). We performed the analysis separately for each of the following MRI modalities/scans:

1. Structural modalities (VBM, FA, and MD)
2. Resting-state scan (spatial maps of DMN, ECN, and FPN)
3. Stress scan (spatial maps of DMN, ECN, and FPN)
4. Stress-aftermath scan (spatial maps of DMN, ECN, and FPN)

## Partial correlations
In addition to the full Spearman correlations in the main analysis, we also performed post hoc partial Spearman correlations to test to what extent the results from the main analysis are driven by age and sex (covariates). We performed these partial Spearman correlations only for the significant results from the main analysis and again addressed the multiple comparisons by applying FDR correction (p<0.05) (*Benjamini and Hochberg, 1995*).

## Supplemental results
### Study population
In total, 258 patients and 80 mentally healthy participants participated in the MRI part of the MIND-Set study. Of these participants, 33 patients and 10 mentally healthy participants were not eligible for our present study and were excluded for the following reasons: not all structural and functional scans were available (among others because of early drop out due to excessive anxiety during scanning) (8 patients, 1 healthy participant), deviations on the scans (4 patients, 1 healthy participant), scanner artifacts (2 patients), excessive motion (14 patients, 6 healthy participants), insufficient quality of the T1 or DTI scan (1 patient, 1 healthy participant), and other technical problems (4 patients, 1 healthy participant). This resulted in 225 patients and 70 mentally healthy participants being included in our study.

In order to get more insight into our sample, the sample was subdivided into the following four broad diagnostic subgroups: mentally healthy controls (n = 70; mean age (± standard deviation [SD]): 37.4 ± 16.0 years; % male: 47.1%), stress-related (n = 84; mean age [± SD]: 39.9 ± 13.8; %

male: 53.6%), neurodevelopmental (n = 55; mean age [± SD]: 33.3 ± 12.2; % male: 69.1%), and comorbidity group (n = 86; mean age [± SD]: 33.9 ± 10.7; % male: 59.3%). There were differences in age between the four subgroups ($F_{(3,291)}$ = 4.20, p=0.006), with higher age in the stress-related group compared to the neurodevelopmental (p=0.004) and comorbidity group (p=0.003). There was no significant difference between the subject subgroups with respect to sex ($\chi^2$(3) = 6.61, p=0.085).

## Correlations between the multimodal ASD component and AQ-50 subscales:

Post hoc correlations were performed between the multimodal ASD component (IC32) and the AQ-50 subscales for the patients with ASD and participants without ASD separately. In patients with ASD, the following results were found: social skill: $r_s$ = –0.237 ($p_{uncorrected}$ = 0.037), difficulty with change/attention switching: $r_s$ = –0.179 ($p_{uncorrected}$ = 0.117), communication: $r_s$ = –0.096 ($p_{uncorrected}$ = 0.404), imagination $r_s$ = –0.052 ($p_{uncorrected}$ = 0.652), and attention to detail: $r_s$ = –0.122 ($p_{uncorrected}$ = 0.287). For the participants without ASD, the results showed the following: social skill: $r_s$ = –0.017 ($p_{uncorrected}$ = 0.807), difficulty with change/attention switching: $r_s$ = –0.039 ($p_{uncorrected}$ = 0.563), communication: $r_s$ = 0.024 ($p_{uncorrected}$ = 0.721), imagination $r_s$ = 0.037 ($p_{uncorrected}$ = 0.587), and attention to detail: $r_s$ = –0.035 ($p_{uncorrected}$ = 0.605).

## Correlations between ECN-FPN components and negative valence symptoms

The *ECN-stress aftermath component* (IC7) was associated with the following symptom dimensions in the negative valence domain: anxiety/somatic ($r_s$ = –0.23, p=0.007) and physical concerns ($r_s$ = –0.20, p=0.037). Here, we provide the correlations of the other ECN-FPN components with these same symptom dimensions for comparison (NB: p-values are FDR-corrected values): IC8: anxiety/somatic: $r_s$ = –0.14, p=0.286, physical concerns: $r_s$ = –0.10, p=0.551; IC13: anxiety/somatic: $r_s$ = –0.14, p=0.257, physical concerns: $r_s$ = –0.10, p=0.548; IC30: anxiety/somatic: $r_s$ = –0.02, p=0.964, physical concerns: $r_s$ = –0.03, p=0.923. Next, we performed post hoc analyses to compare the strength of the correlations between the *ECN-stress aftermath component* (IC7) and the negative valence symptoms with the correlations between the other ECN-FPN components and these same symptoms (uncorrected for multiple comparisons). These analyses showed the following results related to the IDS anxiety/somatic: IC7 versus IC8: z = –1.12, $p_{uncorrected}$ = 0.263; IC7 versus IC13: z = –1.07, $p_{uncorrected}$ = 0.284; IC7 versus IC30: z = –2.60, $p_{uncorrected}$ = 0.009. For ASI physical concerns: IC7 versus IC8: z = –1.18, $p_{uncorrected}$ = 0.237; IC7 versus IC13: z = –1.17, $p_{uncorrected}$ = 0.242; IC7 versus IC30: z = –2.04, $p_{uncorrected}$ = 0.041. Since there is no significant difference between the correlations of the *ECN stress-aftermath component* and the ECN components during the resting-state scan (IC13) or stress scan (IC8), we cannot exclude a threshold effect. At the same time, it is important to note that the *ECN-stress aftermath component* was the most sensitive for finding these relationships at statistically significant levels. Thus, the stress induction may have played an important role in revealing these results (during the stress-aftermath period).

## Linked ICA for separate MRI scans/modalities

While we found 87 significant correlations in the main analysis, in which we analyzed all 12 MRI modalities together, analyzing the modalities/scans separately resulted in maximally 33 significant correlations per analysis. Analyzing the modalities separately confirms that functional scans are more sensitive in finding correlations with symptom dimensions since eight out of nine correlations with symptoms were with the functional scans (*Supplementary file 2b–e*).

## **Partial correlations**

The post hoc partial correlations showed that 37 of the correlations from the main analysis remained significant (FDR-corrected q < 0.027) when correcting for age and sex (*Supplementary file 2a*). Importantly, 9 out of 10 correlations for the symptoms remained significant, indicating that these were in general not driven by age and/or sex. Eight partial correlations remained significant for IC7, and these partial correlations spanned different units of analysis (e.g. symptoms, physiological/cardiovascular measures, BMI, functioning/general health) (*Figure 5A*).

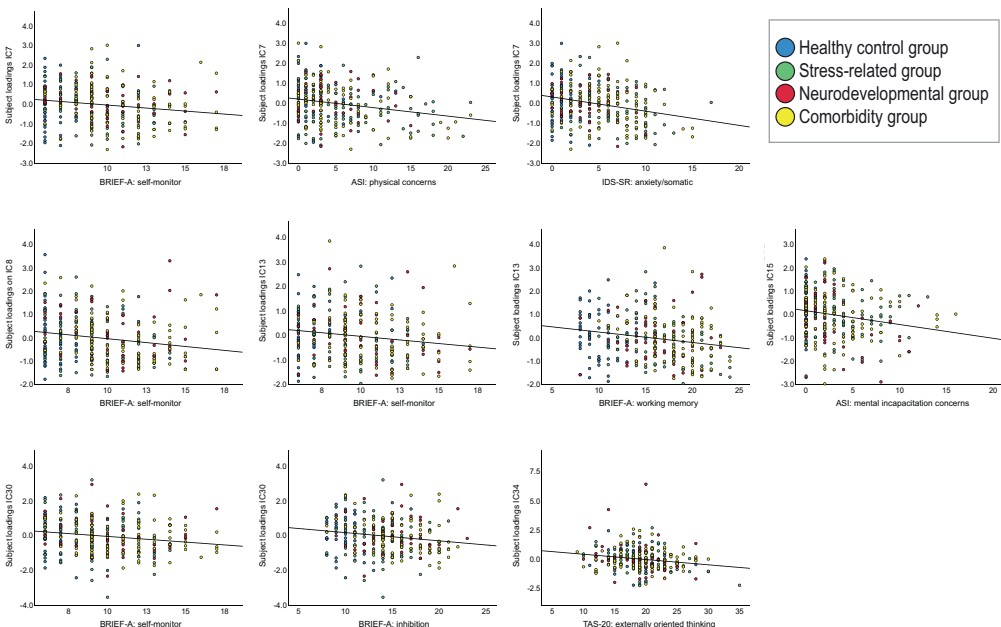

**Appendix 1—figure 1.** Scatterplots displaying the relationships between the symptom dimensions and independent components (ICs).

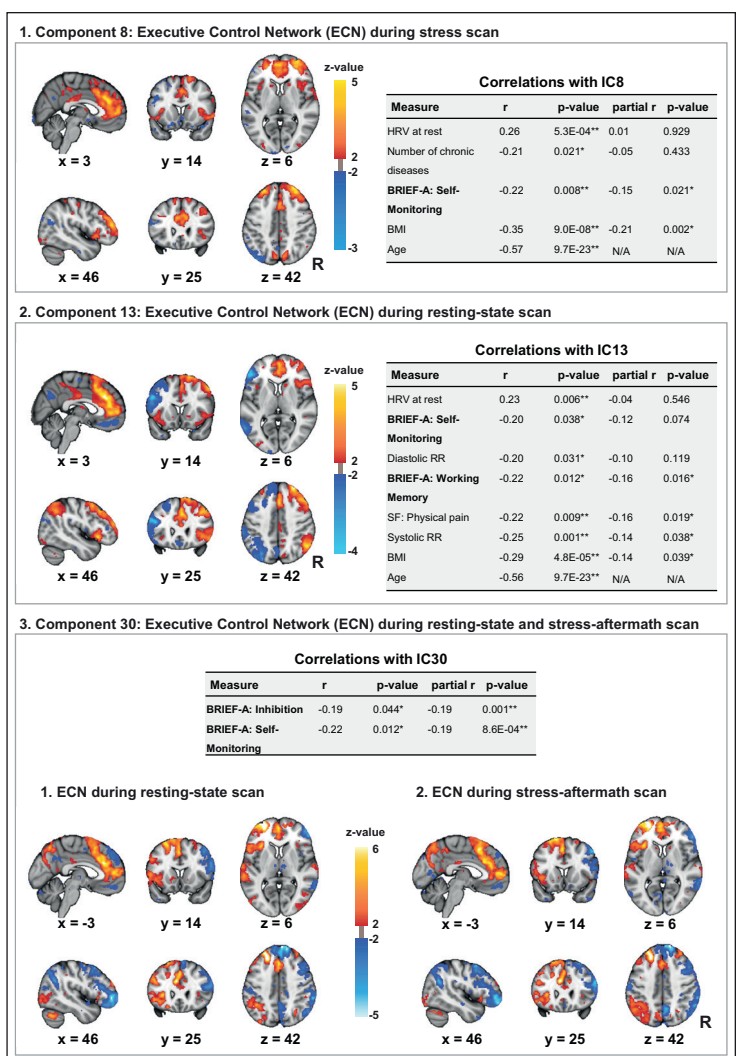

**Appendix 1—figure 2.** The components displayed here have important similarities with independent component 7 since all these components reflect connectivity of the executive control network (ECN) with itself and with the frontoparietal network (FPN). (1) Independent component 8 is driven by the ECN during the stress scan (99.8%). (2) Independent component 13 is driven by the ECN during the resting-state scan (99.9%). (3) Independent component 30 is mainly driven by the ECN during the resting-state scan (42.2%) and stress-aftermath scan (38.3%). For all components in this figure, we used the 10 and 90 percentiles for thresholding, for display purposes, since the underlying distribution was not z-distributed. In this figure, the right side of the brain is displayed on the right side of the image. R: right.

The online version of this article includes the following source data for appendix 1—figure 2:

**Appendix 1—figure 2—source data 1.** Correlations for IC8, IC13, and IC30.

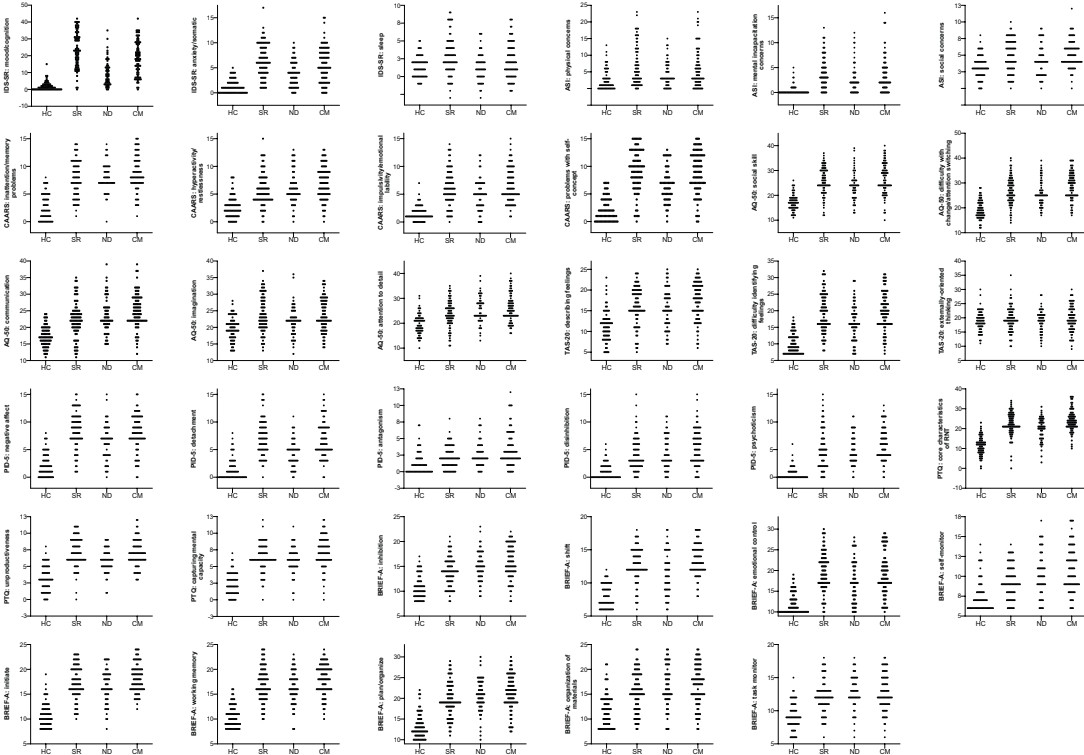

**Appendix 1—figure 3.** Dot plots for symptom dimensions across the subgroups. ASI: Anxiety Sensitivity Index; AQ-50: Autism spectrum Quotient-50; BRIEF-A: Behavior Rating Inventory Executive Function – Adult; CAARS: Conners' Adult ADHD Rating Scale; CM: comorbidity group; HC: mentally healthy controls; IDS-SR: Inventory of Depressive Symptomatology Self Report; ND: neurodevelopmental group; PID-5: Personality Inventory for DSM-5-Short Form; PTQ: Perseverative Thinking Questionnaire; RNT: repetitive negative thinking; SR: stress-related group; TAS-20: Toronto Alexithymia Scale-20.

### Component 1: voxel-based morphometry (VBM)

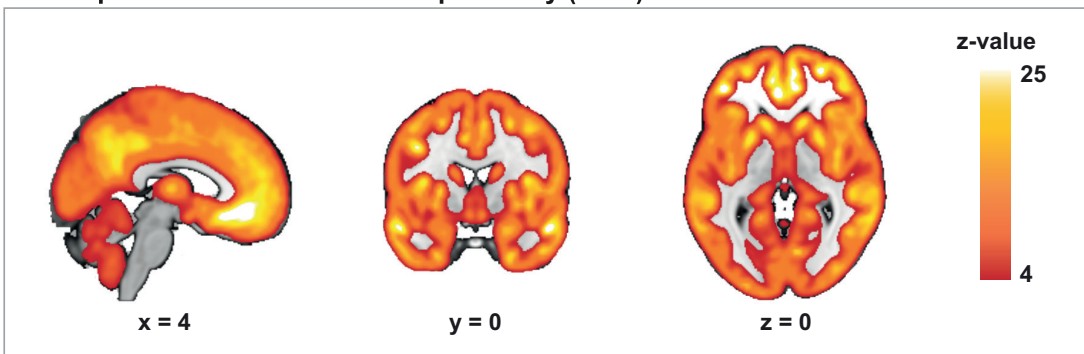

### 2. Correlation between component 1 and age

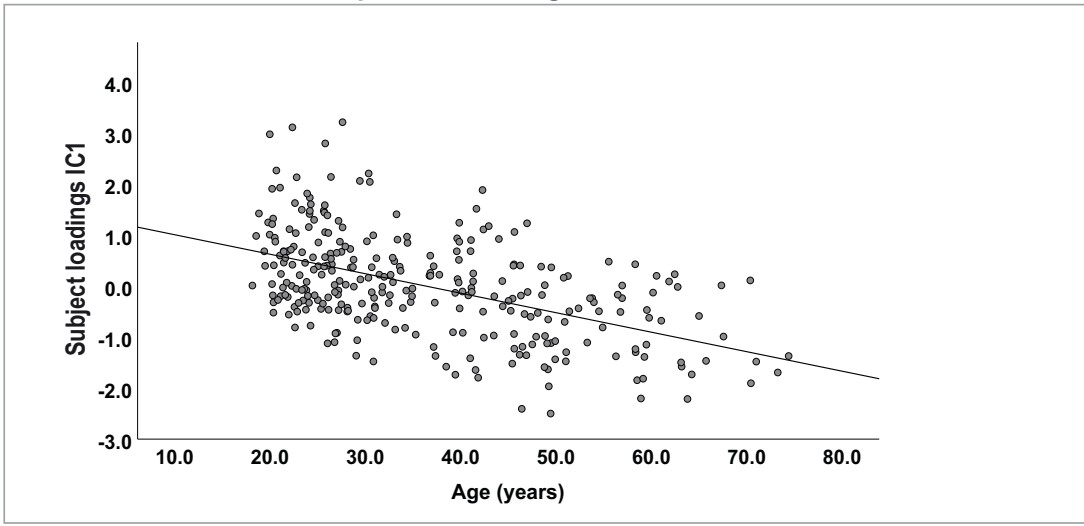

**Appendix 1—figure 4.** Test for direction of correlational results. (**1**) Independent component 1 (IC) is driven by the voxel-based morphometry (VBM) modality (82.7%) and reflects global gray matter volume.(**2**) The subject loadings on IC1 were negatively correlated with age ($r_s$ = –0.50, p=9.7E-18).

**Appendix 1—table 1.** Symptom measures.

| Topic | Questionnaire | Abbreviation questionnaire | Questionnaire subscales (n = 35) | Reference for questionnaire subscales |
|---|---|---|---|---|
| Depressive symptoms | Inventory of Depressive Symptomatology Self Report | IDS-SR | 1.Mood/cognition 2.Anxiety/somatic 3.Sleep | *Wardenaar et al., 2010* |
| Anxiety sensitivity | Anxiety Sensitivity Index | ASI | 1.Physical concerns 2.Mental incapacitation concerns 3.Social concerns | *Rodriguez et al., 2004* |
| Attention-deficit hyperactivity disorder symptoms | Conners' Adult ADHD Rating Scale | CAARS | 1.Inattention/memory problems 2.Hyperactivity/ restlessness 3.Impulsivity/emotional lability 4.Problems with self-concept | *Conners et al., 1999* |
| Autistic traits | Autism spectrum Quotient-50 | AQ-50 | 1.Social skill 2.Difficulty with change/ attention switching 3.Communication 4.Imagination 5.Attention to detail | *Hoekstra et al., 2008* |
| Alexithymia | Toronto Alexithymia Scale-20 | TAS-20 | 1.Describing feelings 2.Difficulty identifying feelings 3.Externally oriented thinking | *Bagby et al., 1994* |
| Personality traits | Personality Inventory for DSM-5-Short Form | PID-5-B-Adult | 1.Negative affect 2.Detachment 3.Antagonism 4.Disinhibition 5.Psychoticism | *Krueger et al., 2012* |
| Repetitive negative thinking (RNT) | Perseverative Thinking Questionnaire | PTQ | 1.Core characteristics of RNT 2.Unproductiveness 3.Capturing mental capacity | *Ehring et al., 2011* |
| Behavioral regulation | Behavior Rating Inventory Executive Function – Adult | BRIEF-A | 1.Inhibition 2.Shift 3.Emotional control 4.Self-monitor 5.Initiate 6.Working memory 7.Plan/organize 8.Organization of materials 9.Task monitor | *Hocking et al., 2015* |

**Appendix 1—table 2.** Demographics and biobehavioral measures of interest*.

| Topic | Measures for correlational analysis (n = 45) | Assessment | Description of measure | Reference for background information |
|---|---|---|---|---|
| Demographics | Age | Demographics standard questionnaire | Age in years | *Stronks et al., 2013* |
| | Sex | | Male/female | |
| | Level of education | | 4 levels: no, low, middle, high | *Ikram et al., 2015*, *Stronks et al., 2013* |
| Anthropometric measure | BMI | Weight scale and stadiometer | BMI = weight/(length^2) (weight in kilogram, length in meter) | *van Eijndhoven et al., 2021* |
| Biological/ physiological measures | Systolic blood pressure | Blood pressure band | Systolic blood pressure (mmHg) | *van Eijndhoven et al., 2021* |
| | Diastolic blood pressure | | Diastolic blood pressure (mmHg) | |
| | Heart rate during resting-state scan | Infrared pulse oximeter in MRI scanner | Heart rate in beats per minute (BPM) during the resting-state scan Calculated using in-house software | *van Oort et al., 2020*, see also *Figure 1* for moment of measurements |
| | Stress-induced change in heart rate (stress – neutral) | | Stress-induced change in heart rate (BPM): during stress movie minus during neutral movie | |
| | Heart rate variability (HRV) during resting-state scan | | HRV is calculated using the a trimmed version (trimming lowest and highest 10% of values) of the root mean square of successive differences (rMSSD) | *Shaffer and Ginsberg, 2017* |
| | Stress-induced change in HRV | | Stress-induced change in trimmed rMSSD score (see above): during stress movie minus during neutral movie | |
| | Baseline cortisol | Salivette for saliva cortisol | Saliva cortisol level during acclimatization period (20 min before scanning) | *Kirschbaum and Hellhammer, 1994*, *van Oort et al., 2020* |
| | Cortisol after stress induction | | Saliva cortisol level ±25 min after the start of the stress induction | *Kirschbaum and Hellhammer, 1994*, *van Oort et al., 2020* |
| | Hair cortisol | Hair sample from scalp | Hair sample from scalp (>3 cm length) | *Staufenbiel et al., 2015* |
| Somatic disorders | Number of chronic somatic disorders | Statistics Netherlands questionnaire (CBS) | Number of chronic disorders including hypertension, for which a participant is under treatment from a doctor and/or for which the participant uses medication | *Bekhuis et al., 2016* |
| Subjective stress | Subjective stress at baseline in scanner | In-house questionnaire | Subjective stress rating on an eleven-point rating scale (0–10) directly after the resting-state scan | See Figure 1 of *van Oort et al., 2020* |
| | Stress-induced change in subjective stress | | Subjective stress rating (see above): stress minus control condition | |
| Trauma history | 1.Emotional neglect 2.Psychological abuse 3.Physical abuse 4.Sexual abuse | NEMESIS-childhood trauma questionnaire | NEMESIS-trauma questionnaire: 4 subscales, one score (0–2) for each domain described in column 2 | *Hovens et al., 2010* |

*Appendix 1—table 2 Continued on next page*

*Appendix 1—table 2 Continued*

| Topic | Measures for correlational analysis (n = 45) | Assessment | Description of measure | Reference for background information |
|---|---|---|---|---|
| Psychiatric classification | Current mood disorder (yes/no) | SCID-I | The Structured Clinical interview for DSM-IV Axis I Disorders (SCID-I): current depression and/or dysthymia (yes/no) | *First et al., 1996* |
| | Current anxiety disorder (yes/no) | SCID-I | Current DSM-IV anxiety disorder according to SCID-I: panic disorder, agoraphobia, social phobia, specific phobia, obsessive compulsive disorder, posttraumatic stress disorder, generalized anxiety disorder and/or anxiety disorder not otherwise specified (yes/no) | |
| | ASD (yes/no) | NIDA | Dutch Interview for ASD in Adults (NIDA): autism spectrum disorder (ASD) (yes/no) | *Vuijk, 2014* |
| | ADHD (yes/no) | DIVA 2.0 | Diagnostic Interview for ADHD in Adults version 2.0: ADHD (yes/no) | *Kooij and Francken, 2010*, *Ramos-Quiroga et al., 2019* |
| | Addiction disorder (yes/no) | MATE-Crimi | Measurements in the Addictions for Triage and Evaluation and Criminality (MATE-Crimi): addiction disorder (yes/no) | *Schippers et al., 2010*; *Schippers and Broekman, 2012* |
| | Psychiatrically healthy (yes/no) | See diagnostic instruments stated above | Psychiatrically healthy control subject or psychiatric patient (with one or more disorders described above) | See also *van Eijndhoven et al., 2021* for extensive description of the diagnostic, classification process |
| Substance use | Level of smoking | MATE-Crimi | Not smoking, light smoker, heavy smoker | *van Eijndhoven et al., 2021* |
| | Cannabis use (yes/no) | In-house questionnaire | Used cannabis last 7 d before scanning (yes/no) | N/A |
| | Alcohol consumption | In-house questionnaire | Number of standard units of alcohol used in the 7 d before scanning | N/A |
| Medication | Antipsychotic (yes/no) | Medication verification: anamnesis and medication list from pharmacy Medication grouping based on ATC code | Antipsychotics (N05A) (only lithium (N05AN) is excluded from this category) (yes/no) | https://www.whocc.no/atc/structure_and_principles/ (last checked date June 7, 2021) |
| | Anxiolytic, hyponotic and/or sedative (yes/no) | | Anxiolytics (N05B), hypnotics and sedatives (N05C) and/or promethazine (R06AD02) (yes/no) | |
| | Antidepressant (yes/no) | | Antidepressants (N06A) (yes/no) | |
| | Central-acting sympathicomimetic (yes/no) | | Central-acting sympathicomimetics (N06BA) (yes/no) | |
| Functional limitations | 1. Cognition 2. Mobiity 3. Self-care 4. Getting along 5. Life activities 6. Participation | WHODAS | WHO-Disability Assessment Schedule 2.0 (WHODAS 2.0): 6 existing subscales of this questionnaire, covering different domains of functioning (see column 2). | *Chwastiak and Von Korff, 2003* |
| General health | 1. Physical functioning 2. Role fulfillment 3. Social functioning 4. Mental health 5. Experienced health 6. Physical pain | SF-20 | Short Form-20 (SF-20): 6 existing subscales of this questionnaire, covering different domains of health (see column 2) | *Stewart et al., 1989* |

ADHD: attention-deficit hyperactivity disorder; BMI: body mass index; N/A: not applicable.

*See also *van Eijndhoven et al., 2021* for extensive description of the measures in this table and for the diagnostic, classification process.

**Appendix 1—table 3.** Use of psychotropic medication at the time of the MRI scan.

| | Healthy controls (n = 70) | Combined patient group* (n = 225) | Stress-related group (n = 84) | Neurodevelopmental group (n = 55) | Comorbidity group (n = 86) |
|---|---|---|---|---|---|
| Current medication use† | | | | | |
| Antidepressant (n = ..) | 0 | 84 | 41 | 9 | 34 |
| Antipsychotic (n = ..) | 0 | 28 | 16 | 2 | 10 |
| Central-acting sympathicomimetic‡ (n = ..) | 0 | 21 | 2 | 6 | 13 |
| Anxiolytic, hyponotic and/or sedative (yes/no) (daily use) (n = ..) | 0 | 27 | 12 | 2 | 13 |
| Mood stabilizer (n = ..) | 0 | 3 | 3 | 0 | 0 |

*The combined patient group consists of all patients that were included in this study, and can be subdivided in the stress-related, neurodevelopmental and comorbidity group.

†Number of participants using the type of medication stated below. Grouping of medication is based on ATC code (see *Appendix 1—table 2* for an extensive description of the medication groups). The mood stabilizer group is added to this table, but was not used in the correlational analysis, given the low number of patients using this type of medication (i.e. valproic acid [n=1] and lithium [n=2]).

‡The central-acting sympathomimetic drugs represent the use of psychostimulants.

