## [Editor Report]

This study presents a valuable method for performing an integrative structural-functional linked ICA analysis, investigating the relationship between the brain and a large set of symptoms and other biobehavioral measures transdiagnostically. The results show relations between multi-modal and unimodal independent components with the presence of autism spectrum disorder and variations in cognitive functioning and negative affect across individuals suffering from mood, anxiety, substance dependence, autism spectrum, or ADHD. Overall, the results are compelling and interesting to a wide readership.

---

## [Decision Letter]

**Decision letter after peer review:**

Thank you for submitting your article "Brain structure and function link to variation in biobehavioral dimensions across the psychopathological continuum: A multimodal linked independent component analysis" for consideration by *eLife*. Your article has been reviewed by 2 peer reviewers, and the evaluation has been overseen by a Reviewing Editor and Christian Büchel as the Senior Editor. The following individuals involved in the review of your submission have agreed to reveal their identity: Ryuichiro Hashimoto (Reviewer #1); Marie-José van Tol (Reviewer #2).

Essential revisions:

1. I think that some important ideas behind the design of this study are only briefly mentioned in the main text, just referring to their original paper on the MIND dataset. Readers may understand the significance better if they are given an explanation about why the different kinds of disorders (ASD, ADHD, mood disorder, anxiety, addiction) can be gathered up (but not other types of disorders) and why they were particularly interested in stress-related responses for the group.

2. I found Figure 1 of the overlap of the disorders interesting, and it may be interesting to examine whether the number of overlaps of the disorders or particular combinations of disorders is particularly related to any neural ICs or physiological/cognitive features.

3. Perhaps also include Morris et al. 2022 (https://doi.org/10.1186/s12916-022-02414-0) in addition to Cuthbert & Insel, 2013, to direct to a revisited version of the RDOC framework.

4. Here I get a bit lost: "The correlational analysis (Figure 2, operation D) resulted in 87 significant correlations (FDR corrected q <0.001) between the components and measures of interest (all p-values mentioned below are FDR corrected values (see Supplemental Table C for all significant correlations)). The analysis resulted in ten correlations between the ICs and symptom dimensions. Interestingly, all symptom correlations were discovered for components that were mainly driven by functional scans. Of the nineteen correlations that were found with multimodal components, most were related to age, sex, BMI, blood pressure, and heart rate variability, while there was also an association with cognitive symptoms and a classification of ASD."

Maybe include: 'of those 87, ten related to symptom dimensions, 19 with etc". The confusion comes from mixing the findings on uni-modal vs multi-modal and findings related to type of biobehavioral measure.

5. I don't understand this sentence "The DMN spatial map displayed within DMN connectivity (angular gyrus, precuneus, supramarginal gyrus)." Line 157/158. This sentence also seems incomplete "Additionally, the VBM feature weighted in the precentral gyrus and the MD feature in the thalamus (Figure 4B). (lines 162/163).

6. Anxiety (symptoms and sensitivity) and sleep problems seem to be pretty low, as are mood symptoms. Are distributions given somewhere else (outside Table 1)?

7. Related to participants, given this information is in the supplement, could it be added when and how patients were recruited? Related to their in/out-patient treatment? Was remission allowed? How was the distribution of past/current diagnoses in each group? What was the correlation between all symptomatic and biobehavioural markers? Could a correlational table/graph help here? Also the relation with medication helps the reader to put everything into context.

*Reviewer #1 (Recommendations for the authors):*

The authors applied a novel integrative analysis called linked independent component analysis (ICA) by which they successfully identified the combinations of abnormal structural and functional brain features that were associated with cross-diagnostic symptom dimensions in an objective, data-driven manner. The combination of the novel multimodal analytical technique and a large-scale, extensively phenotyped transdiagnostic dataset is a good example of the research direction endorsed in the Research Domain Criteria (RDoC). Although the linked ICA is suitable for identifying the co-varying patterns of abnormalities in a multimodal neuroimaging dataset in a data-driven manner, it would be fair to explain some drawbacks. In particular, the problem of the sign indeterminacy of ICs may be critical for interpreting the identified patterns. This is particularly relevant when the authors stated that some ICs showed "negative" correlations with cognitive and affective measures (e.g. p.6-7). It is also possible that the components with reversed signs are positively correlated with the measures. Another concern is that the number of ICs was prespecified as 50 – how was it determined?

The study seems motivated by the transdiagnostic and dimensional approach, but the results included a multimodal component (IC32) categorically associated with the diagnosis of ASD. It would be interesting, from the perspective of the dimensional approach, to examine in an ad-hoc manner whether this component was correlated with continuous variables for the ASD trait as measured by the AQ among people who do not have the ASD diagnosis.

The length of the stress scan: 150 sec may be a little short to identify individual FC network components.

On page 6, it is stated that there were 10 correlations between symptoms and ICs. What were the components other than ICs 7, 8, 13, and 30?

On page 12, it is stated that correlation analysis for symptom dimensions was done using the summed score of 35 subscales of 8 questionnaires. I am not sure whether these scores can be summed with equal weight to form a valid score for the symptom of interest.

On page 11 and in Figure 2 legend, stress induction was done by viewing an aversive movie clip. Is there any supporting evidence that this procedure was successful in inducing stress as expected?

The age range of 18 – 74 is quite large and I wonder whether there is any systematic difference among patients, for instance, people who have the diagnosis of ASD or ADHD and those who do not. Relatedly, on page 11, how was the diagnosis of adult ASD and ADHD made? Was it all based on the DSM without any other supporting evidence such as ADOS or ADI-R?

I think that some important ideas behind the design of this study are only briefly mentioned in the main text, just referring to their original paper on the MIND dataset. Readers may understand the significance better if they are given an explanation about why the different kinds of disorders (ASD, ADHD, mood disorder, anxiety, addiction) can be gathered up (but not other types of disorders) and why they were particularly interested in stress-related responses for the group.

I found Figure 1 of the overlap of the disorders interesting, and it may be interesting to examine whether the number of overlaps of the disorders or particular combinations of disorders is particularly related to any neural ICs or physiological/cognitive features.

*Reviewer #2 (Recommendations for the authors):*

In this work, van Oort et al. analyzed multi-modal brain data and characterization of symptomatology and other biobehavioral markers in 225 people diagnosed (according to DSM-IV) with unimodal depression, anxiety disorders (including PTSD, substance use disorders, autism spectrum disorder, or attention-deficit hyperactivity disorder) and 70 people without such diagnosis and not seeking help for their mental health problems.

The work is inspired by the RDOC framework aimed at understanding mechanisms underpinning specific symptom domains occurring in multiple psychiatric disorders. This approach allows for more dimensional characterization across the spectrum from 'health' to 'disordered' on multiple domains of functioning.

All participants underwent a series of scans, including structural scanning for grey matter morphometry and white matter characterization based on DTI. Also, before, during, and after stress induction, functional acquisitions to study correlational patterns over time were obtained. From all this data, linked ICA was used to obtain independent components, of which some were based on one modality and some included multiple scanning modalities. Next Spearman correlations of these independent components with the symptom dimensions were calculated, in which a dimensional take towards depressive symptomatology was included (not looking at total depression, but at dimensions of mood and cognition, anxiety arousal, and sleep, following the tripartite model of depression).

This study must have been an enormous amount of work and I applaud the authors for making the report so readable and easy to digest as if complexity and acquisition and analysis effort were not an issue. The supplements are informative.

I have some questions that relate to the study design and assumptions underpinning the choice of methods (which diagnosis to include, and which resting state 'challenge' included). Especially given the inclusion of ASD and ADHD, not primarily stress disorders, why was the stress challenge chosen? Also, it might help to include a short section explaining the unique and common symptom dimensions of the included psychopathologies and what we might uniquely learn from studying them jointly. This will help clarify the aim of the study.

For interpretation of the data, it would help to see an overview of group differences (for each diagnostic class [i.e.] hc, mood, anxiety, asd, substance use, adhd) for the biobehavioural measures (in the supplement). I understand group differences are not the aim here, but for interpreting the results further, it helps to see if the scores are basically a continuous reflection of diagnostic labels (yes/no), is not so specific to certain labels. For the same reason, a correlation table of how the symptomatic (and other markers) correlated with each other would help.

Correlational plots of the brain-behaviour relations showing the 'diagnostic' label (color-coded dots) would help to see the transdiagnostic relevance.

A discussion on whether the inclusion of people with a (current?) diagnosis of psychiatric disorders and also including asymptomatic controls is the optimal approach for this RDOC-style analysis would strengthen the discussion. Has it been considered to leave people without a lifetime diagnosis out?

I was a bit surprised to see the first analyses cover the loading of ASD diagnosis present vs absent and not the correlation with a continuous autism spectrum measure. It's worth highlighting and discussing that the ASD (yes/no) classification outperformed continuous ASD measures. This is important for the worth of diagnostic labels vs. continuous characterizations. Also, can the authors discuss whether the sampling of 'patients' vs 'healthy controls' plays a role here and might explain the strength of the effect of a diagnostic label?

Also for the ASD results, does the correlation between stress and non-stress ICNs matter? What does it mean for example that the DMN in the stress-aftermath contributed to ASD (yes/no) and the pre-stress DMN did not? Can one say something about the relevance of the stress-reactivity, or just that it's more sensitive to pick up interindividual variation?

In the discussion, I miss a discussion on the importance of including a stress challenge in their study. Is it worth the trouble? It seems that the correlation between symptoms (cognitive) did not really differ between pre-, during, and after stress. Was a symptom x IC interaction calculated? The fact that a correlation between negative valence with only IC7 was observed, does not mean that this correlation was specific to IC7, it may have occurred just subthreshold for IC8, IC13, and IC30. I believe this information and exploration are needed to conclude that "Together, this indicates the importance of this ECN-stress aftermath component for a broad spectrum of health-related measures".

Related to the allostatic load reasoning, I think the possibility that the physiological/cardiac, metabolic, and general health problems contributed to the development of psychiatric problems can't be excluded. Also, because no measure of the duration of psychiatric problems was included in the presented results, whether this represents a 'load' effect is unclear. I think no causality and indication of allostatic load can be suggested here.

How were medication use and substance use controlled in the analysis? Probably adding these latter as covariates is not optimal, given the high correlation between medication/substance use and various symptom (sub)scales and diagnostic classes. Some discussion on this influence is needed.

Given the rationale and aims outlined in the introduction, the discussion could benefit from some more in-depth discussion of what the results mean for our understanding of the psychiatric disorder, how did they change it, what's new, and what next steps should we entail and what added value the current approach has (worth of multi-modal vs unimodal components for example).

First of all, congratulations on this impressive work and clearly written report. It's obvious to me that this work involves an enormous amount of work and complex analyses. Below you find some comments that relate to readability, that were not covered in the public review. I hope my comments further contribute to making this an impactful work. I'm still searching a bit in what tone and form comments should be given since they are published alongside the manuscript. But overall, this was a very interesting experience.

I appreciate that you included Prof. Schene as an author.

Perhaps also include Morris et al. 2022 (https://doi.org/10.1186/s12916-022-02414-0) in addition to Cuthbert & Insel, 2013, to direct to a revisited version of the RDOC framework.

Here I get a bit lost: "The correlational analysis (Figure 2, operation D) resulted in 87 significant correlations (FDR corrected q <0.001) between the components and measures of interest (all p-values mentioned below are FDR corrected values (see Supplemental Table C for all significant correlations)). The analysis resulted in ten correlations between the ICs and symptom dimensions. Interestingly, all symptom correlations were discovered for components that were mainly driven by functional scans. Of the nineteen correlations that were found with multimodal components, most were related to age, sex, BMI, blood pressure, and heart rate variability, while there was also an association with cognitive symptoms and a classification of ASD."

Maybe include: 'of those 87, ten related to symptom dimensions, 19 with etc". The confusion comes from mixing the findings on uni-modal vs multi-modal and findings related to the type of biobehavioral measure.

I don't understand this sentence "The DMN spatial map displayed within DMN connectivity (angular gyrus, precuneus, supramarginal gyrus)." Line 157/158. This sentence also seems incomplete "Additionally, the VBM feature weighted in the precentral gyrus and the MD feature in the thalamus (Figure 4B). (lines 162/163).

Anxiety (symptoms and sensitivity) and sleep problems seem to be pretty low, as are mood symptoms. Are distributions given somewhere else (outside Table 1)?

Nice Figure 1.

Related to participants, given this information is in the supplement, could it be added when and how patients were recruited? Related to their in/out-patient treatment? Was remission allowed? How was the distribution of past/current diagnoses in each group? What was the correlation between all symptomatic and biobehavioural markers? Could a correlational table/graph help here? Also the relation with medication helps the reader to put everything into context.

---

## [Author Response]

Essential revisions:1. I think that some important ideas behind the design of this study are only briefly mentioned in the main text, just referring to their original paper on the MIND dataset. Readers may understand the significance better if they are given an explanation about why the different kinds of disorders (ASD, ADHD, mood disorder, anxiety, addiction) can be gathered up (but not other types of disorders) and why they were particularly interested in stress-related responses for the group.

We thank the reviewer for raising this important point. We indeed think that these disorders can be gathered up in order to study transdiagnostic mechanisms across these diverse non-psychotic psychiatric disorders (i.e. ASD, ADHD, mood disorder, anxiety, addiction). It is important to study these highly prevalent disorders together in a transdiagnostic manner, as high levels of comorbidity among these disorders suggest shared underlying mechanisms (van Eijndhoven et al., 2021). In addition, numerous studies (often investigating these disorders separately) provide converging evidence that symptoms across various major domains (like negative valence and cognitive systems domain) cut across the diagnostic boundaries of these disorders (Kerns et al., 2015; Koob and Schulkin, 2019; McTeague et al., 2016; Mulders et al., 2022; J Anthony Richey et al., 2015; Shaw et al., 2014a; van Eijndhoven et al., 2021; Woody and Gibb, 2015). Importantly, disturbances in the same brain networks may underlie these symptom dimensions across these diverse disorders (McTeague et al., 2016; Menon, 2011). Therefore, it is crucial to adopt a transdiagnostic approach in order to identify the brain basis for symptom dimensions that transcend these diagnostic categories. Finally, we do want to emphasize here that we do not want to make any claims that the possibility to study transdiagnostic brain mechanisms is limited to the groups included in this study. Future studies should investigate if our findings can be replicated in other groups of disorders or in a transdiagnostic sample with an even a broader range of psychiatric disorders.

With respect to our interest in the stress-related responses: clinically it is well established that there is increased stress sensitivity across a broad range of psychiatric disorders (Ingram and Luxton, 2005). For depression and anxiety disorders it is well known that this is a central feature of these disorders (de Kloet et al., 2005; Sharma et al., 2016). However, also for the other disorders included in this study, there are clear indications for increased stress sensitivity. In addiction disorders, stress plays an important role in the onset, maintenance and recurrence (relapse) in these disorders (Koob, 2003; Koob and Schulkin, 2019). And also in neurodevelopmental disorders there are clear indications for increased stress sensitivity, such as arousal and emotion regulation problems (Kerns et al., 2015; J.A. Richey et al., 2015; Shaw et al., 2014b). We included the stress scans in our experimental set up, as we hypothesized that shared mechanisms of stress vulnerability would become visible under conditions of stress.

Based on this comment we have adjusted the introduction in order to further explain our rationale for the inclusion of these disorders and for the inclusion of stress scans in the experimental set up. We think that these adjustments have significantly improved the introduction.

We have changed the text in the introduction to the following:

“Therefore, in the current study, we used linked ICA within a transdiagnostic, dimensional framework, using the MIND-Set database (Measuring Integrated Novel Dimensions in Neurodevelopmental and Stress-related Mental Disorders) (van Eijndhoven et al., 2021). This database provides us with a sample of mentally healthy participants and patients with diverse, highly prevalent non-psychotic psychiatric disorders (i.e. mood disorders, anxiety disorders, addiction, ASD, ADHD, and their comorbidity). A multimodal imaging battery was performed and all participants were deeply phenotyped. An extensive set of biobehavioral measures was collected, including symptom dimensions, biological/physiological measures (like cortisol and heart rate variability) and also more general measures of physical and mental health. Since psychopathology can best be understood on a continuum from health to mental illness, we also included mentally healthy participants, which allows us to study the brain and biobehavioral dimensions of interest along a wider range from health to psychopathology (Kotov et al., 2017; Morris et al., 2022; van Oort et al., 2022).

We investigated the diverse non-psychotic psychiatric disorders (i.e. mood disorders, anxiety disorders, addiction, ASD, ADHD) together, as high levels of comorbidity suggest shared underlying mechanisms (van Eijndhoven et al., 2021). In addition, numerous studies that investigated these disorders separately provide converging evidence that symptoms across various major domains cut across the diagnostic boundaries of these disorders, with among others transdiagnostic symptom dimensions related to the negative valence domain (e.g. repetitive negative thinking), and cognitive systems domain (e.g. regarding cognitive control problems) (Kerns et al., 2015; Koob and Schulkin, 2019; McTeague et al., 2016; J Anthony Richey et al., 2015; Shaw et al., 2014a; van Eijndhoven et al., 2021; Woody and Gibb, 2015). Interestingly, disturbances in the same brain networks may underlie these core symptom dimensions across these diverse disorders (McTeague et al., 2016; Menon, 2011). Therefore, it is crucial to adopt a transdiagnostic approach in order to identify the brain basis for symptom dimensions that transcend these diagnostic categories.

In our multimodal neuroimaging battery, we included functional scans under conditions of rest as well as under conditions of mild experimentally induced stress. It is clinically well established that vulnerability to stress is a common feature across a broad range of psychiatric disorders (Ingram and Luxton, 2005). Mood and anxiety disorders are characterized by a maladaptive stress response as their central feature (de Kloet et al., 2005; Sharma et al., 2016). Vulnerability to stress also plays a key role in addiction disorders, with impaired coping with stress being implicated in the onset, maintenance, and relapse in these disorders (Koob, 2003; Koob and Schulkin, 2019). While neurodevelopmental disorders have a relatively stable, trait-like course, there are clear indications for increased stress sensitivity, as exemplified by arousal and emotion regulation problems, which in turn may lead to the development of negative valence symptoms and stress-related comorbidity (Kerns et al., 2015; J.A. Richey et al., 2015; Rommelse et al., 2011; Shaw et al., 2014b; van Eijndhoven et al., 2021). We included stress scans within the linked ICA setup as a novel feature of our approach, as we hypothesized that shared mechanisms of stress vulnerability would become visible under conditions of stress.”

2. I found Figure 1 of the overlap of the disorders interesting, and it may be interesting to examine whether the number of overlaps of the disorders or particular combinations of disorders is particularly related to any neural ICs or physiological/cognitive features.

We agree with the reviewer that the high level and diverse patterns of comorbidity that are displayed in Figure 1 are very interesting. At the same time, we think that these complicated patterns of comorbidity also illustrate the complexities (and potential limitations) of investigating subgroups and (particular) combinations of comorbidity. There are a vast number of ways in which the sample can be divided into subgroups to investigate specific patterns of comorbidity, which in some cases may result in the creation of small, underpowered subgroups. In addition, in the current project we wanted to move away from investigating specific subgroups and our aim was to focus on transdiagnostic mechanisms across our diverse sample (see point 1 above for a more elaborate rationale for this approach). Thus, although these are interesting suggestions from the reviewer, we think that adding additional analyses related to these points is outside the scope of the current study.

3. Perhaps also include Morris et al. 2022 (https://doi.org/10.1186/s12916-022-02414-0) in addition to Cuthbert & Insel, 2013, to direct to a revisited version of the RDOC framework.

We thank the reviewer for this suggestion. Based on this suggestion we have added this reference in the introduction, when we first introduce the importance of the RDoC framework. In addition, we added this reference to a newly added text in the introduction, in which we substantiate why we chose to study psychopathology dimensionally across the range from health to psychopathology (see answer to point 1 above).

4. Here I get a bit lost: "The correlational analysis (Figure 2, operation D) resulted in 87 significant correlations (FDR corrected q <0.001) between the components and measures of interest (all p-values mentioned below are FDR corrected values (see Supplemental Table C for all significant correlations)). The analysis resulted in ten correlations between the ICs and symptom dimensions. Interestingly, all symptom correlations were discovered for components that were mainly driven by functional scans. Of the nineteen correlations that were found with multimodal components, most were related to age, sex, BMI, blood pressure, and heart rate variability, while there was also an association with cognitive symptoms and a classification of ASD."Maybe include: 'of those 87, ten related to symptom dimensions, 19 with etc". The confusion comes from mixing the findings on uni-modal vs multi-modal and findings related to type of biobehavioral measure.

We thank the reviewer for this suggestion. We changed the text, so now we first start with discussing all results for our multimodal components. After this we focus more specifically on the significant results for the symptom dimensions. While reading this part of the Results section, it is important to keep in mind that there is also a multimodal component that is associated with symptom dimensions (IC30 is associated with inhibition and self-monitoring; this is a multimodal component that is mainly driven by the following functional scans: resting-state scan and stress-aftermath scan). So, it is not possible to completely separate the discussion on the unimodal vs multimodal components from the findings related to the type of biobehavioral measure.

We have changed the text in the Results section to the following:

“The correlational analysis (Figure 2, operation D) resulted in 87 significant correlations (FDR corrected q <0.001) between the components and measures of interest (all p-values mentioned below are FDR corrected values (unless mentioned otherwise) (see Supplementary File 2a for all significant correlations)). Of these 87 correlations, nineteen were with multimodal components. Most of these correlations were related to age, sex, BMI, blood pressure, and heart rate variability. In addition, we identified a multimodal component that was associated with a classification of ASD (IC32) and a multimodal component associated with cognitive symptoms (inhibition and self-monitoring) (IC30). In addition, there were eight more significant correlations between the ICs and symptom dimensions, which were all with unimodal components (See Appendix 1-figure 1 for scatterplots displaying all ten significant correlations between ICs and symptom dimensions). Interestingly, all symptom correlations were discovered for components that were mainly driven by functional scans.”

5. I don't understand this sentence "The DMN spatial map displayed within DMN connectivity (angular gyrus, precuneus, supramarginal gyrus)." Line 157/158. This sentence also seems incomplete "Additionally, the VBM feature weighted in the precentral gyrus and the MD feature in the thalamus (Figure 4B). (lines 162/163).

With respect to the ‘within DMN connectivity’: we referred here to the DMN (stress-aftermath) modality that is part of this multimodal component and reflects the DMN connectivity during the stress-aftermath scan. More specifically, this is a spatial map that reflects the connectivity of the DMN network template that was derived from Smith and colleagues 2009 (Smith et al., 2009) and was applied into dual regression (see the following paragraph in the methods section ‘MRI preprocessing and feature extraction’, and see also the text in Appendix 1 regarding the methods for the feature extraction for the functional scans). Since this DMN modality shows connectivity with brain regions that are part of/commonly associated with the DMN, this result reflects connectivity of the DMN with itself (i.e. within DMN connectivity). This finding is reported in the paragraph of the Results section that describes the results for the multimodal ASD component. Based on point 5 of reviewer 2, we now also mention explicitly in this paragraph the contribution of all 12 modalities to this multimodal ASD component. Now that we added this to the main text, we now also mention in the main text that the DMN modalities during the different functional scans (i.e. resting-state, stress and stress-aftermath scan) showed similar connectivity patterns, meaning that the different functional scans identified the same connectivity pattern (in the original manuscript we mentioned this only in the Figure legend). Based on this feedback, we have adjusted the text to further clarify the finding related to the ‘within DMN connectivity’ for the readers. Finally, we have also adjusted the other sentence (mentioned by the reviewer), for further clarification.

We have changed the text in the Results section to the following:

“The DMN modalities during the different functional scans (i.e. resting-state, stress and stress-aftermath scan) showed similar spatial configurations, meaning that these different functional scans identified the same connectivity pattern. The DMN modalities revealed loadings in multiple brain regions that are part of (or commonly associated with) the DMN (i.e. the angular gyrus, precuneus, supramarginal gyrus). Thus this reflects connectivity of the DMN network template that was applied into dual regression with DMN (associated) regions (i.e. within DMN connectivity).”

We have changed the text in the Results section to the following:

“Additionally, the VBM and MD feature showed the involvement of the precentral gyrus and thalamus respectively (Figure 4B).”

6. Anxiety (symptoms and sensitivity) and sleep problems seem to be pretty low, as are mood symptoms. Are distributions given somewhere else (outside Table 1)?

Based on this point (and also the other points raised by the reviewers), we think it is important to give further insight into the distribution of the data from the symptom questionnaires. Since the scores on the symptom questionnaires are not normally distributed we display the median and range of the scores in Table 1. Given the diversity of our sample and the accompanying diversity in scores on the symptom questionnaires we realize that the median and range may give too limited insight into these symptoms. In addition, the median may for example be influenced by a large proportion of the participants having relatively low (or high) scores on a questionnaire. We have now made dot plots (Appendix 1-figure 3) in order to visualize the distribution of the symptom dimensions in our sample. Based on (among others) point 2 of reviewer 2, we have made these dot plots for broad diagnostic subgroups in our sample. For this purpose, we have divided our sample in the following 4 broad, heuristic subgroups: (1) mentally healthy control group, (2) neurodevelopmental group, (3) stress-related group and (4) the comorbidity group. We have added text to the methods section and Appendix 1 to describe how we divided our sample into these subgroups. We think that these dot plots illustrate the diversity of our sample, and the variation across participants related to how strongly the symptoms are present that are measured with the various symptom questionnaires (like the mood symptoms on the IDS-SR questionnaire). Taken together, we think that this new Figure gives more insight into our results and is an important addition to the manuscript.

The following text was added to the methods section:

“To provide further insight into the distribution of the symptom dimensions across broad diagnostic groups in our sample, we display these results in dot plots (Appendix 1-figure 3). For this purpose we divided our sample in the following four subgroups: mentally healthy control group, stress-related group, neurodevelopmental group, and comorbidity group (see Appendix 1 for description of these subgroups).”

The following text was added to Appendix 1:

“In order to provide more insight into the distribution of the symptom dimensions across broad diagnostic groups, we divided our samples in the following four broad subgroups: mentally healthy control group, stress-related group, neurodevelopmental group, and comorbidity group. The neurodevelopmental disorders (i.e. ASD and ADHD) were grouped together since both are lifelong disorders that start in early childhood and have a relatively stable, trait-like course and shared heritability (Franke et al., 2018; Rommelse et al., 2011). The group of stress-related disorders consists of the mood, anxiety, and addiction disorders. The mood and anxiety disorders share a common underlying dimension (Kotov et al., 2017), with a maladaptive stress response as a central feature in these disorders (de Kloet et al., 2005; Sharma et al., 2016). The addiction disorders are for this purpose also included in the stress-related group, given the important role of stress in the onset, maintenance and relapse in these disorders (Koob, 2003; Koob and Schulkin, 2019). The comorbidity group consists of patients with both a stress-related and a neurodevelopmental disorder.”

7. Related to participants, given this information is in the supplement, could it be added when and how patients were recruited? Related to their in/out-patient treatment? Was remission allowed? How was the distribution of past/current diagnoses in each group? What was the correlation between all symptomatic and biobehavioural markers? Could a correlational table/graph help here? Also the relation with medication helps the reader to put everything into context.

We agree with the reviewer that the addition of this information helps to better understand the sample and to put the results into context. All patients were recruited from our outpatient clinic. Patients were only included if they had at least one current diagnosis and all diagnoses displayed in Figure 1 are current diagnoses. We have added this information and also information related to the other questions to the manuscript (see below).

The following text was added to the methods section:

“This study uses the database of the MIND-Set project (van Eijndhoven et al., 2021), which includes adult patients with ASD, ADHD, addiction, mood, and/or anxiety disorders. Patients were included in the present study if they had at least one current diagnosis in one of these categories.”

The following text was added to Figure 1 (venn diagram):

“All diagnoses in this Venn diagram represent current diagnoses.”

The text was changed in Appendix 1 to the following:

“MIND-Set (Measuring Integrated Novel Dimensions in Neurodevelopmental and Stress-related Mental Disorders) is an observational cross-sectional study. Inclusion of patients took place at the moment of intake at the outpatient clinic of the psychiatry department of the Radboud university medical center (Radboudumc), Nijmegen, the Netherlands. The MIND-Set study has been approved by the Ethical Review Board of the Radboudumc (Nijmegen, the Netherlands) and all participants signed an informed consent form before participation (van Eijndhoven et al., 2021). Data for our present study were collected from June 2016 through July 2020.”

The following text was added to the manuscript (to refer to the Table in which we now display all correlations between our biobehavioral measures of interest):

“Finally, we performed Spearman correlations (non-normal distribution) between all our biobehavioral measures of interest to provide further insight into these relationships (see Supplementary File 1).”

The following text was added to the manuscript:

“We refer to Appendix 1-Table 3 for information regarding psychotropic medication use at the time of the MRI scan.”

The following text (related to medication and substance use) was added to the limitations section in the discussion:

“Finally, although we did include measures related to psychotropic medication and substance use in our correlational analyses, we did not correct for these factors in the analyses. We did not include these factors as covariates, as these factors are not independent from our other measures of interest, like the symptom dimensions and diagnostic labels. Future studies in samples that are well matched on these measures of interest and only differ in medication status or substance use could help to further disentangle the effects of these factors from other aspects of psychopathology.”

Reviewer #1 (Recommendations for the authors):The authors applied a novel integrative analysis called linked independent component analysis (ICA) by which they successfully identified the combinations of abnormal structural and functional brain features that were associated with cross-diagnostic symptom dimensions in an objective, data-driven manner. The combination of the novel multimodal analytical technique and a large-scale, extensively phenotyped transdiagnostic dataset is a good example of the research direction endorsed in the Research Domain Criteria (RDoC).

We want to thank the reviewer for these kind words and the clear suggestions for improvement, to which we respond in detail below.

1. Although the linked ICA is suitable for identifying the co-varying patterns of abnormalities in a multimodal neuroimaging dataset in a data-driven manner, it would be fair to explain some drawbacks. In particular, the problem of the sign indeterminacy of ICs may be critical for interpreting the identified patterns. This is particularly relevant when the authors stated that some ICs showed "negative" correlations with cognitive and affective measures (e.g. p.6-7). It is also possible that the components with reversed signs are positively correlated with the measures.

We thank the reviewer for raising this important methodological point. Linked ICA is indeed characterized by sign indeterminacy, meaning that the signs (positive and negative) of the component loadings and corresponding components are ambiguous (Comon and Jutten, 2010). In our study, we have inferred the direction of the signs by investigating the relationship between age and global grey matter volume, since this represents a well-known and relatively strong relationship, especially in an adult sample with a large age span like our sample (Good et al., 2001). More specifically, we used component 1 for this inference, which is driven by the VBM modality (contribution: 82.7%). This component covers the grey matter of the entire brain and reflects global grey matter volume and has a negative correlation with age (r_s_ = -0.50, p = 9.7E-18; NB: this is the FDR corrected spearman correlation from the main analysis in the manuscript) (Supplementary File 2a, Appendix 1-figure 4). Based on this comment, we have added additional information to the methods section to describe how we inferred the direction of the associations between the components and our biobehavioral measures of interest. While we think that this is a valid way to determine the direction of the relationships, we agree that the sign indeterminacy is a limitation of linked ICA, and we have added information related to this to our limitations section in the discussion. We think that the addition of this information, and the associated limitations, have improved the manuscript.

The following paragraph was added to the methods section:

“Direction of correlational results:

It should be noted that linked ICA is characterized by sign indeterminacy, meaning that the signs (positive or negative) of the component loadings and corresponding components are ambiguous (Comon and Jutten, 2010). To understand the direction of the correlations, we inferred the direction of the signs by investigating the relationship between global grey matter volume and age, since this is a well-known and relatively strong relationship in an adult sample with a large age span (Good et al., 2001), like our sample. For this purpose we used IC1, which is driven by the VBM modality (contribution: 82.7%). This component covers the whole brain and reflects global grey matter volume (Appendix 1-figure 4). Our correlational results (Supplementary File 2a) show that this component is negatively correlated with age (r_s_ = -0.50, p < 0.001). In line with extensive evidence for a decrease in global grey matter volume related to aging (Good et al., 2001), we can infer from this that younger age should be related to a higher positive z-stat score on this component, and that the positive signs on the components indeed reflect a positive signal.”

The following text has been added to the limitations section in the discussion:

“However, our study has to be interpreted in light of some limitations. First, linked ICA is characterized by sign indeterminacy, which made it necessary to infer the direction of the associations between our components and biobehavioral measures of interest, by studying the relationship between global grey matter volume and age. While we think that this is a valid way of determining the direction of the relationships for our results, we would like to note here that it is a limitation that the direction of the effects is not a mathematical certainty but an inference. Future studies should focus on replicating the findings described in this study.”

Text for Appendix 1-figure 4:

“Appendix 1-figure 4. (1) Independent component 1 (IC1) is driven by the voxel-based morphometry (VBM) modality (82.7%) and reflects global grey matter volume. (**2)** The subject loadings on IC1 were negatively correlated with age (r_s_ = -0.50, p = 9.7E-18).”

2. Another concern is that the number of ICs was prespecified as 50 – how was it determined?

We thank the reviewer for raising this point. In the literature it is recommended to choose a model order for the linked ICA decomposition that constitutes less than 25% of the sample size (Groves et al., 2012, 2011), and also that the “optimal” dimensionality depends on the detail desired from the decomposition (Groves et al., 2012). In line with regular independent component analysis (Smith et al., 2009), components that are identified at a lower dimensionality with linked ICA may split into smaller subdivisions at a higher dimensionality (Groves et al., 2012). Since we were interested in large-scale networks (for which we also studied the spatial maps in our functional scans), we decided a priori to choose for a relatively low-dimensional decomposition. In line with the recommendation for a decomposition into less components than 25% of the sample size (Groves et al., 2012, 2011), we would have to decompose our data into less than 74 components (295 * 0.25 = 73.75). Groves and colleagues have shown that it was possible to robustly perform a linked ICA decomposition of the data of their 484 participants into 50 components (Groves et al., 2012). In addition, Francx and colleagues studied a sample with ADHD and non-ADHD subjects, with a similar sample size as our sample (n = 333), and also used linked ICA to decompose their MRI data into 50 independent components (Francx et al., 2016). Based on these studies and considerations, we decided a priori to also decompose our data into 50 independent components. Importantly, in line with our goal, our results showed that we were able to identify components that reflected large-scale networks, with among others the four components that we describe in our Results section that reflect the connectivity of the executive control network (ECN) with itself and with the frontoparietal network (FPN) (i.e. IC7, IC8, IC13, and IC30). Based on this comment we changed the following text in the methods section to further clarify our rationale for choosing our current model order.

The following text was added to the methods section:

“In general, the linked ICA model order is recommended to be less than 25% of the sample size (Groves et al., 2012, 2011). In addition, the “optimal” dimensionality depends on the detail desired from the decomposition (Groves et al., 2012), as it has been shown that components that are identified with linked ICA at a lower dimensionality may split into finer subdivisions at a higher dimensionality (Groves et al., 2012). Because of our interest in large-scale networks we decided a priori to choose for a relatively low -dimensional decomposition. In line with the lower-dimensionality decomposition performed by Groves and colleagues (Groves et al., 2012), we chose a priori to decompose our data into 50 independent components.”

3. The study seems motivated by the transdiagnostic and dimensional approach, but the results included a multimodal component (IC32) categorically associated with the diagnosis of ASD. It would be interesting, from the perspective of the dimensional approach, to examine in an ad-hoc manner whether this component was correlated with continuous variables for the ASD trait as measured by the AQ among people who do not have the ASD diagnosis.

We thank the reviewer for this suggestion. While the main goal in this study was indeed to identify components that were related to transdiagnostic (symptom) dimensions, we think that it is an interesting finding that we found a multimodal component that was associated with a diagnosis of ASD. We think that is especially interesting that this component was specifically associated with a diagnosis of ASD and not with the ASD (related) symptom dimensions (as measured with the AQ) across our whole sample. Based on this comment from the reviewer we think that this finding warrants a more in-depth discussion. In addition, we agree with the reviewer that it is an important addition to further explore these results with post-hoc correlations. We think it is most appropriate to perform these post-hoc correlations for the AQ subscales as well in the participants who do not have an ASD diagnosis, as in the group that does have an ASD diagnosis. Based on this we have added these analyses and results to the manuscript and adapted the discussion. Finally, we also slightly edited the paragraph in our discussion related to the multimodal ASD component, to include a recent linked ICA study in ASD by Mei and colleagues (Mei et al., 2022). We think these changes constitute important additions to the manuscript.

The following text was added to the Results section:

“While this *multimodal ASD component* was associated with a classification of ASD, it was not correlated with any of the subscales of the Autism spectrum Quotient-50 (AQ-50). To further explore these results, we performed post-hoc correlations between IC32 and the AQ-50 subscales for the patients with a classification of ASD and the participants without ASD separately (uncorrected for multiple comparisons). The only significant correlation was found within the ASD group for the ‘social skill’-subscale (r_s_ = -0.237, p_uncorrected_ = 0.037) (see Appendix 1 for all correlational results).”

The following text was added to Appendix 1:

“Post-hoc correlations between the multimodal ASD component and AQ-50 subscales:

Post-hoc correlations were performed between the multimodal ASD component (IC32) and the AQ-50 subscales for the patients with ASD and participants without ASD separately. In the patients with ASD the following results were found: social skill: r_s_ = -0.237 (p_uncorrected_ = 0.037), difficulty with change/attention switching: r_s_ = -0.179 (p_uncorrected_ = 0.117) , communication: r_s_ = -0.096 (p_uncorrected_ = 0.404), imagination r_s_ = -0.052 (p_uncorrected_ = 0.652), attention to detail: r_s_ = -0.122 (p_uncorrected_ = 0.287). For the participants without ASD the results showed the following: social skill: r_s_ = -0.017 (p_uncorrected_ = 0.807), difficulty with change/attention switching: r_s_ = -0.039 (p_uncorrected_ = 0.563), communication: r_s_ = 0.024 (p_uncorrected_ = 0.721), imagination r_s_ = 0.037 (p_uncorrected_ = 0.587), attention to detail: r_s_ = -0.035 (p_uncorrected_ = 0.605).”

The text in the discussion was adjusted to the following:

“Although set within transdiagnostic research, the linked ICA analysis was sensitive enough to pick up a multimodal component that is associated with a traditional, diagnostic classification of ASD. This relationship was specifically found for ASD, as this component was not associated with the other diagnostic groups, nor with the variable that divided our sample in mentally healthy participants versus patients. The multimodal ASD component loaded on the DMN, precentral gyrus and thalamus. Interestingly, these regions have been implicated in ASD by earlier studies and are associated with core domains of this disorder, like theory of mind (Murdaugh et al., 2012), motor problems (Duffield et al., 2013; Mahajan et al., 2016) and sensitivity to sensory stimuli respectively (Ayub et al., 2021). While both Itahashi and colleagues (2015) and Mei and colleagues (2022) also implicated these regions in ASD in their linked ICA analyses, there were also differences with our findings. In the study of Itahashi and colleagues (2015) these regions did not show up together in one component, and the multimodal component of Mei and colleagues (2022) loaded more extensively on the white matter tracts. While differences with our findings may be related to differences in study setup and specific sample characteristics (e.g. related to the included MRI modalities, patterns of psychiatric comorbidity, IQ, and age range of the participants), this may also be related to the heterogeneous nature of psychiatric classifications like ASD (Itahashi et al., 2015). Still, these multimodal results may help to get a more coherent understanding of the neurobiology of ASD, by not only showing which brain regions are involved, but also how these findings covary across different modalities.

Importantly, this *multimodal ASD component* was associated with a diagnostic label of ASD, but not with the subscales of the autism spectrum quotient-50 (AQ-50) across the whole sample. As a diagnosis of ASD reflects a complex clinical phenotype that spans several functional domains, a multimodal brain component spanning brain regions that are involved in these various functions may be more strongly associated with such a complex phenotype than with specific symptom dimensions (Mei et al., 2022). While the AQ-50 subscales did not show any correlations with this *multimodal ASD component* across the whole sample, the post-hoc tests showed that this component was specifically associated with social skill symptoms within the ASD group. This may be explained by the loading of this component in the DMN, which is involved in social functions related to theory of mind (Murdaugh et al., 2012). Based on these results, an alternative explanation for not finding correlations for the AQ-50 across the whole sample could be related to limitations regarding the application of this questionnaire in our diverse transdiagnostic sample. The AQ-50 may not measure a uniform dimension across our participants. Higher scores on this self-report questionnaire may stem from ASD, but could also stem from other causes, like social interaction problems due to (social) anxiety, or patients may score higher if they have an overly negative judgement of their social skills (e.g. in depression). Thus, when investigating brain behavior relationships using the AQ-50, it may be important to also take into account the judgement of a trained clinician regarding if a participant has ASD or not. These considerations are important, since one of the central goals of RDoC is the identification of reliable and valid measures that can be applied in transdiagnostic research (Morris et al., 2022).”

4. The length of the stress scan: 150 sec may be a little short to identify individual FC network components.

We agree with the reviewer that it is important to consider the length of the scans that were included in the analysis. However, we do think that the length of the stress scan is sufficient for studying the functional connectivity of our networks of interest. We would like to start with emphasizing that a previous study at our center has successfully used this same stressful movie clip (of 150 seconds) to identify functional brain networks (Hermans et al., 2011). In addition, it is important to note that we did not run single subject independent components analysis (ICA) on this stress scan, but performed the analysis using dual regression (Filippini et al., 2009). We also did not identify our networks of interest (i.e. DMN, ECN, and FPN) in this stress scan itself. We used previously identified large scale network maps (identified by Smith and colleagues (2009)), and applied these as spatial maps into dual regression. In this way we have designed a robust analysis pipeline for our current study. Furthermore, our multi-band 6 protocol with a TR of 1,000 ms allowed us to collect 150 volumes during this stress scan. This is a similar number of volumes that various studies have collected to perform analysis using dual regression (e.g. by performing a 5 minute resting-state scan with a TR of 2,000 ms) (e.g. Cowdrey et al., 2014; Pietzuch et al., 2021; Schmidt et al., 2015). Importantly, our results also support that we were able to identify components using this stress scan. First, we reliably identified a similar component that reflected connectivity of the ECN with itself and the right FPN during this stress scan (IC8) (Appendix 1-figure 2) as in the two longer functional scans that were each 500 volumes (i.e. the resting-state (IC13) (Appendix 1-figure 2) and stress aftermath scan (IC7) (Figure 5)). The similarity between these components is further confirmed by the fact that the spatial correlations between the group network maps between the different scans is high, especially between the network map during the stress scan and the stress-aftermath scan (cross correlations: IC8 (stress scan) with IC13 (resting-state scan): 0.58; IC8 (stress scan) with IC7 (stress-aftermath scan): 0.85). Second, the correlational analyses showed that the component identified during the stress scan (IC8) showed the same negative correlation with the ‘BRIEF-A: self-monitoring’ subscale as the other two components mentioned above. Finally, the *DMN-stress component* (IC15) (Figure 5) that we identified was driven by the stress scan (contribution: 88.4%), and showed that the stress scan was able to identify a unique component that was correlated with a relevant symptom dimension (‘ASI – mental incapacitation concerns’-subscale), that was not identified with the other functional scans. Together, these findings support that this stress scan was able to identify as well similar patterns as the other functional scans (that had a longer duration), as a unique result in the form of the *DMN-stress component*.

5. On page 6, it is stated that there were 10 correlations between symptoms and ICs. What were the components other than ICs 7, 8, 13, and 30?

In addition to the components stated in the question, there were also significant correlations between symptoms and ICs 15 and 34. Some components had significant correlations with more than one symptom. For example, IC7 was correlated with 3 symptoms (i.e. ASI: physical concerns, BRIEF-A: self-monitoring, and IDS: anxiety-somatic see Supplementary File 2a and Figure 5). To further clarify for the reader to which correlations we refer in the text, we have added a figure with scatterplots displaying all these 10 significant correlations (Appendix 1-figure 1). We refer to the answer to point 2 of reviewer 2 for a more elaborate description of these scatterplots, and why we color coded the scatterplots for our broad diagnostic subgroups (i.e. healthy controls, stress-related, neurodevelopmental, and comorbidity group).

6. On page 12, it is stated that correlation analysis for symptom dimensions was done using the summed score of 35 subscales of 8 questionnaires. I am not sure whether these scores can be summed with equal weight to form a valid score for the symptom of interest.

We would like to clarify that we ran bivariate correlations between the subject loadings on the components and the 35 subscales of in total 8 different questionnaires, thus for the symptom dimensions we ran 35 correlations per component (Appendix 1-Table 1 displays the 8 questionnaires and the subscales for each of the questionnaires). We have now removed the word ‘summed’ to prevent any confusion related to this matter.

We have changed the following text in the methods section:

For the correlational analyses, we used the scores of the 35 subscales of these eight questionnaires (Appendix 1-Table 1).

7. On page 11 and in Figure 2 legend, stress induction was done by viewing an aversive movie clip. Is there any supporting evidence that this procedure was successful in inducing stress as expected?

We thank the reviewer for raising this important point. We have now included the analyses to confirm that stress was induced successfully in the manuscript. Based on these analyses we conclude (in line with our earlier study in the MIND-Set database (van Oort et al., 2020)), that our experimentally well-controlled stressor successfully induced mild psychological stress.

We have changed the text in the ‘Procedure’-paragraph in the methods section to the following:

“Stress was induced with a mild psychological stressor using an experimentally well-controlled paradigm in the form of an aversive movie clip (Qin et al., 2009; van Oort et al., 2020). A neutral movie clip served as the control condition. We used the following two measures to assess stress levels during scanning: heart rate (beats per minute) and subjective stress (11-point rating scale: 0 = no stress, 10 = maximal stress). The subjective stress level was assessed directly after the aversive and neutral movie and the heart rate was measured during these two movie clips. For these two measures, we assessed the effects of stress using a Wilcoxon signed-rank test (non-normal distribution).”

We have added the following text to the paragraph ‘study population and general results’ in the Results section:

“The analysis confirmed that our experimentally well-controlled stressor induced mild psychological stress, with an increase in both subjective stress (median subjective stress score after neutral movie: 3, after aversive movie: 5; T = -12.50, p < 0.001) and heart rate (median heart rate during neutral movie (beats per minute): 65.59, during aversive movie: 67.10; T = -8.60, p < 0.001).”

8. The age range of 18 – 74 is quite large and I wonder whether there is any systematic difference among patients, for instance, people who have the diagnosis of ASD or ADHD and those who do not. Relatedly, on page 11, how was the diagnosis of adult ASD and ADHD made? Was it all based on the DSM without any other supporting evidence such as ADOS or ADI-R?

Based on this comment we have added information with respect to these points in our manuscript. In this revised manuscript we now also divide our sample into broad diagnostic subgroups, in order to provide more insight into the sample (see the answer to point 10 of reviewer 1 below for a more detailed explanation related to these subgroups). We divided our sample for this purpose in the following 4 broad diagnostic subgroups: (1) mentally healthy control group, (2) stress-related group, (3) neurodevelopmental group, and (4) comorbidity group. We now test for differences in age and sex between these broad groups. While there is a significant difference in age between these 4 subgroups (the stress-related group is slightly older than neurodevelopmental and comorbidity group (see results below)), there is a non-significant trend for differences in sex between the 4 subgroups (χ^2^(3) = 6.61, p = 0.085). We would like to emphasize that we have already performed post-hoc partial correlations in order to control for the effects of age and sex.

With respect to the diagnostic process for ASD and ADHD: we performed an extensive diagnostic process for these disorders (including the use of screening instruments, an extensive clinical evaluation at the psychiatry department, and the use of well-validated semi-structured interviews in the presence of a partner and/or family member). We have now added more elaborate information regarding this diagnostic process to Appendix 1, so that this process is more clear to the reader.

The following text was added to the text regarding the methods in Appendix 1:

“Age and sex across diagnostic subgroups

For our broad diagnostic subgroups (i.e. mentally healthy controls, stress-related, neurodevelopmental, and comorbidity group), we tested if there were any differences between these groups related to age (ANOVA) and sex (chi-square test).”

The following text was added to the Results section in Appendix 1:

“In order to get more insight into our sample, the sample was subdivided into the following 4 broad diagnostic subgroups: mentally healthy controls (n = 70; mean age (± standard deviation (SD)): 37.4 ± 16.0; %-male: 47.1%), stress-related (n = 84; mean age (± SD): 39.9 ± 13.8; %-male: 53.6%), neurodevelopmental (n = 55; mean age (± SD): 33.3 ± 12.2; %-male: 69.1%), and comorbidity group (n = 86; mean age (± SD): 33.9 ± 10.7; %-male: 59.3%). There were differences in age between the 4 subgroups (F(3,291) = 4.20, p = 0.006), with higher age in the stress-related group compared to the neurodevelopmental (p = 0.004), and comorbidity group (p = 0.003). There was no significant difference between the subject subgroups with respect to sex (χ^2^(3) = 6.61, p = 0.085).”

The text in Appendix 1 was changed to the following (with respect to the diagnostic process):

“Patients were diagnosed and classified by a trained clinician according to the Diagnostic and Statistical Manual of Mental Disorders (DSM) using semi-structured interviews (see van Eijndhoven et al. (2021) for an extensive description of the diagnostic process). Mood and anxiety disorders were diagnosed using the Structured Clinical interview for DSM-IV Axis I Disorders (SCID-I) (First et al., 1996), and addiction disorders with the MATE-Crimi (Schippers et al., 2010; Schippers and Broekman, 2012). Attention deficit hyperactivity disorder (ADHD) and autism spectrum disorder (ASD) were assessed with a two-step diagnostic procedure. First, screeners were used. We used the World Health Organization Adult ADHD Self-Report Scale (ASRS)-short version for ADHD screening (Adler and Kessler, 2004; Kessler et al., 2005). The ASRS is an ADHD screenings questionnaire, consisting of 6 items (cut-off >3) with good psychometric properties (Kim et al., 2013). We screened for ASD by assessing autistic traits using the Autism-spectrum Quotient (AQ-50) (50 items, cut-off >25) (Baron-Cohen et al., 2006, 2001). Next, semi-structured interviews were performed for these disorders in case of a positive score on these screening instruments or if there was a clinical suspicion on one of these disorders during the extensive three-hour clinical evaluation at the psychiatry department. We assessed the presence of ADHD with the semi-structured Diagnostic Interview for ADHD in Adults version 2.0 (Dutch: Diagnostisch Interview voor ADHD bij volwassenen 2.0 (DIVA 2.0)) (Kooij and Francken, 2010; Ramos-Quiroga et al., 2016). For ASD we administered the Dutch Interview for ASD in Adults (Dutch: Nederlands Interview ten behoeve van Diagnostiek Autismespectrumstoornissen bij volwassenen (NIDA)) (Vuijk, 2014). Both the DIVA and NIDA were completed in the presence of a partner and/or family member of the patient so that we were able to retrospectively and collaterally ascertain information on a broad range of symptoms in childhood and adulthood.”

9. I think that some important ideas behind the design of this study are only briefly mentioned in the main text, just referring to their original paper on the MIND dataset. Readers may understand the significance better if they are given an explanation about why the different kinds of disorders (ASD, ADHD, mood disorder, anxiety, addiction) can be gathered up (but not other types of disorders) and why they were particularly interested in stress-related responses for the group.

We thank the reviewer for raising this important point. We indeed think that these disorders can be gathered up in order to study transdiagnostic mechanisms across these diverse non-psychotic psychiatric disorders (i.e. ASD, ADHD, mood disorder, anxiety, addiction). It is important to study these highly prevalent disorders together in a transdiagnostic manner, as high levels of comorbidity among these disorders suggest shared underlying mechanisms (van Eijndhoven et al., 2021). In addition, numerous studies (often investigating these disorders separately) provide converging evidence that symptoms across various major domains (like negative valence and cognitive systems domain) cut across the diagnostic boundaries of these disorders (Kerns et al., 2015; Koob and Schulkin, 2019; McTeague et al., 2016; Mulders et al., 2022; J Anthony Richey et al., 2015; Shaw et al., 2014a; van Eijndhoven et al., 2021; Woody and Gibb, 2015). Importantly, disturbances in the same brain networks may underlie these symptom dimensions across these diverse disorders (McTeague et al., 2016; Menon, 2011). Therefore, it is crucial to adopt a transdiagnostic approach in order to identify the brain basis for symptom dimensions that transcend these diagnostic categories. Finally, we do want to emphasize here that we do not want to make any claims that the possibility to study transdiagnostic brain mechanisms is limited to the groups included in this study. Future studies should investigate if our findings can be replicated in other groups of disorders or in a transdiagnostic sample with an even a broader range of psychiatric disorders.

With respect to our interest in the stress-related responses: clinically it is well established that there is increased stress sensitivity across a broad range of psychiatric disorders (Ingram and Luxton, 2005). For depression and anxiety disorders it is well known that this is a central feature of these disorders (de Kloet et al., 2005; Sharma et al., 2016). However, also for the other disorders included in this study, there are clear indications for increased stress sensitivity. In addiction disorders, stress plays an important role in the onset, maintenance and recurrence (relapse) in these disorders (Koob, 2003; Koob and Schulkin, 2019). And also in neurodevelopmental disorders there are clear indications for increased stress sensitivity, such as arousal and emotion regulation problems (Kerns et al., 2015; J.A. Richey et al., 2015; Shaw et al., 2014b). We included the stress scans in our experimental set up, as we hypothesized that shared mechanisms of stress vulnerability would become visible under conditions of stress.

Based on this comment we have adjusted the introduction in order to further explain our rationale for the inclusion of these disorders and for the inclusion of stress scans in the experimental set up. We think that these adjustments have significantly improved the introduction.

We have changed the text in the introduction to the following:

“Therefore, in the current study, we used linked ICA within a transdiagnostic, dimensional framework, using the MIND-Set database (Measuring Integrated Novel Dimensions in Neurodevelopmental and Stress-related Mental Disorders) (van Eijndhoven et al., 2021). This database provides us with a sample of mentally healthy participants and patients with diverse, highly prevalent non-psychotic psychiatric disorders (i.e. mood disorders, anxiety disorders, addiction, ASD, ADHD, and their comorbidity). A multimodal imaging battery was performed and all participants were deeply phenotyped. An extensive set of biobehavioral measures was collected, including symptom dimensions, biological/physiological measures (like cortisol and heart rate variability) and also more general measures of physical and mental health. Since psychopathology can best be understood on a continuum from health to mental illness, we also included mentally healthy participants, which allows us to study the brain and biobehavioral dimensions of interest along a wider range from health to psychopathology (Kotov et al., 2017; Morris et al., 2022; van Oort et al., 2022).

We investigated the diverse non-psychotic psychiatric disorders (i.e. mood disorders, anxiety disorders, addiction, ASD, ADHD) together, as high levels of comorbidity suggest shared underlying mechanisms (van Eijndhoven et al., 2021). In addition, numerous studies that investigated these disorders separately provide converging evidence that symptoms across various major domains cut across the diagnostic boundaries of these disorders, with among others transdiagnostic symptom dimensions related to the negative valence domain (e.g. repetitive negative thinking), and cognitive systems domain (e.g. regarding cognitive control problems) (Kerns et al., 2015; Koob and Schulkin, 2019; McTeague et al., 2016; J Anthony Richey et al., 2015; Shaw et al., 2014a; van Eijndhoven et al., 2021; Woody and Gibb, 2015). Interestingly, disturbances in the same brain networks may underlie these core symptom dimensions across these diverse disorders (McTeague et al., 2016; Menon, 2011). Therefore, it is crucial to adopt a transdiagnostic approach in order to identify the brain basis for symptom dimensions that transcend these diagnostic categories.

In our multimodal neuroimaging battery, we included functional scans under conditions of rest as well as under conditions of mild experimentally induced stress. It is clinically well established that vulnerability to stress is a common feature across a broad range of psychiatric disorders (Ingram and Luxton, 2005). Mood and anxiety disorders are characterized by a maladaptive stress response as their central feature (de Kloet et al., 2005; Sharma et al., 2016). Vulnerability to stress also plays a key role in addiction disorders, with impaired coping with stress being implicated in the onset, maintenance, and relapse in these disorders (Koob, 2003; Koob and Schulkin, 2019). While neurodevelopmental disorders have a relatively stable, trait-like course, there are clear indications for increased stress sensitivity, as exemplified by arousal and emotion regulation problems, which in turn may lead to the development of negative valence symptoms and stress-related comorbidity (Kerns et al., 2015; J.A. Richey et al., 2015; Rommelse et al., 2011; Shaw et al., 2014b; van Eijndhoven et al., 2021). We included stress scans within the linked ICA setup as a novel feature of our approach, as we hypothesized that shared mechanisms of stress vulnerability would become visible under conditions of stress.

10. I found Figure 1 of the overlap of the disorders interesting, and it may be interesting to examine whether the number of overlaps of the disorders or particular combinations of disorders is particularly related to any neural ICs or physiological/cognitive features.

We agree with the reviewer that the high level and diverse patterns of comorbidity that are displayed in Figure 1 are very interesting. At the same time, we think that these complicated patterns of comorbidity also illustrate the complexities (and potential limitations) of investigating subgroups and (particular) combinations of comorbidity. There are a vast number of ways in which the sample can be divided into subgroups to investigate specific patterns of comorbidity, which in some cases may result in the creation of small, underpowered subgroups. In addition, in the current project we wanted to move away from investigating specific subgroups and our aim was to focus on transdiagnostic mechanisms across our diverse sample (see the response to point 9 above for a more elaborate rationale for this approach). Thus, although these are interesting suggestions from the reviewer, we think that adding additional analyses related to these points is outside the scope of the current study.

Reviewer #2 (Recommendations for the authors):In this work, van Oort et al. analyzed multi-modal brain data and characterization of symptomatology and other biobehavioral markers in 225 people diagnosed (according to DSM-IV) with unimodal depression, anxiety disorders (including PTSD, substance use disorders, autism spectrum disorder, or attention-deficit hyperactivity disorder) and 70 people without such diagnosis and not seeking help for their mental health problems.The work is inspired by the RDOC framework aimed at understanding mechanisms underpinning specific symptom domains occurring in multiple psychiatric disorders. This approach allows for more dimensional characterization across the spectrum from 'health' to 'disordered' on multiple domains of functioning.All participants underwent a series of scans, including structural scanning for grey matter morphometry and white matter characterization based on DTI. Also, before, during, and after stress induction, functional acquisitions to study correlational patterns over time were obtained. From all this data, linked ICA was used to obtain independent components, of which some were based on one modality and some included multiple scanning modalities. Next Spearman correlations of these independent components with the symptom dimensions were calculated, in which a dimensional take towards depressive symptomatology was included (not looking at total depression, but at dimensions of mood and cognition, anxiety arousal, and sleep, following the tripartite model of depression).This study must have been an enormous amount of work and I applaud the authors for making the report so readable and easy to digest as if complexity and acquisition and analysis effort were not an issue. The supplements are informative.

We want to thank the reviewer for these kind words and for the important feedback, to which we respond in detail below.

1. I have some questions that relate to the study design and assumptions underpinning the choice of methods which diagnosis to include, and which resting state 'challenge' included. Especially given the inclusion of ASD and ADHD, not primarily stress disorders, why was the stress challenge chosen? Also, it might help to include a short section explaining the unique and common symptom dimensions of the included psychopathologies and what we might uniquely learn from studying them jointly. This will help clarify the aim of the study.

We thank the reviewer for raising these important points that lie at the center of the study design. Since these points overlap with point 9 of reviewer 1, we refer to that point for the answer to these questions and the changes we made to the manuscript based on this feedback.

2. For interpretation of the data, it would help to see an overview of group differences (for each diagnostic class [i.e.] hc, mood, anxiety, asd, substance use, adhd) for the biobehavioural measures (in the supplement). I understand group differences are not the aim here, but for interpreting the results further, it helps to see if the scores are basically a continuous reflection of diagnostic labels (yes/no), is not so specific to certain labels. For the same reason, a correlation table of how the symptomatic (and other markers) correlated with each other would help.Correlational plots of the brain-behaviour relations showing the 'diagnostic' label (color-coded dots) would help to see the transdiagnostic relevance.

We thank the reviewer for raising these points and for these suggestions. We agree with the reviewer that it improves the manuscript to give further insight into the sample and our measures of interest. As the reviewer already mentions, the aim of the current study is not to investigate group differences, and we aimed to move away from investigating specific subgroups. However, we do agree that it improves the manuscript to provide more insight into how our results are related to broad diagnostic groups in our sample. For this purpose, we have divided our sample in the following 4 broad, diagnostic subgroups: healthy control group, neurodevelopmental group, stress-related group and the comorbidity group (see the text below this response for a more elaborate description of these subgroups). Based on this comment, we have added the following 2 figures to the manuscript: (1) scatterplots (Appendix 1-figure 1) for the ten significant correlations between the symptoms and independent components (ICs) (in which the data is color coded according to these broad diagnostic groups), and (2) dot plots (Appendix 1-figure 3) for all 35 symptom dimensions in order to display the distribution of these symptoms dimensions across the 4 broad diagnostic groups. As can be seen in the dot plot figures, there is much overlap in the scores on the symptom questionnaires across the different diagnostic subgroups. We think this displays nicely the transdiagnostic occurrence of these symptom dimensions. Finally, based on this feedback, we have also added a Table displaying the correlations between our biobehavioral measures of interest (Supplementary File 1). We think that these Figures and Table are important additions and substantially improved the manuscript.

The following text was added to the methods section:

“To provide further insight into the distribution of the symptom dimensions across broad diagnostic groups in our sample, we display these results in dot plots (Appendix 1-figure 3). For this purpose we divided our sample in the following four subgroups: mentally healthy control group, stress-related group, neurodevelopmental group, and comorbidity group (see Appendix 1 for description of these subgroups).”

The following text was added to Appendix 1:

“In order to provide more insight into the distribution of the symptom dimensions across broad diagnostic groups, we divided our samples in the following four broad subgroups: mentally healthy control group, stress-related group, neurodevelopmental group, and comorbidity group. The neurodevelopmental disorders (i.e. ASD and ADHD) were grouped together since both are lifelong disorders that start in early childhood and have a relatively stable, trait-like course and shared heritability (Franke et al., 2018; Rommelse et al., 2011). The group of stress-related disorders consists of the mood, anxiety, and addiction disorders. The mood and anxiety disorders share a common underlying dimension (Kotov et al., 2017), with a maladaptive stress response as a central feature in these disorders (de Kloet et al., 2005; Sharma et al., 2016). The addiction disorders are for this purpose also included in the stress-related group, given the important role of stress in the onset, maintenance and relapse in these disorders (Koob, 2003; Koob and Schulkin, 2019). The comorbidity group consists of patients with both a stress-related and a neurodevelopmental disorder.”

The following text was added to the Results section:

“See Appendix 1-figure 1 for scatterplots displaying all ten significant correlations between ICs and symptom dimensions.”

3. A discussion on whether the inclusion of people with a (current?) diagnosis of psychiatric disorders and also including asymptomatic controls is the optimal approach for this RDOC-style analysis would strengthen the discussion. Has it been considered to leave people without a lifetime diagnosis out?

We have indeed discussed this point within our research team and considered both options. We have chosen to include the healthy controls, in line with an earlier study on this MIND-Set database (van Oort et al., 2022). An important reason to include the healthy controls is that the evidence to date suggests that psychopathology can best be understood on a continuum from health to mental illness (Kotov et al., 2017; Morris et al., 2022). The inclusion of this healthy group allows us to investigate our biobehavioral measures of interest along a wider range from health to psychopathology (Morris et al., 2022; van Oort et al., 2022). Based on this comment, we have adjusted our introduction.

The following text was added to the introduction:

“Since psychopathology can best be understood on a continuum from health to mental illness, we also included mentally healthy participants, which allows us to study the brain and biobehavioral dimensions of interest along a wider range from health to psychopathology (Kotov et al., 2017; Morris et al., 2022; van Oort et al., 2022).”

4. I was a bit surprised to see the first analyses cover the loading of ASD diagnosis present vs absent and not the correlation with a continuous autism spectrum measure. It's worth highlighting and discussing that the ASD (yes/no) classification outperformed continuous ASD measures. This is important for the worth of diagnostic labels vs. continuous characterizations. Also, can the authors discuss whether the sampling of 'patients' vs 'healthy controls' plays a role here and might explain the strength of the effect of a diagnostic label?

We agree with the reviewer that it is a striking finding and that this deserves a more in-depth discussion. We would like to emphasize that we included a variable in the correlational analysis called ‘Psychiatrically healthy (yes/no)’ (see Appendix 1-Table 2), which divided the participants into the groups mentioned by the reviewer (i.e. ‘patients’ versus ‘healthy controls’). Since this division in ‘healthy controls’ versus ‘patients’ did not have a significant correlation with the *‘multimodal ASD component’* (or any other component in the main analysis), we do not think that this result is driven by a general difference between patients and psychiatrically healthy controls. Furthermore, based on point 3 of reviewer 1, we have now performed post-hoc correlations between the *multimodal ASD component* and the subscales of the Autism spectrum Quotient-50 (AQ-50) in the patients with a classification of ASD and the participants without ASD separately.

Based on this feedback, the feedback of reviewer 1, and the results from the post-hoc correlations, we have adapted the discussion. We now provide a more in-depth discussion of these results and how our findings could be interpreted

The following text was added to the Results section:

“While this *multimodal ASD component* was associated with a classification of ASD, it was not correlated with any of the subscales of the Autism spectrum Quotient-50 (AQ-50). To further explore these results, we performed post-hoc correlations between IC32 and the AQ-50 subscales for the patients with a classification of ASD and the participants without ASD separately (uncorrected for multiple comparisons). The only significant correlation was found within the ASD group for the ‘social skill’-subscale (r_s_ = -0.237, p_uncorrected_ = 0.037) (see Appendix 1 for all correlational results).”

The following text was added to Appendix 1:

“Post-hoc correlations between the multimodal ASD component and AQ-50 subscales:

Post-hoc correlations were performed between the multimodal ASD component (IC32) and the AQ-50 subscales for the patients with ASD and participants without ASD separately. In the patients with ASD the following results were found: social skill: r_s_ = -0.237 (p_uncorrected_ = 0.037), difficulty with change/attention switching: r_s_ = -0.179 (p_uncorrected_ = 0.117) , communication: r_s_ = -0.096 (p_uncorrected_ = 0.404), imagination r_s_ = -0.052 (p_uncorrected_ = 0.652), attention to detail: r_s_ = -0.122 (p_uncorrected_ = 0.287). For the participants without ASD the results showed the following: social skill: r_s_ = -0.017 (p_uncorrected_ = 0.807), difficulty with change/attention switching: r_s_ = -0.039 (p_uncorrected_ = 0.563), communication: r_s_ = 0.024 (p_uncorrected_ = 0.721), imagination r_s_ = 0.037 (p_uncorrected_ = 0.587), attention to detail: r_s_ = -0.035 (p_uncorrected_ = 0.605).”

The text in the discussion was adjusted to the following:

“Although set within transdiagnostic research, the linked ICA analysis was sensitive enough to pick up a multimodal component that is associated with a traditional, diagnostic classification of ASD. This relationship was specifically found for ASD, as this component was not associated with the other diagnostic groups, nor with the variable that divided our sample in mentally healthy participants versus patients. The multimodal ASD component loaded on the DMN, precentral gyrus and thalamus. Interestingly, these regions have been implicated in ASD by earlier studies and are associated with core domains of this disorder, like theory of mind (Murdaugh et al., 2012), motor problems (Duffield et al., 2013; Mahajan et al., 2016) and sensitivity to sensory stimuli respectively (Ayub et al., 2021). While both Itahashi and colleagues (2015) and Mei and colleagues (2022) also implicated these regions in ASD in their linked ICA analyses, there were also differences with our findings. In the study of Itahashi and colleagues (2015) these regions did not show up together in one component, and the multimodal component of Mei and colleagues (2022) loaded more extensively on the white matter tracts. While differences with our findings may be related to differences in study setup and specific sample characteristics (e.g. related to the included MRI modalities, patterns of psychiatric comorbidity, IQ, and age range of the participants), this may also be related to the heterogeneous nature of psychiatric classifications like ASD (Itahashi et al., 2015). Still, these multimodal results may help to get a more coherent understanding of the neurobiology of ASD, by not only showing which brain regions are involved, but also how these findings covary across different modalities.

Importantly, this *multimodal ASD component* was associated with a diagnostic label of ASD, but not with the subscales of the autism spectrum quotient-50 (AQ-50) across the whole sample. As a diagnosis of ASD reflects a complex clinical phenotype that spans several functional domains, a multimodal brain component spanning brain regions that are involved in these various functions may be more strongly associated with such a complex phenotype than with specific symptom dimensions (Mei et al., 2022). While the AQ-50 subscales did not show any correlations with this *multimodal ASD component* across the whole sample, the post-hoc tests showed that this component was specifically associated with social skill symptoms within the ASD group. This may be explained by the loading of this component in the DMN, which is involved in social functions related to theory of mind (Murdaugh et al., 2012). Based on these results, an alternative explanation for not finding correlations for the AQ-50 across the whole sample could be related to limitations regarding the application of this questionnaire in our diverse transdiagnostic sample. The AQ-50 may not measure a uniform dimension across our participants. Higher scores on this self-report questionnaire may stem from ASD, but could also stem from other causes, like social interaction problems due to (social) anxiety, or patients may score higher if they have an overly negative judgement of their social skills (e.g. in depression). Thus, when investigating brain behavior relationships using the AQ-50, it may be important to also take into account the judgement of a trained clinician regarding if a participant has ASD or not. These considerations are important, since one of the central goals of RDoC is the identification of reliable and valid measures that can be applied in transdiagnostic research (Morris et al., 2022).”

5. Also for the ASD results, does the correlation between stress and non-stress ICNs matter? What does it mean for example that the DMN in the stress-aftermath contributed to ASD (yes/no) and the pre-stress DMN did not? Can one say something about the relevance of the stress-reactivity, or just that it's more sensitive to pick up interindividual variation?

in an attempt to write up the results concisely, we only mentioned some of the modality contributions to the *multimodal ASD component* in the main text, and referred to Figure 4, where we displayed the contributions of all modalities. Based on this comment, we realize that the current text in the Results section may lead to misinterpretations of the results. As displayed in Figure 4A the DMN modalities from the different functional scans did contribute to this component (DMN during rest: 17.5%, DMN during stress: 11.1%, DMN during stress-aftermath: 17.9%). Since the DMN modality during the resting-state scan contributes more or less the same to this component as the DMN during the stress-aftermath period, we cannot make any claims related to the rest or stress scans being more or less important for this component. We do think that it is important to state the general importance of the DMN modalities for this component, since together the DMN modalities from the different functional scans contribute 46.5% to this component. Based on this comment, we have adapted the Results section in order to make the modality contributions more clear to the reader.

We have changed the text in the Results section to the following:

“The analysis revealed a multimodal component associated with ASD (IC32). The relative contributions from the different modalities to this component were: 10.1% for VBM, 8.8% FA, 6.1% MD, 17.5% DMN (rest), 11.1% DMN (stress), 17.9% DMN (stress-aftermath), 6.3% FPN (rest), 5.4% FPN (stress), 7.8% FPN (stress-aftermath), 3.8% ECN (rest), 1.6% ECN (stress), 3.6% ECN (stress-aftermath) (Figure 4A). Besides the relatively large contributions to this component from the DMN modalities themselves, various other modalities also showed the involvement of regions of the DMN, highlighting its centrality within this component.”

6. In the discussion, I miss a discussion on the importance of including a stress challenge in their study. Is it worth the trouble? It seems that the correlation between symptoms (cognitive) did not really differ between pre-, during, and after stress. Was a symptom x IC interaction calculated? The fact that a correlation between negative valence with only IC7 was observed, does not mean that this correlation was specific to IC7, it may have occurred just subthreshold for IC8, IC13, and IC30. I believe this information and exploration are needed to conclude that "Together, this indicates the importance of this ECN-stress aftermath component for a broad spectrum of health-related measures".

We agree with the reviewer that we should further reflect on the value of including a stress challenge in the discussion. With respect to the correlational results: in order to give more insight into these results, we have now added the correlation coefficients of the other ECN-FPN components (i.e. IC8, IC13, and IC30) with the negative valence symptoms to the manuscript. In addition, we now perform post-hoc analyses to investigate if the correlation coefficients between the *ECN stress-aftermath component* and negative valence symptoms differ from the correlation coefficients for the other ECN-FPN components mentioned above (using Fisher's r to z transform). We found for example the following spearman correlations for the IDS anxiety/somatic subscale in our main analyses: IC7 (during stress-aftermath): r_s_ = -0.23, p_FDR-corrected_ = 0.007, IC13 (during resting-state): r_s_ = -0.14, p_FDR-corrected_ = 0.257. The post-hoc analyses (Fisher’s r to z transform) showed that these correlations did not differ significantly (z = -1.07, p_uncorrected_ = 0.284) (see the text below this response for a complete overview of the results from these post-hoc analyses). Based on these results and the other results from the post-hoc analyses, we think that we cannot exclude a threshold effect At the same time, we think it is important to note that relatively few correlations with the symptom questionnaires survived multiple comparison correction (there were ten significant correlations with symptom questionnaires in the main analysis, while 1,750 correlations were performed (50 ICs x 35 symptom subscales)). So, it is important to consider which correlations survive multiple comparison correction. Together, this suggests that, while we cannot exclude a threshold effect, the *ECN stress-aftermath component* was the most sensitive for finding relationships with negative valence symptoms. Thus, the stress induction may have played an important role in revealing these results (at a statistically significant level). This in turn also relates to the other point of the reviewer regarding if it is worth the effort to include a stress challenge. Our results suggest that it depends on what measures you are interested in, if it is of added value to include a stress induction procedure. For studying cognitive symptoms, it may be less important to include a stress challenge, since (as the reviewer points out) IC13 (during the resting-state scan) is also able to pick up the relationship with cognitive symptoms. Interestingly, the components that showed relationships with negative valence symptoms were specifically components related to the stress induction (i.e. the *DMN-stress component* and the *ECN-stress aftermath component*)*.* Thus, the inclusion of a stress challenge may be especially important when studying the negative valence domain. Finally, we have now removed the sentence *"Together, this indicates the importance of this ECN-stress aftermath component for a broad spectrum of health-related measures".* Based on these considerations we have made several changes to the Results section, Appendix 1, and discussion.

The text in the Results section was changed to the following text:

“While all four components are negatively correlated with cognitive symptoms, IC7 is the only component with a negative correlation with symptoms from the negative valence domain. This component negatively correlated with the following symptom dimensions: anxiety/somatic (r_s_ = -0.23, p = 0.007) and physical concerns (r_s_ = -0.20, p = 0.037). In parallel with the correlations with the symptom dimensions, a lower subject-loading on the *ECN-stress aftermath component* (IC7) was also associated with various measures, that are generally associated with worse health, such as: higher age (r_s_ = -0.56, p = 9.5E-23), higher BMI (r_s_ = -0.36, p = 4.6E-8), lower heart rate variability (r_s_ = 0.32, p = 5.7E-6), more physical pain (r_s_ = -0.22, p = 0.012) and worse experienced health (r_s_ = 0.20, p = 0.029).

Next, we performed post-hoc tests in order to explore if the correlations between the *ECN-stress aftermath component* (IC7) and the negative valence symptoms differed from the correlations between the other ECN-FPN components (i.e. IC8, IC13, and IC30) and these same negative valence symptoms (using Fisher's r to z transform; α = 0.05). We refer to Appendix 1 for a complete overview of these results. Here, we would like to note that the results showed that the correlations did not differ between the *ECN-stress aftermath component* (IC7) and IC13 (ECN during the resting-state scan) (IDS anxiety/somatic: IC7 r_s_ = -0.23; IC13: r_s_ = -0.14, z = -1.07, p_uncorrected_ = 0.284; ASI physical concerns: IC7: r_s_ = -0.20; IC13: r_s_ = -0.10; z = -1.17, p_uncorrected_ = 0.242). Based on the results of these post-hoc analyses, we cannot exclude that it is a threshold effect that we only found these relationships with negative valence symptoms for the *ECN-stress aftermath component*. At the same time it is important to note that in our analyses relatively few results for the symptom questionnaires survived multiple comparison correction, and that the *ECN-stress aftermath component* was the most sensitive component (of these ECN-FPN components) for finding relationships with the negative valence symptoms. Thus, the stress induction may have played an important role in revealing these results at a statistically significant level.”

The following text was added to Appendix 1:

“Correlations between ECN-FPN components and negative valence symptoms:

The *ECN-stress aftermath component* (IC7) was associated with the following symptom dimensions in the negative valence domain: anxiety/somatic (r_s_ = -0.23, p = 0.007) and physical concerns (r_s_ = -0.20, p = 0.037). Here, we provide the correlations of the other ECN-FPN components with these same symptom dimensions for comparison (NB: p-values are FDR corrected values): IC8: anxiety/somatic: r_s_ = -0.14, p = 0.286, physical concerns: r_s_ = -0.10, p = 0.551; IC13: anxiety/somatic: r_s_ = -0.14, p = 0.257, physical concerns: r_s_ = -0.10, p = 0.548; IC30: anxiety/somatic: r_s_ = -0.02, p = 0.964, physical concerns: r_s_ = -0.03, p = 0.923. Next, we performed post-hoc analyses to compare the strength of the correlations between the *ECN-stress aftermath component* (IC7) and the negative valence symptoms with the correlations between the other ECN-FPN components and these same symptoms (uncorrected for multiple comparisons). These analyses showed the following results related to the IDS anxiety/somatic: IC7 versus IC8: z = -1.12, p_uncorrected_ = 0.263; IC7 versus IC13: z = -1.07, p_uncorrected_ = 0.284; IC7 versus IC30: z = -2.60, p_uncorrected_ = 0.009. For ASI physical concerns: IC7 versus IC8: z = -1.18, p_uncorrected_ = 0.237; IC7 versus IC13: z = -1.17, p_uncorrected_ = 0.242; IC7 versus IC30: z = -2.04, p_uncorrected_ = 0.041. Since there is no significant difference between the correlations of the *ECN stress-aftermath component* and the ECN components during the resting-state scan (IC13) or stress scan (IC8), we cannot exclude a threshold effect. At the same time it is important to note that the *ECN-stress aftermath component* was the most sensitive for finding these relationships at statistically significant levels. Thus, the stress induction may have played an important role in revealing these results (during the stress-aftermath period).”

The following part was added to the discussion:

“The inclusion of a stress challenge is a novel feature in our linked ICA analysis. While several components that reflect the connectivity between the ECN and FPN (including the resting-state component) were associated with cognitive symptoms, two components that were related to the stress induction, were most sensitive in revealing relationships with negative valence symptoms (í.e. the *ECN-stress aftermath component* and *DMN-stress component*). This could indicate that it may be especially relevant to include a stress induction paradigm, when studying the negative valence domain.”

7. Related to the allostatic load reasoning, I think the possibility that the physiological/cardiac, metabolic, and general health problems contributed to the development of psychiatric problems can't be excluded. Also, because no measure of the duration of psychiatric problems was included in the presented results, whether this represents a 'load' effect is unclear. I think no causality and indication of allostatic load can be suggested here.

While we think that allostasis and allostatic load are important concepts that may help to explain our results, we also agree with the reviewer that based on our cross-sectional study we cannot conclude that allostatic load explains our results. Since the statements related to allostatic load are too hypothetical at this moment, we have removed this text. We think it is mainly important to emphasize the complex relationships between mental health, physical health and coping with stress, and the importance of an integrative approach to these factors. Based on this, we have adapted the discussion as described below.

The text in the discussion was changed to the following:

“Our extensive, exploratory analyses allowed us to identify not only a relationship of this ECN component with symptom dimensions, but also with a broad range of negative health outcomes related to physiological/cardiovascular measures, BMI, and also general health/functioning. It is known from the literature that there are complex relationships between mental health, physical health and coping with stress (de Kloet et al., 2005; Juster et al., 2010; McEwen, 2003). Physical and mental health problems may affect each other and may also result in a maladaptive stress response. Vice versa a repeatedly maladaptive stress response, including inadequate recovery in the aftermath of stress, may lead to mental and physical wear-and-tear (Juster et al., 2010; Mcewen, 2000; McEwen, 2003). As the brain is the central organ that coordinates the stress response (Ulrich-Lai and Herman, 2009), our results suggest the key importance of the ECN and FPN in this complex web of relationships. Together, these results emphasize the importance of an integrative approach to stress, and physical and mental health.”

8. How were medication use and substance use controlled in the analysis? Probably adding these latter as covariates is not optimal, given the high correlation between medication/substance use and various symptom (sub)scales and diagnostic classes. Some discussion on this influence is needed.

We agree with the reviewer that this is an important point. We also agree that using medication and substance use as covariates is not the best approach, because of the reasons mentioned above. We would like to emphasize that we did include variables for medication use (of broad classes of medication) and substance use (level of smoking, and use of cannabis and alcohol in the week before scanning (see Appendix 1-Table 2)) in the correlational analyses. Although these are rough measures for medication and substance use, we did investigate in this way if the brain components were related to these measures. The only relationship we found for these variables in our main analysis was between component 1 and antidepressant use. We think that future studies should try to further disentangle the effects of medication/substance use from the other factors we investigated in our current study. We think that this for example can be done by studies that include subgroups of participants that only differ in medication/substance use, and at the same time are matched on the measures of interest (like symptom dimensions). Based on this comment we have added additional information on medication use to our manuscript (to give the readers more insight into this) and we have also added a point to the limitation section in order to discuss the potential influence of medication use.

The following text was added to the manuscript:

“We refer to Appendix 1-Table 3 for information regarding psychotropic medication use at the time of the MRI scan.”

The following text (related to medication and substance use) was added to the limitations section in the discussion:

“Finally, although we did include measures related to psychotropic medication and substance use in our correlational analyses, we did not correct for these factors in the analyses. We did not include these factors as covariates, as these factors are not independent from our other measures of interest, like the symptom dimensions and diagnostic labels. Future studies in samples that are well matched on these measures of interest and only differ in medication status or substance use could help to further disentangle the effects of these factors from other aspects of psychopathology.”

9. Given the rationale and aims outlined in the introduction, the discussion could benefit from some more in-depth discussion of what the results mean for our understanding of the psychiatric disorder, how did they change it, what's new, and what next steps should we entail and what added value the current approach has (worth of multi-modal vs unimodal components for example).

We thank the reviewer for this suggestion. Based on this feedback we have added a paragraph to the discussion in which we discuss possible future directions. We think that this has improved the manuscript.

The following text was added as the last paragraph of the discussion:

“Our results provide initial insight into the neural mechanisms underlying transdiagnostic (bio)behavioral dimensions and provide avenues for future research. First, further research is necessary into which (biobehavioral) dimensions serve as reliable and valid measures for transdiagnostic research and how they can be assessed best (Morris et al., 2022). While various dimensions can be measured well with a self-report questionnaire, in other cases it may be important to complement these measures with more objective measures (e.g. neuropsychological tests or assessments by a trained clinician). In addition, the field of multimodal imaging is still relatively new. Multimodal imaging techniques have the potential to take maximal advantage of the strengths of the different types of imaging data, with each data type having a limited, but complementary view of the brain (Sui et al., 2012). Since our results show the potential of the functional scans for revealing relationships with transdiagnostic symptom dimensions, future studies should investigate how multimodal analysis techniques may further capitalize on this potential of the functional scans. It should be investigated which structural and functional modalities can best be combined and how multimodal analyses techniques can be optimized in order to further the integrative understanding of brain function and structure (Sui et al., 2012).”

10. Perhaps also include Morris et al. 2022 (https://doi.org/10.1186/s12916-022-02414-0) in addition to Cuthbert & Insel, 2013, to direct to a revisited version of the RDOC framework.

We thank the reviewer for this suggestion. Based on this suggestion we have added this reference in the introduction, when we first introduce the importance of the RDoC framework. In addition, we added this reference to a newly added text in the introduction, in which we substantiate why we chose to study psychopathology dimensionally across the range from health to psychopathology.

11. Here I get a bit lost: "The correlational analysis (Figure 2, operation D) resulted in 87 significant correlations (FDR corrected q <0.001) between the components and measures of interest (all p-values mentioned below are FDR corrected values (see Supplemental Table C for all significant correlations)). The analysis resulted in ten correlations between the ICs and symptom dimensions. Interestingly, all symptom correlations were discovered for components that were mainly driven by functional scans. Of the nineteen correlations that were found with multimodal components, most were related to age, sex, BMI, blood pressure, and heart rate variability, while there was also an association with cognitive symptoms and a classification of ASD."Maybe include: 'of those 87, ten related to symptom dimensions, 19 with etc". The confusion comes from mixing the findings on uni-modal vs multi-modal and findings related to the type of biobehavioral measure.

We thank the reviewer for this suggestion. We changed the text, so now we first start with discussing all results for our multimodal components. After this we focus more specifically on the significant results for the symptom dimensions. While reading this part of the Results section, it is important to keep in mind that there is also a multimodal component that is associated with symptom dimensions (IC30 is associated with inhibition and self-monitoring; this is a multimodal component that is mainly driven by the following functional scans: resting-state scan and stress-aftermath scan). So, it is not possible to completely separate the discussion on the unimodal vs multimodal components from the findings related to the type of biobehavioral measure.

We have changed the text in the Results section to the following:

“The correlational analysis (Figure 2, operation D) resulted in 87 significant correlations (FDR corrected q <0.001) between the components and measures of interest (all p-values mentioned below are FDR corrected values (unless mentioned otherwise) (see Supplementary File 2a for all significant correlations)). Of these 87 correlations, nineteen were with multimodal components. Most of these correlations were related to age, sex, BMI, blood pressure, and heart rate variability. In addition, we identified a multimodal component that was associated with a classification of ASD (IC32) and a multimodal component associated with cognitive symptoms (inhibition and self-monitoring) (IC30). In addition, there were eight more significant correlations between the ICs and symptom dimensions, which were all with unimodal components (See Appendix 1-figure 1 for scatterplots displaying all ten significant correlations between ICs and symptom dimensions). Interestingly, all symptom correlations were discovered for components that were mainly driven by functional scans.”

12. I don't understand this sentence "The DMN spatial map displayed within DMN connectivity (angular gyrus, precuneus, supramarginal gyrus)." Line 157/158. This sentence also seems incomplete "Additionally, the VBM feature weighted in the precentral gyrus and the MD feature in the thalamus (Figure 4B). (lines 162/163).

With respect to the ‘within DMN connectivity’: we referred here to the DMN (stress-aftermath) modality that is part of this multimodal component and reflects the DMN connectivity during the stress-aftermath scan. More specifically, this is a spatial map that reflects the connectivity of the DMN network template that was derived from Smith and colleagues 2009 (Smith et al., 2009) and was applied into dual regression (see the following paragraph in the methods section ‘MRI preprocessing and feature extraction’, and see also the text in Appendix 1 regarding the methods for the feature extraction for the functional scans). Since this DMN modality shows connectivity with brain regions that are part of/commonly associated with the DMN, this result reflects connectivity of the DMN with itself (i.e. within DMN connectivity). This finding is reported in the paragraph of the Results section that describes the results for the *multimodal ASD component*. Based on point 5 of reviewer 2, we now also mention explicitly in this paragraph the contribution of all 12 modalities to this *multimodal ASD component*. Now that we added this to the main text, we now also mention in the main text that the DMN modalities during the different functional scans (i.e. resting-state, stress and stress-aftermath scan) showed similar connectivity patterns, meaning that the different functional scans identified the same connectivity pattern (in the original manuscript we mentioned this only in the Figure legend). Based on this feedback, we have adjusted the text to further clarify the finding related to the ‘within DMN connectivity’ for the readers. Finally, we have also adjusted the other sentence (mentioned by the reviewer), for further clarification.

We have changed the text in the Results section to the following:

“The DMN modalities during the different functional scans (i.e. resting-state, stress and stress-aftermath scan) showed similar spatial configurations, meaning that these different functional scans identified the same connectivity pattern. The DMN modalities revealed loadings in multiple brain regions that are part of (or commonly associated with) the DMN (i.e. the angular gyrus, precuneus, supramarginal gyrus). Thus this reflects connectivity of the DMN network template that was applied into dual regression with DMN (associated) regions (i.e. within DMN connectivity).”

We have changed the text in the Results section to the following:

“Additionally, the VBM and MD feature showed the involvement of the precentral gyrus and thalamus respectively (Figure 4B).”

13. Anxiety (symptoms and sensitivity) and sleep problems seem to be pretty low, as are mood symptoms. Are distributions given somewhere else (outside Table 1)?

Based on this point (and also the other points raised by the reviewers), we think it is important to give further insight into the distribution of the data from the symptom questionnaires. Since the scores on the symptom questionnaires are not normally distributed we display the median and range of the scores in Table 1. Given the diversity of our sample and the accompanying diversity in scores on the symptom questionnaires we realize that the median and range may give too limited insight into these symptoms. In addition, the median may for example be influenced by a large proportion of the participants having relatively low (or high) scores on a questionnaire. We have now made dot plots (Appendix 1-figure 3) in order to visualize the distribution of the symptom dimensions in our sample. Based on (among others) point 2 of reviewer 2, we have made these dot plots for broad diagnostic subgroups in our sample. For this purpose, we have divided our sample in the following 4 broad, heuristic subgroups: (1) mentally healthy control group, (2) neurodevelopmental group, (3) stress-related group and (4) the comorbidity group. We refer to the answer to point 2 of reviewer 2 for a more elaborate description regarding these subgroups. We think that these dot plots illustrate the diversity of our sample, and the variation across participants related to how strongly the symptoms are present that are measured with the various symptom questionnaires (like the mood symptoms on the IDS-SR questionnaire). Taken together, we think that this new Figure gives more insight into our results and is an important addition to the manuscript.

14. Related to participants, given this information is in the supplement, could it be added when and how patients were recruited? Related to their in/out-patient treatment? Was remission allowed? How was the distribution of past/current diagnoses in each group? What was the correlation between all symptomatic and biobehavioural markers? Could a correlational table/graph help here? Also the relation with medication helps the reader to put everything into context.

We agree with the reviewer that the addition of this information helps to better understand the sample and to put the results into context. All patients were recruited from our outpatient clinic. Patients were only included if they had at least one current diagnosis and all diagnoses displayed in Figure 1 are current diagnoses. We have added this information and also information related to the other questions to the manuscript (see below).

The following text was added to the methods section:

“This study uses the database of the MIND-Set project (van Eijndhoven et al., 2021), which includes adult patients with ASD, ADHD, addiction, mood, and/or anxiety disorders. Patients were included in the present study if they had at least one current diagnosis in one of these categories.”

The following text was added to Figure 1 (venn diagram):

“All diagnoses in this Venn diagram represent current diagnoses.”

The text was changed in Appendix 1 to the following:

“MIND-Set (Measuring Integrated Novel Dimensions in Neurodevelopmental and Stress-related Mental Disorders) is an observational cross-sectional study. Inclusion of patients took place at the moment of intake at the outpatient clinic of the psychiatry department of the Radboud university medical center (Radboudumc), Nijmegen, the Netherlands. The MIND-Set study has been approved by the Ethical Review Board of the Radboudumc (Nijmegen, the Netherlands) and all participants signed an informed consent form before participation (van Eijndhoven et al., 2021). Data for our present study were collected from June 2016 through July 2020.”

The following text was added to the manuscript (to refer to the Table in which we now display all correlations between our biobehavioral measures of interest):

“Finally, we performed Spearman correlations (non-normal distribution) between all our biobehavioral measures of interest to provide further insight into these relationships (see Supplementary File 1).”

References

Adler L, Kessler R. 2004. Adult Self-Report Scale (ASRS) version 1.1.

Ayub R, Sun KL, Flores RE, Lam VT, Jo B, Saggar M, Fung LK. 2021. Thalamocortical connectivity is associated with autism symptoms in high-functioning adults with autism and typically developing adults. *Transl Psychiatry* 11. doi:10.1038/s41398-021-01221-0

Baron-Cohen S, Hoekstra RA, Knickmeyer R, Wheelwright S. 2006. The Autism-Spectrum Quotient (AQ) – Adolescent Version. *J Autism Dev Disord* 36:343–350. doi:10.1007/s10803-006-0073-6

Baron-Cohen S, Wheelwright S, Skinner R, Martin J, Clubley E. 2001. The Autism-Spectrum Quotient (AQ): Evidence from Asperger Syndrome/High-Functioning Autism, Males and Females, Scientists and Mathematicians. *J Autism Dev Disord* 31:5–17. doi:10.1023/A:1005653411471

Comon P, Jutten C. 2010. Handbook of Blind Source Separation: Independent Component Analysis and Applications. Academic Press.

Cowdrey FA, Filippini N, Park RJ, Smith SM, Mccabe C. 2014. Increased resting state functional connectivity in the default mode network in recovered anorexia nervosa. *Hum Brain Mapp* 35:483–491. doi:10.1002/hbm.22202

de Kloet ER, Joëls M, Holsboer F. 2005. Stress and the brain: from adaptation to disease. *Nat Rev Neurosci* 6:463–75. doi:10.1038/nrn1683

Duffield T, Trontel H, Bigler ED, Froehlich A, Prigge MB, Travers B, Green RR, Cariello AN, Cooperrider J, Nielsen J, Alexander A, Anderson J, Fletcher PT, Lange N, Zielinski B, Lainhart J. 2013. Neuropsychological investigation of motor impairments in autism. *J Clin Exp Neuropsychol* 35:867–881. doi:10.1080/13803395.2013.827156.Neuropsychological

Filippini N, Macintosh BJ, Hough MG, Goodwin GM, Frisoni GB, Smith SM, Matthews PM, Beckmann CF, Mackay CE. 2009. Distinct patterns of brain activity in young carriers of the APOE-ε4 allele 106:7209–7214.

First MB, Spitzer RL, Gibbon M, Williams JBW. 1996. Structured clinical interview for DSM-IV axis I disorders, clinician version (SCID-CV). Washington, DC.

Francx W, Llera A, Mennes M, Zwiers MP, Faraone S V., Oosterlaan J, Heslenfeld D, Hoekstra PJ, Hartman CA, Franke B, Buitelaar JK, Beckmann CF. 2016. Integrated analysis of gray and white matter alterations in attention-deficit/hyperactivity disorder. *NeuroImage Clin* 11:357–367. doi:10.1016/j.nicl.2016.03.005

Franke B, Michelini G, Asherson P, Banaschewski T, Bilbow A, Buitelaar JK, Cormand B, Faraone S V, Ginsberg Y, Haavik J, Kuntsi J, Larsson H, Lesch KP, Ramos-Quiroga JA, Réthelyi JM, Ribases M, Reif A. 2018. Live fast, die young? A review on the developmental trajectories of ADHD across the lifespan. *Eur Neuropsychopharmacol* 28:1059–1088. doi:10.1016/j.euroneuro.2018.08.001

Good CD, Johnsrude IS, Ashburner J, Henson RNA, Friston KJ, Frackowiak RSJ. 2001. A voxel-based morphometric study of ageing in 465 normal adult human brains. *Neuroimage* 14:21–36. doi:10.1006/nimg.2001.0786

Groves AR, Beckmann CF, Smith SM, Woolrich MW. 2011. Linked independent component analysis for multimodal data fusion. *Neuroimage* 54:2198–2217. doi:10.1016/j.neuroimage.2010.09.073

Groves AR, Smith SM, Fjell AM, Tamnes CK, Walhovd KB, Douaud G, Woolrich MW, Westlye LT. 2012. Benefits of multi-modal fusion analysis on a large-scale dataset: Life-span patterns of inter-subject variability in cortical morphometry and white matter microstructure. *Neuroimage* 63:365–380. doi:10.1016/j.neuroimage.2012.06.038

Hermans EJ, Marle HJF Van, Ossewaarde L, Henckens MJAG, Qin S, Kesteren MTR Van, Schoots VC, Cousijn H, Rijpkema M, Oostenveld R, Fernández G. 2011. Stress-Related Noradrenergic Activity Prompts Large-Scale Neural Network Reconfiguration. *Science (80- )* 334:1151–1153. doi:10.1126/science.1209603

Ingram RE, Luxton DD. 2005. Dvelopment of psychopathology: A vulnerability-stress perspective. Chapter: vulnerability-stress models. CA: Sage.

Itahashi T, Yamada T, Nakamura M, Watanabe H, Yamagata B, Jimbo D, Shioda S, Kuroda M, Toriizuka K, Kato N, Hashimoto R. 2015. Linked alterations in gray and white matter morphology in adults with high-functioning autism spectrum disorder: A multimodal brain imaging study. *NeuroImage Clin* 7:155–169. doi:10.1016/j.nicl.2014.11.019

Juster R-P, McEwen BS, Lupien SJ. 2010. Allostatic load biomarkers of chronic stress and impact on health and cognition. *Neurosci Biobehav Rev* 35:2–16. doi:10.1016/j.neubiorev.2009.10.002

Kerns CM, Newschaffer CJ, Berkowitz SJ. 2015. Traumatic Childhood Events and Autism Spectrum Disorder. *J Autism Dev Disord* 45:3475–3486. doi:10.1007/s10803-015-2392-y

Kessler RC, Adler L, Ames M, Delmer O, Faraone S, Hiripi E, Howes MJ, Jin R, Secnik K, Spencer T, Ustun TB, Walters EE. 2005. The World Health Organization Adult ADHD Self-Report Scale (ASRS): A Short Screening Scale for Use in the General Population. *Psychol Med* 35:245–256.

Kim J, Lee E, Joung Y. 2013. The WHO Adult ADHD Self-Report Scale: reliability and validity of the Korean version. *Psychiatry Investig* 10:41–46.

Koob GF. 2003. Alcoholism: Allostasis and beyond. *Alcohol Clin Exp Res* 27:232–243. doi:10.1097/01.ALC.0000057122.36127.C2

Koob GF, Schulkin J. 2019. Addiction and stress: An allostatic view. *Neurosci Biobehav Rev* 106:245–262. doi:10.1016/j.neubiorev.2018.09.008

Kooij JJ, Francken MH. 2010. Diagnostic Interview for ADHD in Adults Version 2.0 (Dutch: Diagnostisch Interview voor ADHD bij volwassenen versie 2.0 (DIVA 2.0)). The Hague, The Netherlands: DIVA Foundation.

Kotov R, Krueger RF, Watson D, Bagby M, Carpenter WT, Caspi A. 2017. The hierarchical taxonomy of psychopathology (HiTOP): A dimensional alternative to traditional nosologies. *J Abnorm Psychol* 1–48.

Mahajan R, Dirlikov B, Crocetti D, Mostofsky SH. 2016. Motor Circuit Anatomy in Children with Autism Spectrum Disorder With or Without Attention Deficit Hyperactivity Disorder. *Autism Res* 9:67–81. doi:10.1002/aur.1497

Mcewen BS. 2000. Allostasis and Allostatic Load: Implications for Neuropsychopharmacology. *Neuropsychopharmacology* 22:108–124.

McEwen BS. 2003. Mood disorders and allostatic load. *Biol Psychiatry* 54:200–207. doi:10.1016/S0006-3223(03)00177-X

McTeague LM, Goodkind MS, Etkin A. 2016. Transdiagnostic impairment of cognitive control in mental illness. *J Psychiatr Res* 83:37–46. doi:10.1016/j.jpsychires.2016.08.001

Mei T, Forde NJ, Floris DL, Dell’Acqua F, Stones R, Ilioska I, Durston S, Moessnang C, Banaschewski T, Holt RJ, Baron-Cohen S, Rausch A, Loth E, Oakley B, Charman T, Ecker C, Murphy DGM, Group the E-AL, Beckmann CF, LLera A, Buitelaar JK, Brandeis D. 2022. Autism Is Associated With Interindividual Variations of Gray and White Matter Morphology. *Biol Psychiatry Cogn Neurosci Neuroimaging*. doi:https://doi.org/10.1016/j.bpsc.2022.08.011

Menon V. 2011. Large-scale brain networks and psychopathology: a unifying triple network model. *Trends Cogn Sci* 15:483–506. doi:10.1016/j.tics.2011.08.003

Morris SE, Sanislow CA, Pacheco J, Vaidyanathan U, Gordon JA, Cuthbert BN. 2022. Revisiting the seven pillars of RDoC. *BMC Med* 20:1–11. doi:10.1186/s12916-022-02414-0

Mulders PCR, van Eijndhoven PFP, van Oort J, Oldehinkel M, Duyser FA, Kist JD, Collard RM, Vrijsen JN, Haak K V., Beckmann CF, Tendolkar I, Marquand AF. 2022. Striatal connectopic maps link to functional domains across psychiatric disorders. *Transl Psychiatry* 12:1–7. doi:10.1038/s41398-022-02273-6

Murdaugh DL, Shinkareva S V., Deshpande HR, Wang J, Pennick MR, Kana RK. 2012. Differential Deactivation during Mentalizing and Classification of Autism Based on Default Mode Network Connectivity. *PLoS One* 7. doi:10.1371/journal.pone.0050064

Pietzuch M, Bindoff A, Jamadar S, Vickers JC. 2021. Interactive effects of the APOE and BDNF polymorphisms on functional brain connectivity: the Tasmanian Healthy Brain Project. *Sci Rep* 11:1–13. doi:10.1038/s41598-021-93610-0

Ramos-Quiroga JA, Nasillo V, Richarte V, Corrales M, Palma F, Ibáñez P, Michelsen M, Van de Glind G, Casas M, Kooij JJS. 2016. Criteria and Concurrent Validity of DIVA 2.0: A Semi-Structured Diagnostic Interview for Adult ADHD. *J Atten Disord*. doi:10.1177/1087054716646451

Richey J Anthony, Damiano CR, Sabatino A, Rittenberg A, Petty C, Bizzell J, Voyvodic J, Heller AS, Coffman MC, Smoski M, Davidson RJ, Dichter GS. 2015. Neural Mechanisms of Emotion Regulation in Autism Spectrum Disorder. *J Autism Dev Disord* 45:3409–3423. doi:10.1007/s10803-015-2359-z

Richey J.A., Petty C, Bs B, Bizzell J, Ms B, Voyvodic J. 2015. Neural mechanisms of emotion regulation in autism spectrum disorder. *J Autism Dev Disord* 45:3409–3423. doi:10.1007/s10803-015-2359-z.Neural

Rommelse NNJ, Geurts HM, Franke B, Buitelaar JK, Hartman CA. 2011. A review on cognitive and brain endophenotypes that may be common in autism spectrum disorder and attention-deficit / hyperactivity disorder and facilitate the search for pleiotropic genes. *Neurosci Biobehav Rev* 35:1363–1396. doi:10.1016/j.neubiorev.2011.02.015

Schippers GM, Broekman TG. 2012. MATE-Crimi 2.1. Handleiding en protocol. Nederlandse bewerking: G.M. Schippers & T.G. Broekman.

Schippers GM, Broekman TG, Buchholz A, Koeter MWJ, Brink W Van Den. 2010. Measurements in the Addictions for Triage and Evaluation (MATE): An Instrument Based on the WHO Family of International Classifications. *Addiction* 105:862–871. doi:10.1111/j.1360-0443.2009.02889.x

Schmidt A, Denier N, Magon S, Radue EW, Huber CG, Riecher-Rossler A, Wiesbeck GA, Lang UE, Borgwardt S, Walter M. 2015. Increased functional connectivity in the resting-state basal ganglia network after acute heroin substitution. *Transl Psychiatry* 5. doi:10.1038/TP.2015.28

Sharma S, Powers A, Bradley B, Ressler KJ. 2016. Gene × Environment Determinants of Stress- and Anxiety-Related Disorders. doi:10.1146/annurev-psych-122414-033408

Shaw P, Stringaris A, Nigg J, Leibenluft E. 2014a. Emotion Dysregulation in Attention Deficit Hyperactivity Disorder. *Am J Psychiatr Rehabil* 276–293.

Shaw P, Stringaris A, Nigg J, Leibenluft E. 2014b. Emotion Dysregulation in Attention Deficit Hyperactivity Disorder. *Am J Psychiatry* 171:276–293. doi:10.1176/appi.ajp.2013.13070966

Smith SM, Fox PT, Miller KL, Glahn DC, Fox PM, Mackay CE, Filippini N, Watkins KE, Toro R, Laird AR, Beckmann CF. 2009. Correspondence of the brain’s functional architecture during activation and rest. *Proc Natl Acad Sci U S A* 106:13040–5. doi:10.1073/pnas.0905267106

Sui J, Adali T, Li Y-O, Yang H, Calhoun VD. 2012. A review of multivariate methods for multimodal fusion of brain imaging data. *Med Imaging 2010 Biomed Appl Mol Struct Funct Imaging* 204:68–81. doi:10.1016/j.jneumeth.2011.10.031

Ulrich-Lai YM, Herman JP. 2009. Neural regulation of endocrine and autonomic stress responses. *Nat Rev Neurosci* 10:397–409. doi:10.1038/nrn2647

van Eijndhoven PFP, Collard RM, Vrijsen JN, Geurts DGM, Arias-Vasquez A, Schellekens AFA, van den Munckhof E, Brolsma SCA, Duyser FA, Bergman A, van Oort J, Tendolkar I, Schene AH. 2021. Measuring Integrated Novel Dimensions in Neurodevelopmental and Stress-related Mental Disorders (MIND-Set): a cross-sectional comorbidity study from an RDoC perspective. *medRxiv*. https://www.medrxiv.org/content/10.1101/2021.06.05.21256695v1

van Oort J, Kohn N, Vrijsen JN, Collard R, Duyser FA, Brolsma SCAA, Fernández G, Schene AH, Tendolkar I, van Eijndhoven PF. 2020. Absence of default mode downregulation in response to a mild psychological stressor marks stress-vulnerability across diverse psychiatric disorders. *NeuroImage Clin* 25:102176. doi:10.1016/j.nicl.2020.102176

van Oort J, Tendolkar I, Collard R, Geurts DEM, Vrijsen JN, Duyser FA, Kohn N, Fernández G, Schene AH, van Eijndhoven PFP. 2022. Neural correlates of repetitive negative thinking: Dimensional evidence across the psychopathological continuum. *Front Psychiatry* 13. doi:10.3389/fpsyt.2022.915316

Vuijk R. 2014. Diagnostic Interview for ASD in Adults (Dutch: Nederlands Interview ten behoeve van Diagnostiek Autismespectrumstoornis bij Volwassenen (NIDA)).

Woody ML, Gibb BE. 2015. Integrating NIMH Research Domain Criteria (RDoC) into Depression Research. *Curr Opin Psychol* 4:6–12. doi:10.1016/j.copsyc.2015.01.004